 

Registered report

health and disease and epidemiology/psychology

COVID-19, worry, stress, compliance behaviour, trust, social psychology

# Stress and worry in the 2020 coronavirus pandemic: relationships to trust and compliance with preventive measures across 48 countries in the COVIDiSTRESS global survey

Andreas Lieberoth[1,2], Shiang-Yi Lin[3], Sabrina Stöckli[4], Hyemin Han[5], Marta Kowal[6], Rebekah Gelpi[7], Stavroula Chrona[8], Thao Phuong Tran[9], Alma Jeftić[10], Jesper Rasmussen[11], Huseyin Cakal[12], Taciano L. Milfont[13] and The COVIDiSTRESS global survey consortium

[1]School of Culture and Society (Interacting Minds Center), and [2]Danish School of Education (DPU), Aarhus University, Aarhus, Denmark
[3]Hong Kong Institute of Education, Education University of Hong Kong, New Territories, Hong Kong
[4]University of Bern, Bern, Switzerland
[5]Educational Psychology Program, University of Alabama, Tuscaloosa, AL, USA
[6]Wroclaw University Institute of Psychology, Wroclaw 50-527, Poland
[7]Department of Psychology, University of Toronto, Toronto, Ontario, Canada
[8]Department of European and International Studies, King's College London, London, UK
[9]Department of Psychology, Colorado State University, Fort Collins, CO, USA
[10]Peace Research Institute, International Christian University, Mitaka, Tokyo, Japan
[11]Department of Political Science, Aarhus University, Aarhus, Denmark
[12]School of Psychology, Keele University, Keele, Staffordshire, UK
[13]School of Psychology, University of Waikato, Wellington, New Zealand

AL, 0000-0003-0214-5791; SS, 0000-0002-8322-2906; HH, 0000-0001-7181-2565; MK, 0000-0001-9050-1471; RG, 0000-0001-6899-0520; SC, 0000-0002-0914-4938; TPT, 0000-0002-4038-8471; AJ, 0000-0002-9285-2061; JR, 0000-0002-0248-7065; HC, 0000-0002-6227-9698

**Author for correspondence:**
Andreas Lieberoth
e-mail: andreas@edu.au.dk

The COVIDiSTRESS global survey collects data on early human responses to the 2020 COVID-19 pandemic from 173 429 respondents in 48 countries. The open science study was co-designed by an international consortium of researchers to investigate how psychological responses differ across countries and cultures, and how this has impacted behaviour, coping and trust in government efforts to slow the spread of the virus.

Starting in March 2020, COVIDiSTRESS leveraged the convenience of unpaid online recruitment to generate public data. The objective of the present analysis is to understand relationships between psychological responses in the early months of global coronavirus restrictions and help understand how different government measures succeed or fail in changing public behaviour. There were variations between and within countries. Although Western Europeans registered as more concerned over COVID-19, more stressed, and having slightly more trust in the governments' efforts, there was no clear geographical pattern in compliance with behavioural measures. Detailed plots illustrating between-countries differences are provided. Using both traditional and Bayesian analyses, we found that individuals who worried about getting sick worked harder to protect themselves and others. However, concern about the coronavirus itself did not account for all of the variances in experienced stress during the early months of COVID-19 restrictions. More alarmingly, such stress was associated with less compliance. Further, those most concerned over the coronavirus trusted in government measures primarily where policies were strict. While concern over a disease is a source of mental distress, other factors including strictness of protective measures, social support and personal lockdown conditions must also be taken into consideration to fully appreciate the psychological impact of COVID-19 and to understand why some people fail to follow behavioural guidelines intended to protect themselves and others from infection. The Stage 1 manuscript associated with this submission received in-principle acceptance (IPA) on 18 May 2020. Following IPA, the accepted Stage 1 version of the manuscript was preregistered on the Open Science Framework at https://osf.io/ytbcs. This preregistration was performed prior to data analysis.

# 1. Background and research questions

This registered report (RR) presents the COVIDiSTRESS global survey, which collects early data on human responses to the 2020 COVID-19 pandemic (caused by the coronavirus SARS-CoV-2) in 48 countries. This open science study was co-designed by researchers from numerous universities across the world (for a full list of universities, see §2; for a full list of consortium co-authors, see electronic supplementary material, co-authorship statement and appendix) to investigate how psychological and behavioural responses to the COVID-19 pandemic differed across countries and cultures, and how this impacted social behaviour, coping and trust in government efforts to slow the spread of the coronavirus.

Global data gathering efforts like the COVIDiSTRESS project are of pressing importance during the COVID-19 crisis, in order to understand how different government measures succeed or fail in achieving objectives to change public behaviour. By generating data rapidly, and disseminating them continuously as open data on the Open Science Framework (OSF), the international research network behind the COVIDiSTRESS project aims to yield rapid insights for the use of practitioners such as health communicators, government officials and policy makers at all levels, as well as members of the academic community. Currently, there are many psychologists and social scientists making a great effort to better understand the psychological effects and implications of the COVID-19 pandemic (e.g. [1–3]) and provide suggestions for practitioners (e.g. [4–6]). To our knowledge, there is currently no other project comparing human responses to COVID-19 across such a large number of countries. Further, the COVIDiSTRESS project specifically attempted to leverage the convenience of unpaid online populations to quickly acquire a large number of responses from 48 countries by using a snowball sample via online and traditional media platforms, rather than pre-specified samples from paid subject pools.

1. What is the impact of psychological stress on trust in and compliance with government efforts to prevent COVID-19 compared across countries?
2. To what degree is psychological stress related to concerns over COVID-19 itself, and what proportion of the variance in psychological stress can be explained by other factors?

This registered report (RR) analyses data from a swiftly executed global survey study meant to allow both hypothesis testing and exploratory cross-country comparisons. As an open science project, the COVIDiSTRESS effort as a whole intends to supply data to the public, allowing researchers and other stakeholders to independently conduct their own analyses [7]. As such, the present RR investigates 10 central hypotheses, focusing on people's experiences and behaviours during the early COVID-19 crisis. The present RR does not include a number of other variables from the COVIDiSTRESS global survey.

In early 2020, the COVID-19 pandemic quickly became a matter of serious concern for public health throughout the global community. To prevent the spread of coronavirus, governments imposed a range of measures, which included quarantines, physical distancing and limits to civil liberties. Inevitably, these

changes have generated a variety of psychological responses [8], which in turn shape the level of compliance with preventive efforts. Extant research on the factors that shape willingness to comply with public health efforts has highlighted the importance of psychological responses such as anxiety and risk aversion [9–11] as well as trust in state authorities [12–14]. A recent study on risk and compliance during the COVID-19 pandemic in Singapore found that high levels of trust in the government led to lower risk perception and non-compliance with preventive measures [11]. Along the same lines, a study on the implications of COVID-19 in New Zealand found an increase in levels of institutional trust in the early stages of the lockdown pointing to the rally-round-the-flag effect [2]. This varied body of findings clearly signifies the urgent need for further exploration of the psychological mechanisms at play that shape responses to preventive measures.

Not only do pandemics foster a fear of contagion, but also misinformation, uncertainty and the lack of clarity about how to react [15,16]. Concern over a disease outbreak has typically been found to be positively correlated with preventive behaviours or compliance with health guidelines [17–19]. However, this relationship is complex; worry, for instance, has been found to be related to trust in disease-related information and media [20–22], the perception of openness and reliability of governments and health organizations [23].

Concern over one's medical risk during a pandemic can also be a source of ongoing stress (e.g. in H1N1, [24]; in MERS, [22]). For instance, stress was widespread during the 2003 SARS epidemic [25,26]. This relationship is potentially exacerbated by the fact that even some of the most efficient methods of slowing the spread of a disease, such as self-isolation and quarantine, also take an immediate and potentially long-lasting psychological toll among affected populations [8,27,28]. Surveys of quarantined healthcare workers [29] and members of the general population [30] following the 2003 SARS outbreak found evidence of acute stress responses in cases with a longer duration of quarantine and greater perceived difficulty in compliance with recommendations. Similarly, studies of patient groups typically find a negative relationship between experienced stress and compliance with prescribed behaviours, often with direct medical consequences [31–33], and this correlation has been extended to compliance with measures intended to minimize the spread of virulent disease [34].

Both the medical situation and the psychological effects of isolation and confinement [35,36] thus need to be taken into consideration when prolonged periods of quarantine are implemented, in order to understand the acceptance of government measures and compliance with preventive measures. Compliance with medical guidelines has been shown to decrease not just as a result of heightened stress levels [37], but also of minor everyday stressors such as workplace conflict or household responsibilities [38]. Efforts such as closing down schools and workplaces and calls for people to self-isolate in their homes are likely to constitute a source of both existential and practical stress, unrelated to the fear of contracting the disease. For instance, some stress is known to result from an inability to participate in work or from interpersonal problems in the family [39,40]. Moreover, a pandemic may also exacerbate existing financial precarity, generating another source of stress for those experiencing financial hardship [10]. Prolonged states of emergency and the chronic psychological, social and economic stressors related to them [8,27] may decrease compliance with behavioural objectives during pandemics. Sources of stress and worry may thus have direct implications not simply for well-being, but also for the effectiveness of collective efforts and the degree of public compliance with self-isolation measures. For this reason, we tested the hypotheses that stress and concern over the coronavirus will be adversely correlated with compliance and preventive behavioural measures suggested in each country.

In sum, it seems that worry, trust and situational stress are associated with each other and jointly influence behaviour during pandemics. It is thus an interesting question whether the known negative relationship between stress and compliance apply to the population at large, especially since, for example, physical distancing measures and hand-washing are recommended to protect not only the individual in question, but also other people in their surroundings. The extent of compliance with preventive measures has repeatedly been found to be influenced by the perceived gravity of the pandemic, as well as trust in the preventive measures against it [18,24,41,42]. Thus, we also expected that trust in the preventive measures taken in each country will lead to better compliance and less worry over the COVID-19 medical situation.

In this context, it is also worth investigating what factors may alleviate perceived stress, and thus, positively impact not just well-being, but also the likelihood of acceptance and compliance with behavioural guidelines. Social and practical support from groups such as one's family, friends and colleagues can moderate the effect of concern for the disease or other sources of stress on one's psychological well-being [43,44]. Indeed, interpersonal communication has been shown to affect the

relationship between risk perception and actual behaviour during health crises, where mass media messages may fall short [45]. Various theoretical dimensions of social support, including social integration and reassurance of worth [46] are captured by the Social Provisions Scale (SPS) and its shorter versions (SPS-10; [47]). According to the stress-buffering hypothesis [48], social support through, for example, interpersonal communication buffers people from other negative consequences of distressing events. For instance, those who were 'isolated but not lonely' [3] might be less susceptible to negative effects of isolation or other stressors related to a pandemic by maintaining strong social contacts while physically distanced. Further, social support could promote less threatening interpretations of stressful situations [47,49], and thus, possibly more positive appraisals of institutional and governmental efforts to stop the spread of a pandemic. Empirical evidence shows that perceived social support has helped build up resilience against disaster or catastrophic events, and exerts its protective function on alleviating psychological distress [50]. Furthermore, social support has been found to correlate positively with compliance in patient populations [51].

It is thus reasonable to investigate if we can extend the previous findings related to the stress-buffering hypothesis to the context of acceptance and compliance with behavioural guidelines during the pandemic. Here, we explored the relationship between experienced stress and trust in the efforts and behavourial guidelines set up by governments to curb the spread of COVID-19, as well as in relation to social provisions. In particular, we hypothesized that social provisions would negatively moderate the negative effect of stress on compliance with behavioural objectives during the pandemic.

So far, little is known about the cross-national differences in reactions to preventive measures during the 2020 pandemic. However, cross-cultural differences have been observed in responding to stressors [52,53] and to various countermeasures against pandemic outbreaks [54]. Further, it has been suggested that cultural variables such as the degree of collectivism, or social 'tightness', may lead to different attitudes regarding compliance with social norms [3], and one study observed that cultural differences in relational mobility, or the perceived easiness to engage with strangers and freely choose friends, was associated with the spread of COVID-19 [55]. Thus, optimal institutional responses to the COVID-19 crisis may require culturally and situationally appropriate public health interventions, both to alleviate human distress and to communicate the need for preventive measures in an effective manner. In the present study, we took a multilevel analytic approach with individual responses nested within each country in order to estimate the variability in various dependent variables (DVs) residing at both individual and country levels [56]. By taking into consideration non-independence among individual responses within a country, as well as the differences between countries, we can better assess the extent to which individual-level variables measured in the present study explain variance in various criterion variables at the individual level and country level, respectively.

## 2. Methods

A global survey study was conducted with the purpose of mapping the psychological and behavioural responses to the COVID-19 pandemic across countries, and how these have impacted social behaviour and trust in government efforts to slow the spread of the coronavirus.

The COVIDiSTRESS survey (COVIDiSTRESS global survey network, 2020—available at https://osf.io/z39us/) consists of two parts. The first part collected general demographic data including proximate effects of the COVID-19 pandemic (e.g. isolation status, first-hand experience, attenuated risk), personality traits and self-reported variables, such as loneliness and perceived stress [57], daily behaviours including compliance with general and social preventive measures. The second part contains sets of more specific items related to distress, coping and social provisions. Firstly, we inquired about people's experiences of distress and worry during the ongoing outbreak of coronavirus, e.g. access to amenities, loss of work, adapting work, education and social interactions on digital platforms, the social stresses of confinement with adults and children. We also asked questions about the factors people experience as soothing and positive to coping (e.g. social contact, staying informed, dedicating oneself to preparation, hobbies, religion) and also included the Social Provisions Scale (SPS; [47]). Validated short versions of established measures were used whenever possible and available in local languages. In order to protect participants' data and avoid sensitive information, participants were not asked about COVID-19 symptoms or other data with direct medical implications that would allow third-parties to identify them.

Translations were completed for 46 languages and locations (Afrikaans, Albanian, Arabic, Bangla, Indonesian, Bosnian, Bulgarian, Chinese [Simplified and Traditional], Croatian, Czech, Danish, Dutch

[Belgium, Netherlands], English, Spanish [Argentina, Colombia, Cuba, Mexico, Spain], Filipino, Finnish, French, German, Greek [Greece and Cyprus], Hebrew, Hindi, Hungarian, isiXhosa, isiZulu, Italian, Japanese, Korean, Lithuanian, Nepali, Persian, Polish, Portuguese [Brazil, Portugal], Romanian, Russian, Slovakian, Serbian, Turkish, Urdu, Vietnamese) with the possibility for more translations in future waves. Translations were completed by a forward translator from the original English version and then validated through both panel and back-translation process by separate translators when possible.

Given the urgent call for COVID-19 research, the survey received a waiver to commence data collection from the IRB office at Aarhus University, Denmark, followed by formal approval in June 2020 (case number: 2019-616-000009). The survey and data extraction schedule was pre-registered at the Open Science Framework [58]. The full annotated dataset has also been published in *Scientific Data* [59].

# 3. Key independent and dependent variables

The COVIDiSTRESS global survey contains a number of variables (full survey at https://osf.io/z39us/). The Royal Society Open Science registration files including the *in principle accepted* stage one analysis protocol and electronic supplementary material are available at the Open Science Framework https://osf.io/vz73d/. The final version of the cleaned and preprocessed dataset is available at https://osf.io/cjxua/. This urgent call RR focuses on 10 hypotheses (see §§1 and 4) and comparisons between countries. Table 1 gives an overview of all variables relevant for our hypotheses.

# 4. Hypotheses

First, this RR was intended to draw a map of differences in the psychological impact of COVID-19, and the trust in government measures and compliance with preventive guidelines in the countries represented in the first data extraction on Monday, 6 April 2020 (see inclusion criteria in §6). We expected the COVIDiSTRESS global survey to show significant differences between countries.

— H1a: Perceived stress will differ between countries.
— H1b: Concern over the coronavirus will differ between countries.
— H1c: Trust in government efforts to slow the spread of coronavirus will differ between countries.
— H1d: Compliance with behavioural guidelines to slow the spread of coronavirus will differ between countries.

Exploratory cross-country comparisons and descriptive statistics are provided in the Results section for H1 and electronic supplementary material.

Second, the present analyses focused on how both stress and concerns about the COVID-19 pandemic (e.g. isolation, changes in public life and worry over the virus itself) are related to compliance with preventive guidelines and trust in government efforts intended to reduce the spread of coronavirus. Compliance with behavioural guidelines:

— H2a: Across countries, perceived stress will be negatively correlated with compliance to behavioural guidelines to slow the spread of coronavirus.
— H2b: Across countries, concern over the coronavirus will be positively correlated with compliance to behavioural guidelines to slow the spread of coronavirus.

Trust in government efforts:

— H3a: Across countries, trust in government efforts to slow the spread of coronavirus will be negatively correlated with perceived stress.
— H3b: Across countries, trust in government efforts to slow the spread of coronavirus will be negatively correlated with concern over the coronavirus.

Third, we analysed the sources of stress during the global COVID-19 crisis. We assumed that concern over the coronavirus, arising from the direct health threat of COVID-19 and uncontrollable nature of the pandemic, will be a direct stressor.

— H4: Across countries, concern over the coronavirus will be a predictor of perceived stress.

**Table 1.** A list of variables of interest in the present study.

| variable name | description | measurement | remarks |
|---|---|---|---|
| independent and dependent variables | | | |
| [Scale_PSS10] | perceived stress for the past week | Perceived Stress scale (PSS-10). 10 items. 5-Point Likert scale (1 = never, 5 = very often). Validated language versions where available. Back-translated where necessary. Sum score was computed. Continuous variable. | Cohen *et al.* [57] |
| [OECD_institutions] | trust in country's efforts to handle the coronavirus situation | On a scale of 0–10 (0 = not at all, 10 = completely), how much do you personally trust each of the institutions: 'government's effort to handle coronavirus' (added). Note that five other institutions were rated which are not relevant to the present research. Continuous variable. | OECD guidelines on measuring institutional trust [60] |
| [Corona_concerns] | concern over coronavirus | Five self-reported items to capture concerns about coronavirus consequences on a 6-point Likert scale (1 = strongly disagree, 6 = strongly agree): Concern about consequences of the coronavirus (1) … for yourself, (2) … for your family, (3) … for your close friends, (4) … for your country, (5) … for other countries across the globe. Mean score was computed. Continuous variable. | — |
| [Compliance] | compliance with local prevention guidelines | Item 'I have done everything I could possibly do as an individual, to reduce the spread of coronavirus' captures compliance with local prevention guidelines on a 6-point Likert scale (1 = strongly disagree, 6 = strongly agree). Continuous variable. | — |
| control variables (covariates) | | | |
| [age] | participants' age | Continuous variable. | — |
| [gender] | participants' gender | 0 = male, 1 = female, 2 = other/would rather not say). Categorical variable. | — |

**Table 1.** (*Continued.*)

| variable name | description | measurement | remarks |
|---|---|---|---|
| [education] | participants' education | 1 = PhD / doctorate, 2 = college degree, 3 = some college or equivalent, 4 = up to 12 years of school, 5 = up to 9 years of school, 6 = up to 6 years of school, 7 = none).<br>Categorical variable. | — |
| [country] | country of residence | List of all countries where the survey was disseminated/spread. Categorical variable (factor). | |
| [SPS] | available social provisions in critical/distressing situations | Social Provisions scale short form SPS-10. 10 items. 6-point Likert scale (1 = strongly disagree, 6 = strongly agree). Validated language versions where available. Back-translated where necessary. Mean scores will be computed. Continuous variable. | Steigen & Bergh [47] |
| [population_size]).<br>[GDP]).<br>[edu_attainment]).<br>[unemployment]).<br>[gini_coefficient] | country data | — Population size.<br>— GDP per capita.<br>— Education attainment.<br>— Unemployment rate.<br>— Income inequality (GINI coefficients).<br>Continuous variables. | Country demographic variables from OECD). [61]).<br>We added country-level variables (population size, GDP per capita, education attainment, unemployment rate, and GINI coefficient) as control variables to predict the DVs. |

Fourth, based on the stress-buffering hypothesis, we analysed whether social support can alleviate the negative effect of perceived stress on the extent to which people comply with preventive measures. While stressful situations may be unavoidable during a pandemic, the availability of support systems may promote less threatening interpretations of the situation, and thus, lead to a more positive appraisal of efforts to combat the coronavirus.

— H5: Across countries, availability of social provisions will negatively moderate the effect of perceived stress over coronavirus on compliance with behavioural guidelines to slow the spread of coronavirus.

Lastly, individual-level control variables/covariates were included in our multilevel analyses that examine hypotheses 2–5. These variables/covariates are respondent gender, age and education level. According to Coffé & Bolzendahl [62], women were less interested and involved in politics than men. Because trust in government can be easily affected by salient political issues (such as government's efficacy to handle the pandemic), we expected that respondent gender may predict trust in government efforts. In addition, we expect that respondents of different ages may react differently in terms of their experienced stress and worry [63]. As for respondent education, Dalton [64] indicated that education level was related to a decrease in trust in government. Based on the above reasons, respondent age, gender and education level served as individual-level covariates/control variables when testing our hypotheses 2–5. In addition, we added several macro-level (country-level) control variables that may be associated with the outcomes. We were particularly interested in each country's socio-economic growth (e.g. GDP, Gini coefficient and educational attainment). Country-level socio-economic factors have been reported to be related to psychological well-being indicators (e.g. stress, negative affect), so they have a merit to be included in our models [65]. Moreover, country-level education attainment was also considered given the negative association between the overall educational level and poor psychological well-being at the country level [66].

# 5. Number of observations to be collected and rule of termination

Given that this large-scale COVIDiSTRESS survey was spread by numerous researchers all over the world, we had limited control over the total number of responses per country. Thus, we defined a set of stopping rules that are practically feasible.

Statistical power in multilevel design is rather complex and greatly depends on the nested structure of the data [67]. Therefore, a general rule of thumb (i.e. 30–30 rule; [68]) is unlikely to be applicable to our data structure. Thus, we sought to refer to a more specific guideline to plan sample sizes according to the estimates acquired from pre-existing databases.

Due to a lack of previous research that could be used for reasonable power analyses, we ran a planned multilevel analysis for H5 (specified in §9), as there was a dataset at hand, which we perceived as sufficiently suitable. We used the 2018 European Social Survey (ESS) database to estimate the effect sizes of fixed and random effects as well as intraclass coefficients (ICCs). The ESS database consists of 43 215 respondents from 23 European countries (on average 1878 respondents per country). Although the variables available in the ESS database are not identical to those in the COVIDiSTRESS survey, given their similarity in the underlying constructs, we used the health condition variable (a higher value indicating poorer health condition) and satisfaction about the government in the ESS data as proxies for stress, compliance and trust in government efforts for our original variables of interest, respectively. The ICC from the observed ESS data was 0.26, indicating that 26% of variability comes from between-country differences. Since random effects are often distorted while the variables are standardized, the random effects were estimated with grand-mean-centred continuous variables. The same multilevel analyses were run again with standardized variables only in order to obtain the standardized estimates of fixed effects. Standardized estimates of fixed effects of the individual-level predictor (individual's health condition: $\beta = 0.09$; social provisions: $\beta = 0.21$) and its country-level predictor (mean health condition in each country: $\beta s = -0.35$) were deemed to have small and medium effect sizes, respectively. Their cross-level interaction effects had only small effect sizes ($\beta s = 0.03$–$0.10$).

Subsequently, we referred to the guidelines from Arend & Schäfer [67], which provided a fast and frugal power estimation for each combination of effect size, the value of ICC, as well as the size of random slope variance. Based on Arend & Schäfer's Table 8 of power simulation results for required sample size and group size (in our case, the number of countries) to detect such effects with 80% statistical power (p. 17), we planned to recruit at least *30 participants per country* so as to detect both the effects of individual- and country-level predictors. However, since our data collection process was

still ongoing at the time of stage one submission, the potential number of group size and sample size were greater than we had initially reported in this section.

# 6. Inclusion criteria

All participating researchers received the same task of starting to distribute the survey in their language on 30 March, 2020 (exception: Denmark started on 26 March, 2020, see §7). All possible channels were allowed, e.g. social media, panels, e-mails to friends or organizations, use of media contacts such as the newspaper or TV, websites of organizations such as universities or NGOs involved in health communication. Moreover, we also asked all participants to help spread the survey after they finished their own responses.

Data was extracted each Monday and made available at the Open Science Framework. In order to be considered for the present analysis, however, a country needed at least 30 respondents at the time of acceptance of the RR stage 1 (i.e. the most recent data extracted at time of acceptance). In order to be considered as a valid participant for the present analyses, a respondent must have reported their country of residence and submitted valid responses to the variables treated in each analysis (see also §7).

# 7. Exclusion criteria

We defined exclusion criteria on the country and on the respondent level.

## 7.1. Country level

Active dissemination of the survey and calls for participation were carried out via online and traditional media platforms in Afghanistan, Argentina, Australia, Austria, Bangladesh, Belgium, Bosnia, Brazil, Bulgaria, Canada, China, Colombia, Croatia, Republic of Cyprus, Czech Republic, Denmark, Finland, France, Germany, Greece, India, Indonesia, Israel, Italy, Japan, South Korea, Lithuania, Malaysia, Mexico, The Netherlands, Pakistan, The Philippines, Poland, Portugal, Russia, Serbia, Singapore, South Africa, Spain, Switzerland, Taiwan, Turkey, United Kingdom, United States of America and Vietnam. Countries that failed to generate 30 respondents by the time of extraction were not included in the final dataset. Sections 5 and 9 provide a detailed justification for the exclusion of countries that did not reach 30 respondents. In short, 30 respondents per country was the minimal number of respondents required to ensure sufficient statistical power.

## 7.2. Individual level

On the individual level, the length of the survey can lead some participants to skip questions, but also give repetitive or unrepresentative answers (e.g. [69]), leading to misclassification of participants and responses that do not reflect real experience [70]. We employed the following exclusion measures to protect against these threats.

1. First, the predicted duration of the survey in Qualtrics was 22 min if all questions were answered and free text boxes were filled out. On that basis, we excluded all responses who completed the whole survey in less than 2 min and 12 s, equivalent to one-tenth of the estimated time.
2. Second, we used Mahalanobis distance to detect multivariate outliers due to random or carelessly invalid responses [71,72]. Participants with a $p < 0.001$ in the chi-square test were excluded.

# 8. Quality checks

Regarding quality checks, we checked for floor/ceiling effects within the following scales/item batteries: Perceived Stress Scale (PSS-10), Corona concerns, Compliance, SPS-10. We evaluated floor/ceiling effects on the base of the percentage of the respondents with the minimum/maximum scores. Therefore, we provided the percentage (%) and $n$ for respondents with the minimum/maximum scores. Floor/ceiling effects will be considered as present if minimum/maximum scores occur in 15% or more of the respondents, following the previous recommended threshold (see [73]).

Furthermore, we tested the measurement invariance of the included scales: the PSS-10 (see [57,74]), Social Provisions Scale (SPS, short form of PSP-10; [47]) and coronavirus concerns (self-reported measure, constructed for the purpose of this study). The measurement invariance test was required

since we wanted to compare scores measured by the aforementioned scales across different countries. Further details regarding the measurement invariance test procedures are described in the next section[1]. We implemented the examination of the equivalence of measures at the suggestion of a reviewer.

Given that the number of countries to be analysed directly influences the power of the planned multilevel modelling, we conducted the multilevel modelling analyses as follows:

1. If our planned analyses can be conducted without any issue (e.g. failed convergence) with the dataset that passed the measurement invariance test, then we would use composite scores and report the findings.
2. If 1 failed, then we would apply the alignment method and use aligned factor mean scores instead of composite scores for the planned analyses. If 1 failed and findings from 2 were reported, we would discuss limitations and caveats regarding the interpretation of the findings.

For quality check reasons we provide histograms of the main variables in the electronic supplementary material. Specifically, we depict histograms for *perceived stress* (electronic supplementary material, figure S1), *concern over the coronavirus* (electronic supplementary material, figure S2), *trust in government efforts* (electronic supplementary material, figure S3) and *compliance with behavioural guidelines* (electronic supplementary material, figure S4).

# 9. Analysis plan

## 9.1. Measurement invariance testing

We investigated the cross-cultural equivalence of the PSS-10 (see [57,74]), the SPS, short form of PSP-10 [47] and the coronavirus concern scale (self-reported measure, constructed for the purpose of this study) prior to any main analyses. Using a multi-group factor analysis, we compared the models assuming the one-factor structure for SPS and coronavirus concerns, and the two-factor structure for PSS-10 (positive and negative, with the latter consisting of reversed items; [75,76]) across all countries (configural invariance), with a model with factor loadings and latent correlations constrained to be equal (metric invariance), and items' intercepts to be the same in all groups (scalar invariance).

When evaluating the model fit, we relied on the usually applied criteria [77], in which a comparative fit index (CFI) and Tucker–Lewis index (TLI) above 0.90 would indicate adequate fit, whereas a standardized root mean square residual (SRMR) below 0.06, and a root mean square error of approximation (RMSEA) below 0.08 indicate no misfit. When evaluating the measurement equivalence, we compared the configural invariance model with the metric invariance model, and then the metric invariance model with the scalar invariance model [78]. As these models are characterized by a growing complexity (each subsequent model is nested within the previous one), while assessing models' superiority, we relied on cut-off criteria recommended for testing measurement invariance: a change of CFI less than 0.01 ($\Delta$CFI < 0.01), a change of RMSEA of less than 0.015 ($\Delta$RMSEA < 0.015) and a change of SRMR less than 0.01 ($\Delta$SRMR < 0.01) indicating that two compared models do not differ in terms of model fit [79,80]. However, in the case that the equivalence of invariance was not achieved, based on recommendations indicating that partial invariance may allow for reasonable comparisons (see [81]), we also estimated models with partial invariance. If the equivalence of partial invariance was achieved, we proceeded with the cross-countries comparisons.

Nevertheless, considering the fact that even the equivalence of partial invariance is sometimes difficult to achieve when comparing numerous countries [82–84], we used the alignment method implemented in *Mplus* as an alternative [85]. After applying the alignment method, we used the adjusted factor mean scores that were produced by the alignment process for our planned analyses.

## 9.2. General data analysis approach

— The main analysis used the lavaan, tidyverse and brms R packages.
— Descriptive statistics were computed for all variables/scales (i.e. $M$, s.d., $\alpha$, $r$s).
— Multilevel models (MLMs) were run using SAS PROC MIXED with restricted maximum-likelihood (REML) and Kenward–Roger denominator degrees of freedom.

---

[1]Registered report note: tests of measurement invariance were not part of the stage one report.

**Table 2.** Bayesian cut-off criteria with the interpretation. (Note: descriptions are italicized when either a null or alternative hypothesis is supported by evidence.)

| Bayes Factor (BFH$_{10}$) | Interpretation |
|---|---|
| $\geq 10$ | *The effect is strongly supported by evidence.* |
| $3 \leq BF < 10$ | *The effect is positively supported by evidence but not strongly.* |
| $1/3 < BF < 3$ | The current evidence is insufficient to make any decisive decision although the non-zero effect is likely to exist. |
| $1/3 < BF < 3$ | The current evidence is insufficient to make any decisive decision although the null hypothesis (effect = 0) is likely to be the case. |
| $1/10 < BF \leq 1/3$ | *The null hypothesis (effect = 0) is supported by evidence but not strongly.* |
| $\leq 1/10$ | *The null hypothesis (effect = 0) is strongly supported by evidence.* |

— To test H1a to H1d, we compared different models with MLM with SAS. We tested whether the model with a random intercept, the country, is significantly better than the null model to examine whether the dependent variables are significantly different across different countries. The inclusion of random intercept was assessed by both ICC and goodness of model fit: ICC was calculated to estimate the size of variability that resides at the country level, and a likelihood ratio (LR) test was performed to compare model fit between Model 0 and Model 1:

  o Model 0: Null model (no predictors or random effects added).
  o Model 1: Random intercept-only model (Model 0 + country as a random intercept).
  o Model 2: Country-effect model (the fixed effect of country).

— In Model 2, if the fixed effect of the country reached significance, we conducted a series of *post hoc* comparisons to examine the between-country difference. Then, we performed Bayesian MLM with Model 2 with brms to confirm whether the effect was significantly different from zero (table 2; [86]). Once we found that the effect of the country is non-zero, we performed a *post hoc* test with Scheffe's method to compare the dependent variables between countries.

— Similarly, while testing H2 to H5, to identify the best model, we compared the following five models for each hypothesis. We performed a LR test to examine whether the addition of random effects significantly improves the model (Model 1 versus Model 2). For other comparisons, i.e. Models 2 versus 3, Models 3 versus 4, Models 4 versus 5, we employed other methods, such as the pseudo-$R^2$ comparison and/or ominous F test because the SAS macro allows the use of the LR test only for the comparison of models with versus without random effects. Bayesian MLM with brms was performed with the best model.

  o Model 1: Null model (no predictors or random effects added).
  o Model 2: Random intercept-only model (Model 1 + country as a random intercept).
  o Model 3: Model with fixed effects of individual-level predictors (Model 2 + fixed effects).
  o Model 4: Model with country-level control variables (Model 3 + control variables).
  o Model 5: Full model (Model 4 + random slopes for predictors).

— For frequentist analyses, 95% confidence intervals and the conventional 5% significance level ($p < 0.05$) were used for H0 significance testing. Categorical variables (such as countries) were dummy coded; continuous variables were centred on grand means. Variables for individual-level predictors (country-mean variables; see §3) were computed and centred on grand means in order to separate variance of country part from the variance of the individual part (see [56]). We used the variance inflation factor (VIF) as a diagnostic for multicollinearity. If VIF > 3, we did the following: If the high VIF concerned the main independent variable (IV), we excluded the IV's collinear variable(s). If the high VIF concerned covariates, we performed a PCA to reduce the number of covariates to a set of uncorrelated covariates.

— For multilevel modelling, the significance of fixed effects was examined using Wald tests with degrees of freedom adjusted with Kenward–Roger method; random effects were tested via likelihood ratio tests (−2ΔLL with degrees of freedom equal to the number of new random effects variance and covariance). The effect size for fixed effects was examined via pseudo-$R^2$ for the proportion reduction in each variance component, along with the change in total $R^2$, i.e. the

squared correlation between the actual outcome and predicted outcome by the predictive models [87]. Moreover, to increase the power of multilevel modelling, we performed the same analyses with two different datasets: one before and one after the respondent-level exclusion screening (see §7 for criteria). We primarily presented the results of MLM with the whole dataset and those with the screened dataset in the appendix in the Open Science Framework repository for reference. We performed MLM with the whole dataset before the screening to maximize the statistical power by retaining as many samples as possible.

— For Bayesian analyses, Bayes factor $\geq 10$ (BF10 $\geq 10$) was employed, which indicates the presence of strong evidence supporting H1 against H0 for Bayesian testing in general [86,88]. In addition to this main Bayes factor criterion, we used Bayes factor $\geq 3$ (BF10 $\geq 3$) to indicate the presence of positive but not strong evidence supporting our hypothesis auxiliary.

— We interpreted the outcomes based on both $p$-values and Bayes factors. Further details regarding how to make decisions are presented in table 2. Although $p$-values and Bayes factors indicate the same direction of the result (e.g. If Bayes factor $\geq 3$, then $p$ should be less than 0.05; see [89,90]), if there is an error in either frequentist or Bayesian MLM, they might provide contradictory results. Thus, for robustness reasons, we used both indicators to examine whether the planned MLM analysis is completed without any methodological errors.

In table 3, we specify the analyses for all hypotheses.

## 9.3 Model specification for multilevel model

Below, we illustrated the model specification for a multilevel model designed to test hypotheses 2A and 5: Compliance with behavioural guidelines was a function of fixed and random intercepts (B00 and R0), perceived stress (B40 and R1) and availability of social provisions (B50), when controlling for both individual-level control variables (B10–B30) and country-level control variables (B01–B05). An interaction term was created to probe into the moderating role of social provisions in the relationship between individual-level stress and compliance (B60).

**Level 1 (Individual):**
Y = P0 + P1*(Gender) + P2*(Age) + P3*(Education) + P4*(Stress) + P5*(Social Provisions) + P6*(Stress × Social Provisions) + E
**Level 2 (Country):**
P0 = B00 + B01*(Population  Size) + B02*(GDP) + B03*(Education  Attainment) + B04*(Unemployment) + B05*(Gini Coefficient) + R0
P1 = B10,  P2 = B20,  P3 = B30
P4 = B40 + R1
P5 = B50
P6 = B60
**Compositional Model for Multilevel Analysis:**
Compliance = B00 + B10*(Gender) + B20*(Age) + B30*(Education) + B01*(Population Size) + B02*(GDP) + B03*(Educational  Attainment) + B04*(Unemployment) + B05*(Gini  Coefficient) + (B40 + R1)*(Stress) + B50*(Social Provisions) + B60*(Stress × Social Provisions) + R0 + E

## 9.4. Effect size analysis

In addition to the originally planned significance testing, we also computed effect sizes of predictor(s) of interest in each model. In the cases of H1a–H1d, we examined intraclass correlation coefficients (ICC). ICCs explain to what extent the variances in a dependent variable of interest (i.e. perceived stress, concern over the coronavirus, trust in government efforts and compliance with behavioural guidelines) are attributable to differences across countries. For H2a–H5, we employed *lme.dscore* implemented in the R package, *EMAtools*, to estimate Cohen's *D* of predictor(s) of interest in each model. The R source code used for this process is available via the OSF (https://osf.io/zscdh/; data and other related files are available in the same folder).

**Table 3.** An overview of the study's hypotheses and analyses plan.

| question/hypothesis | sampling plan (e.g. power analysis) | analysis | interpretation given different outcomes | test result |
|---|---|---|---|---|
| H1a<br>Perceived stress will differ between countries | Given that the analysis that required the greatest sample size was the planned MLM (see H5) due to its complexity, we followed the sample size estimation for the MLM (see S5 for further details). In addition, we examined the resultant Bayes factor ≥ 10 to see whether the corrected evidence is sufficient for hypothesis testing. | We performed MLM to examine the international differences.<br>DV:<br>Perceived stress ([Scale_PSS10])<br>IV:<br>Country ([country]) (as random and fixed effects)<br>We compared three models as proposed above. Once we found that the fixed effect of the country was statistically significant, we then performed *post hoc* analysis with Scheffe's method for international comparison.<br>In addition, we performed the same Bayesian MLM with the brms R package. We examined whether Bayes factor of the estimated B for the country is 10 or greater (BF10 ≥ 10; table 2 for further details).<br>Because we tested whether B is non-zero (H0), we used non-informative priors centred around zero for brms and Bayes factor calculation. Following Rouder and Morey's [91] suggestions on Bayesian multivariate regression analysis, we used a Cauchy prior (*d* = 0.00, scale = 1.00) for coefficients of interest. For other indicators, we used default priors set by brms. | We used both the resultant *p*-value and Bayes factor to examine whether the effect of the country (H1a) is non-zero. We used both the *p*-value and Bayes factor of the B for the country for completeness of our analysis. To interpret the resultant indicators, we employed the criteria provided in table 2. | Supported. |
| H1b<br>Concern over the coronavirus will differ between countries | See descriptions provided in H1a and S5. | We performed MLM to examine the international differences.<br>DV:<br>Concern over the coronavirus ([Corona_concerns])<br>IV:<br>Country ([country]) (as random and fixed effects)<br>We compared three models as proposed above. Once we found that the fixed effect of the country was statistically significant, we then performed *post hoc* analysis with Scheffe's method for international comparison.<br>In addition, we performed the same Bayesian MLM with the brms R package (see descriptions provided in H1a). | See descriptions provided in H1a. We applied the same criteria to examine whether the effect of the country is non-zero (H1b). | Supported. |

(*Continued.*)

**Table 3.** (*Continued.*)

| question/hypothesis | sampling plan (e.g. power analysis) | analysis | interpretation given different outcomes | test result |
|---|---|---|---|---|
| H1c Trust in government efforts to slow the spread of coronavirus will differ between countries | See descriptions provided in H1a and §5. | We performed mixed MLM to examine the international differences. DV: Trust in country's government efforts ([OECD_institutions]) IV: Country ([country]) (as random and fixed effects) We compared three models as proposed above. Once we found that the fixed effect of the country was statistically significant, we then performed *post hoc* analysis with Scheffe's method for international comparison. In addition, we performed the same Bayesian MLM with the brms R package (see descriptions provided in H1a). | See descriptions provided in H1a. We applied the same criteria to examine whether the effect of the country is non-zero (H1c). | Supported. |
| H1d Compliance with behavioural guidelines to slow the spread of coronavirus will differ between countries | See descriptions provided in H1a and §5. | We performed mixed MLM to examine the international differences. DV: Compliance with behavioural guidelines to slow the spread of coronavirus ([Compliance]) IV: Country ([country]) (as random and fixed effects) We compared three models as proposed above. Once we found that the fixed effect of the country was statistically significant, we then performed *post hoc* analysis with Scheffe's method for international comparison. In addition, we performed the same Bayesian MLM with the brms R package (see descriptions provided in H1a). | See descriptions provided in H1a. We applied the same criteria to examine whether the effect of the country is non-zero (H1d). | Supported. |

(*Continued.*)

**Table 3.** (*Continued.*)

| question/hypothesis | sampling plan (e.g. power analysis) | analysis | interpretation given different outcomes | test result |
|---|---|---|---|---|
| **H2a**<br><br>Across countries, perceived stress will be negatively correlated with compliance with behavioural guidelines to slow the spread of coronavirus (see the model specified in 'Model specification for multilevel model' section) | See descriptions provided in H1a and §5. | We examined a MLM without interaction effects to test the relationship between stress and compliance.<br><br>DV:<br>Compliance with behavioural guidelines to slow the spread of coronavirus [Compliance]<br>Level 1 IVs:<br>Perceived stress ([Scale_PSS10], individual demographic variables ([age], [gender], [edu])<br>Level 2 IVs:<br>Country-level indicators and country demographic variables ([population_size], [GDP per capita], [edu_attainment], [unemployment], [gini_coefficient])<br>We included [country] as a random intercept in the model.<br>In addition, we performed the same Bayesian MLM with the brms R package. We examined whether Bayes factor of the estimated B for perceived stress at Level 1 is 10 or greater (BF10 $\geq$ 10; table 2 for further details).<br>Because we tested whether B is non-zero (H0), we used non-informative priors centred around zero for brms and Bayes factor calculation. Following Rouder and Morey's [91] suggestions on Bayesian multivariate regression analysis, we used a Cauchy prior ($d = 0.00$, scale $= 1.00$) for Level 1 coefficients. For other indicators, we used default priors set by brms. | We used both the resultant *p*-value and Bayes factor to examine whether the effect of perceived stress (H2a) is non-zero. We used both the *p*-value and Bayes factor of the B for the Level 1 stress for completeness of our analysis. To interpret the resultant indicators, we employed the criteria provided in table 2. | Supported. |

**Table 3.** (*Continued.*)

| question/hypothesis | sampling plan (e.g. power analysis) | analysis | interpretation given different outcomes | test result |
|---|---|---|---|---|
| H2b<br><br>Across countries, concern over the coronavirus will be positively correlated with compliance with behavioural guidelines to slow the spread of coronavirus | See descriptions provided in H1a and §5. | We examined a MLM without interaction effects to test the relationship between concern over coronavirus and compliance.<br><br>DV:<br>Compliance with behavioural guidelines to slow the spread of coronavirus [Compliance]<br>Level 1 IVs:<br>Concern over coronavirus ([Corona_concerns]), individual demographic variables ([age], [gender], [edu])<br>Level 2 IVs:<br>Country-level indicators and country demographic variables ([population_size], [GDP], [edu_attainment], [unemployment], [gini_coefficient])<br>We included [country] as a random intercept in the model.<br>In addition, we performed the same Bayesian MLM with the brms R package (see descriptions provided in H2a). | See descriptions provided in H2a. We applied the same criteria to examine whether the effect of concern over the coronavirus is non-zero (H2b). | Supported. |
| H3a<br><br>Across countries, trust in government efforts to slow the spread of coronavirus will be negatively correlated with perceived stress | See descriptions provided in H1a and §5. | We examined a MLM without interaction effects to test the relationship between trust in government efforts to slow the spread of coronavirus and perceived stress.<br><br>DV:<br>Perceived stress ([Scale_PSS10])<br>Level 1 IVs:<br>Trust in government efforts to slow the spread of coronavirus ([OECD_institutions]) individual demographic variables ([age], [gender], [edu])<br>Level 2 IVs:<br>Country-level indicators and country demographic variables ([population_size], [GDP], [edu_attainment], [unemployment], [gini_coefficient])<br>We included [country] as a random intercept in the model.<br>In addition, we performed the same Bayesian MLM with the brms R package (see descriptions provided in H2a). | See descriptions provided in H2a. We applied the same criteria to examine whether the effect of trust is non-zero (H3a). | Supported. |

(*Continued.*)

**Table 3.** (*Continued.*)

| question/hypothesis | sampling plan (e.g. power analysis) | analysis | interpretation given different outcomes | test result |
|---|---|---|---|---|
| H3b<br><br>Across countries, trust in government efforts to slow the spread of coronavirus will be negatively correlated with concern over coronavirus | See descriptions provided in H1a and §5. | We examined a MLM without interaction effects to test the relationship between concern over coronavirus and trust in government efforts to slow the spread of coronavirus.<br><br>DV:<br><br>Concern over coronavirus ([Corona_concerns])<br><br>Level 1 IVs:<br><br>Trust in government efforts to slow the spread of coronavirus ([OECD_institutions]), individual demographic variables ([age], [gender], [edu])<br><br>Level 2 IVs:<br><br>Country-level indicators and country demographic variables ([population_size], [GDP], [edu_attainment], [unemployment], [gini_coefficient])<br><br>We included [country] as a random intercept in the model.<br><br>In addition, we performed the same Bayesian MLM with the brms R package (see descriptions provided in H2a). | See descriptions provided in H2a. We applied the same criteria to examine whether the effect of trust is non-zero (H3b). | Not supported. |
| H4<br><br>Across countries, concern over the coronavirus will be a predictor of perceived stress | See descriptions provided in H1a and §5. | We examined a MLM without interaction effects to test the relationship between concern over coronavirus and stress.<br><br>DV:<br><br>Perceived stress ([Scale_PSS10])<br><br>Level 1 IVs:<br><br>Concern over coronavirus ([Corona_concerns]), individual demographic variables ([age], [gender], [edu])<br><br>Level 2 IVs:<br><br>Country-level indicators and country demographic variables ([population_size], [GDP], [edu_attainment], [unemployment], [gini_coefficient])<br><br>We included [country] as a random intercept in the model.<br><br>In addition, we performed the same Bayesian MLM with the brms R package (see descriptions provided in H2a). | See descriptions provided in H2a. We applied the same criteria to examine whether the effect of concern over coronavirus is non-zero (H4). | Supported. |

*(Continued.)*

**Table 3.** (Continued.)

| question/hypothesis | sampling plan (e.g. power analysis) | analysis | interpretation given different outcomes | test result |
|---|---|---|---|---|
| H5<br><br>Across countries, availability of social provisions will negatively moderate the effect of perceived stress over coronavirus on compliance with behavioural guidelines to slow the spread of coronavirus (see the model specified in 'Model specification for multilevel model' section) | See descriptions provided in H1a and §5. | We examined a MLM with the intended moderator.<br><br>DV:<br>Compliance with behavioural guidelines to slow the spread of coronavirus [Compliance]<br><br>Level 1 IVs:<br>Perceived stress ([Scale_PSS10]), social provisions ([SPS]), perceived stress × social provisions, individual demographic variables ([age], [gender], [education])<br><br>Level 2 IVs:<br>Country-level indicators, aggregated variables (country-level stress), and country demographic variables ([population_size], [GDP], [edu_attainment], [unemployment], [gini_coefficient])<br>We included [country] as a random intercept and a random slope of perceived stress in the model.<br><br>The planned multilevel models were run step by step to make sure that each random effect is added by order (i.e. country random intercept, the random slope of perceived stress), so that the model reaches convergence. We kept a more parsimonious model and dropped the random effect in the case of non-convergence.<br>In addition, we performed the same Bayesian MLM with the brms R package (see descriptions provided in H2a). We added the aforementioned interaction effects in the brms model. | To test the moderation effect, we examined both the p-value and Bayes factor of the B for the moderator of interest. Similar to the previous tests, we employed the p-value and the criteria provided in table 2 to interpret Bayes factor while examining whether the effect of the moderator is non-zero. | Not supported (direction reversed). |

# 10. Data processing

Data collection started on 30 March 2020 in all participating countries, with a pre-launch in Denmark on 26 March 2020 to test if the survey worked in practice and get a first impression on the response rate. Prior to submitting the RR stage 1, we had not accessed or analysed any data, except following response rates to give all COVIDiSTRESS participant researchers feedback on the recruiting progress.

This RR was submitted under the COVID-19 rapid response call for papers in Spring 2020. Upon acceptance of the stage 1 protocol, we analysed the newest weekly data extraction that fit minimum criteria (see above). The full dataset collected is available to any other researcher for further exploration or hypotheses testing via the Open Science Framework.

The version of the COVIDiSTRESS dataset analysed here was extracted on 30 May 2020. A total of 173 429 responses were recorded and cleaned using the R-code recommended by the COVIDiSTRESS team (available at https://osf.io/z39us/files/). The cleaned dataset consisted of 125 306 responses.

Following our pre-registered plan of analysis, we first checked whether the equivalence of invariance of our measures across countries could be reached. For invariance models to converge, we initially excluded 2538 participants (2.03%) (i) from countries with less than 100 participants and (ii) who did not indicate the country of residence. Next, 1223 participants (0.99%) were excluded who completed the survey in less than 10% of the Qualtrics estimated survey duration (i.e. less than 2 min and 12 s). When the actual duration was examined, the mean duration was 2118.38 s. The 1223 excluded participants completed the survey in the top 0.99% fastest speed. Further, we used Mahalanobis distance to detect multivariate outliers due to random or carelessly invalid responses [71,72]. A total of 5189 (4.27%) participants with a $p < 0.001$ in the chi-square test were excluded. After these exclusions, the dataset consisted of 116 356 participants (see electronic supplementary material, table S1).

In the second step, following our pre-registered quality check plan, we investigated floor and ceiling effects for our independent variables, testing how many participants' self-reported compliance with preventive guidelines as well as composite scores of perceived stress (PSS-10), available social provision (SPS) and concern over coronavirus reached the minimum or maximum value. If more than 15% of respondents reached either the maximum or minimum value for a measure, the ceiling/floor effect was considered present. For SPS, a ceiling effect was detected for two countries: Brazil ($N = 81$, 16.0%) and Panama ($N = 78$, 16.1%). For concern over coronavirus, ceiling effects were found for 11 countries: Mexico ($N = 1505$, 19.2%), United States ($N = 322$, 16.0%), Panama ($N = 103$, 16.1%), Philippines ($N = 89$, 17.9%), Brazil ($N = 112$, 18.3%), Colombia ($N = 28$, 17.2%), Malaysia ($N = 137$, 29.7%), South Africa ($N = 24$, 21.4%), Greece ($N = 105$, 18.8%), Pakistan ($N = 40$, 15.2%) and Portugal ($N = 153$, 17.0%). In the third step, we assessed measurement invariance. As detailed in the RR, we expected that the COVIDiSTRESS global survey would show significant differences between countries, and five hypotheses were proposed. In order to test our hypotheses, we needed to examine the extent to which participants from all countries completed the focal measures similarly. Using a classical approach, we reached partial invariance in the case of PSS-10 and SPS-10, but not for concern over coronavirus. Thus, we proceeded with the alignment method (see OSF for the results of measurement invariance tests and alignments; links: https://osf.io/qp96b/ and https://osf.io/hu683/). The three measures examined were the corona concern scale, the PSS and the SPS (table 1).

We followed four main analytical steps in Mplus for measurement testing. First, we conducted an exploratory factor analysis using maximum-likelihood estimation and geomin rotation for each scale to confirm their dimensionality. One-factor structures were confirmed for all three scales. Second, we examined the extent to which the one-factor model was well-fitting across the entire sample using a pooled dataset. This was achieved by adjusting variables so that cases from each country have equal weighing on the covariance matrix. Third, we conducted a multi-group confirmatory factor analysis testing a configural invariance model to examine the measurement invariance of the one-factor model across countries. Fourth, we tested measurement invariance more properly with the alignment optimization method (see detailed description below). Finally, we conducted Monte Carlo simulations to provide a validation check of the alignment results, indicating whether the rank order of latent factors across countries is trustworthy.

The latent factor scores for each scale obtained from the alignment analyses were combined to be used in the key hypotheses testing analyses. We examined data from 104 332 participants from 48 countries. Sample size ranged from 109 (New Zealand) to 20 165 (Finland), with a mean of 2173 and median of 921 participants per participating country. Based on the results reported in detail below, we deemed that measurement invariance of the three measures was achieved. In brief, the results provided strong

evidence for the one-factor structure of all three measures, that this unifactorial factor structure was confirmed across all 48 countries, that participants from all countries responded to the items relatively similarly, and the Monte Carlo simulations confirmed the trustworthiness of the latent factor scores obtained.

The final dataset consisted of 116 356 participants from 48 countries with the mean age of 39.4 (s.d. = 14.1), 84 942 (73.00%) of participants identified as female, 29 869 (25.51%) as male, and 1283 (1.10%) provided other responses or declined to respond. For detailed demographic statistics, see electronic supplementary material, (electronic supplementary material, table S1).

For the multilevel models (H1a to H5), we dummy coded individual-level categorical variables (participant gender and education level) and centred the continuous variables (predictors and control variables) on grand means (see electronic supplementary material, table S1 to S4 with descriptive statistics). Because the ICCs showed that a considerable amount of variability of outcome variables was accounted for by between-country difference (see the Results section), country-mean variables for individual-level predictors were created by aggregating individual scores at the country level and centred on the grand means in order to separate variance of country part from the variance of the individual part (see [56]).

Country-level covariates (population size, GDP per capita, Gini coefficient and educational attainment) were acquired from the World Bank World Development Indicators database [92]. We used PPP-adjusted GDP per capita, which represents each country's level of economic activities controlling for purchasing power (i.e. the differences in price levels between countries; Eurostat). Gini coefficient measures the extent to which the distribution of income among individuals in a given country deviates from a perfectly equal distribution (ranging from 0 to 100—0 represents absolute equality, whereas 100 represents absolute inequality; [92]). Educational attainment referred to the percentage of 25 years old or above in the population who completed at least upper secondary education.

# 11. Results

## 11.1. Mapping cross-country differences

Hypotheses 1a to 1d tested between-countries differences in *perceived stress*, *concern* over the coronavirus, *trust* in government efforts and *compliance* with behavioural guidelines.

The ICCs obtained in MLM for the random intercept-only models for *stress, concern, trust in government efforts* and *compliance with behavioural guidelines* are 12.16%, 13.04%, 26.97% and 5.59%, respectively, suggesting that a considerable proportion of variances—from 5 to 27%—were attributable to differences across countries (see electronic supplementary material, table S5). Examination of random variance revealed that the hypothesized cross-country differences in *perceived stress* (H1a), *concern over the coronavirus* (H1b), *trust in government efforts* (H1c) and *compliance with behavioural guidelines* (H1d) were supported by statistically significant country random intercepts (*stress*: $B = 0.16$, s.e. = 0.03, Wald $Z = 4.81$, $p < 0.001$; *concern*: $B = 0.24$, s.e. = 0.05, Wald $Z = 4.79$, $p < 0.001$; *trust in government efforts*: $B = 2.26$, s.e. = 0.47, Wald $Z = 4.82$, $p < 0.001$; *compliance*: $B = 0.05$, s.e. = 0.01, Wald $Z = 4.74$, $p < 0.001$).

Bayesian MLM also supported the results from the frequentist MLM. In H1a to H1d, the resultant BF10 s were infinite. These results supported the presence of the non-zero country random intercepts for H1a to H1d (see OSF for the full report generated by brms; link: https://osf.io/7g8ej/).

*Post hoc* comparisons with Scheffe correction were conducted to examine differences across countries (see figures 1–4). Overall, the figures revealed interesting patterns for *stress, concern, trust* and *compliance*. For instance, similar patterns were observed for between-country differences in terms of perceived *stress* and perceived *concern*. More specifically, while the level of *stress* and *concern* in Western European countries appeared to be particularly high, several countries displayed characteristically lower levels of both *stress* and *concern*. Among others, Turkey, Portugal, Poland, Philippines, Bulgaria, Brazil and to some extent Bosnia and Herzegovina, experienced comparatively lower levels of stress. Switzerland, Sweden, The Netherlands, Finland, Denmark and to a certain degree Lithuania reported higher levels of stress. On the other hand, Western European countries were somewhat low in *trust* in their governments' efforts to handle the coronavirus. Regarding *compliance* with behavioural guidelines, the between-country differences seemed to be less pronounced, with the exception of Japan, which seemed to be relatively compliant.

cross-country differences in perceived stress
red tiles indicate positive difference (i.e. higher stress) and vice versa

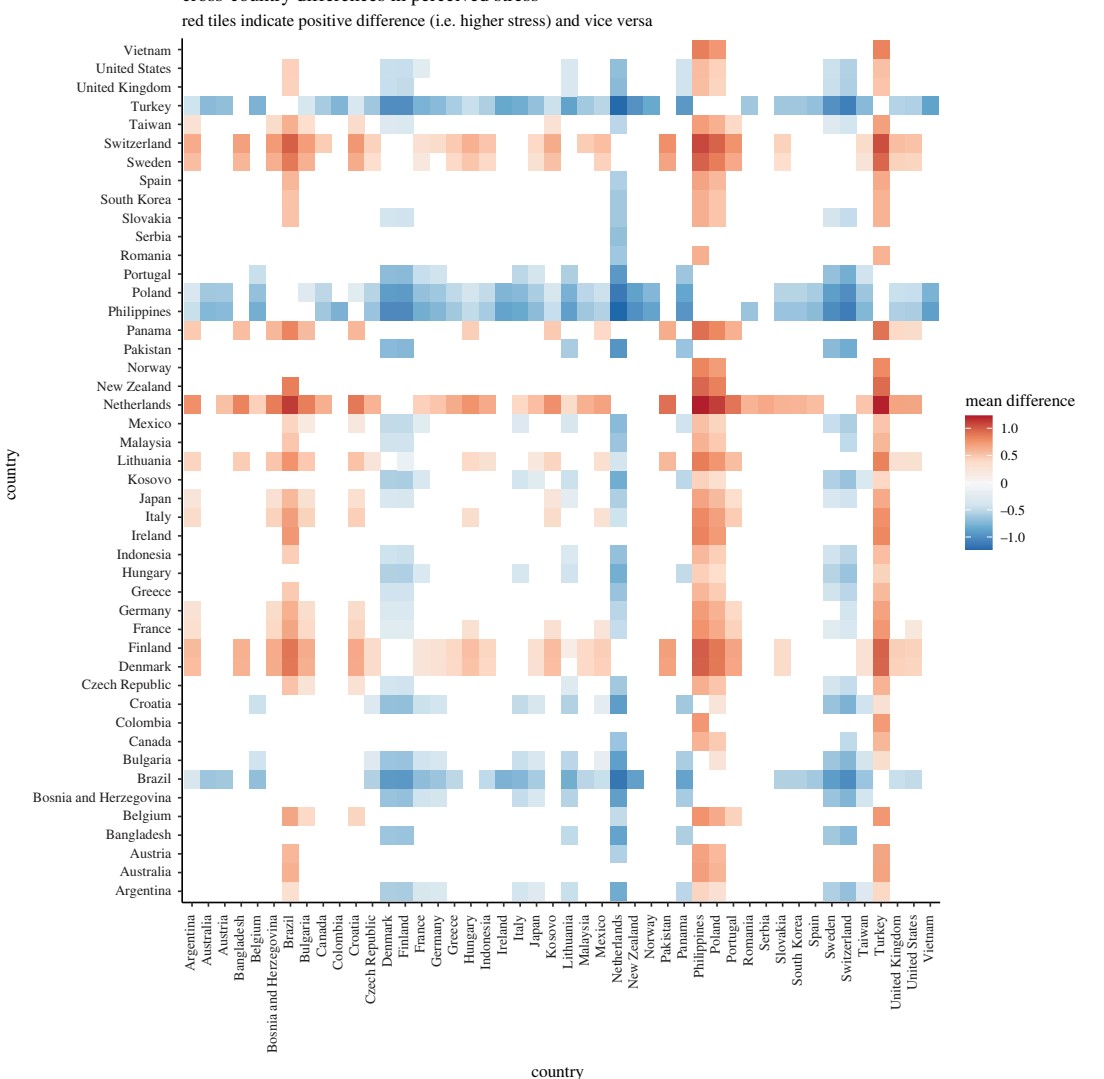

**Figure 1.** Cross-country differences in perceived stress (based on latent scores from alignment measurement invariance for between-country comparison; original scale: 1 = never, 5 = very often).

## 11.2. Perceived stress and compliance with behavioural guidelines

Hypothesis 2a examined the relationship between *stress* and self-reported *compliance* with guidelines for disease-preventive behaviour. The fixed effects of the individual-level predictor (*perceived stress*) and control variables (*age, gender and education*) were first added to the model to predict *compliance* (see electronic supplementary material, table S6). Together all the individual-level variables accounted for 3.40% of the level-1 within-country variance (0.75 as compared with 0.77 in Model 2). As predicted, *perceived stress* was negatively associated with *compliance with behavioural guidelines* to slow the spread of coronavirus ($B = -0.03$, s.e. = 0.01, $t_{100613} = 10.96$, $p < 0.001$). *Age* positively predicted *compliance*, so did *gender*, such that older participants showed greater *compliance* ($B = 0.01$, s.e. = 0.00, $t_{100520} = 38.17$, $p < 0.001$), and that females showed a higher level of *compliance* than males ($B = 0.25$, s.e. = 0.01, $t_{100612} = 37.83$, $p < 0.001$). In Model 4, both country-mean variables and country-level control variables were added to the model. Inclusion of country-level variables accounted for 51.71% of the level-2 between-country variance (0.02 as compared with 0.05 in Model 3). We found that individual-level perceived *stress* was still a statistically significant predictor of *compliance* ($B = -0.03$, s.e. = 0.00, $t_{100567} = 10.98$, $p < 0.001$; figure 5) after controlling for country-level predictors. Thus, H2a was supported by the data.

In Model 5, where a random slope for *stress* was added to the model, the random slope was statistically significant ($B = 0.00$, s.e. = 0.00, Wald $Z = 2.40$, $p = 0.02$), suggesting that the negative relationship between *stress* and *compliance* varied significantly across countries; the result of likelihood

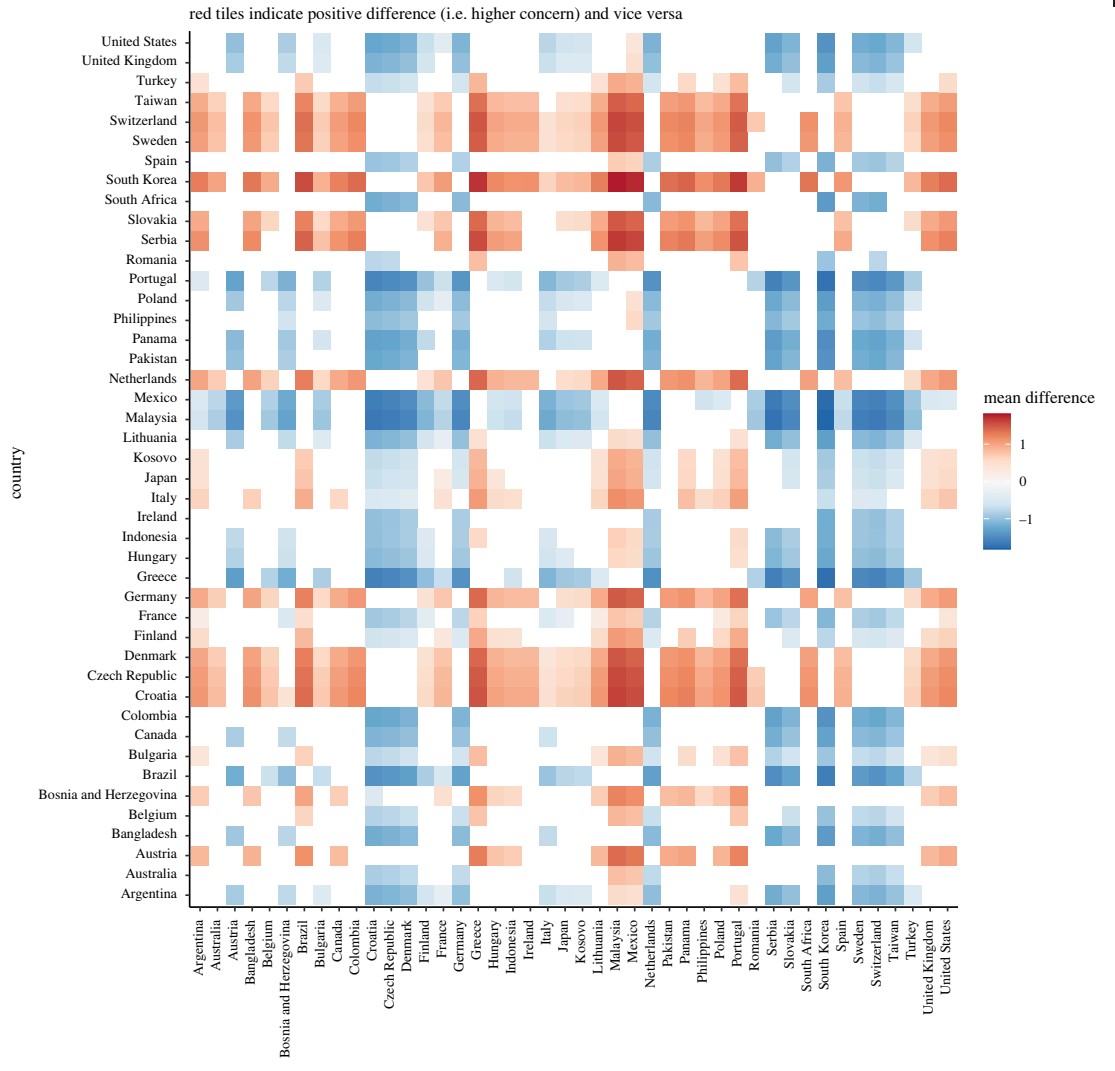

**Figure 2.** Cross-country differences in concern over the coronavirus (based on latent scores from alignment measurement invariance for between-country comparison; original scale: 1 = strongly disagree, 6 = strongly agree).

ratio test indicated that the inclusion of *stress* random slope significantly increased the model fit ($\chi^2_2 = 33.20$, $p < 0.001$).

The result from the Bayesian MLM was consistent with the frequentist MLM. The Bayesian MLM indicated very strong support (BF10 = infinite) for the presence of the non-zero effect of individual-level perceived *stress* on self-reported *compliance* (see OSF for the full report generated by brms; link: https://osf.io/7g8ej/). The estimated effect size of perceived stress was Cohen's $D = -0.06$, suggesting that the effect was very small.

## 11.3. Concern over the coronavirus and compliance with behavioural guidelines

Hypothesis 2b examined if *concern* over the coronavirus positively predicted *compliance* with behavioural guidelines. In Model 3, the individual-level variables together accounted for 6.29% and 11.07% of the level-1 within-country and level-2 between-country variance, respectively (0.72 and 0.04 versus = 0.77 and = 0.05 in Model 2; see electronic supplementary material, table S7). As predicted, *concern* over coronavirus positively predicted *compliance* with behavioural guidelines to slow the spread of coronavirus ($B = 0.12$, s.e. = 0.00, $t_{100185} = 55.43$, $p < 0.001$). In Model 4, the inclusion of country-level variables accounted for 51.10% of the level-2 between-country variance (0.02 as compared with 0.04 in

cross-country differences in trust in government efforts
red tiles indicate positive difference (i.e. higher trust) and vice versa

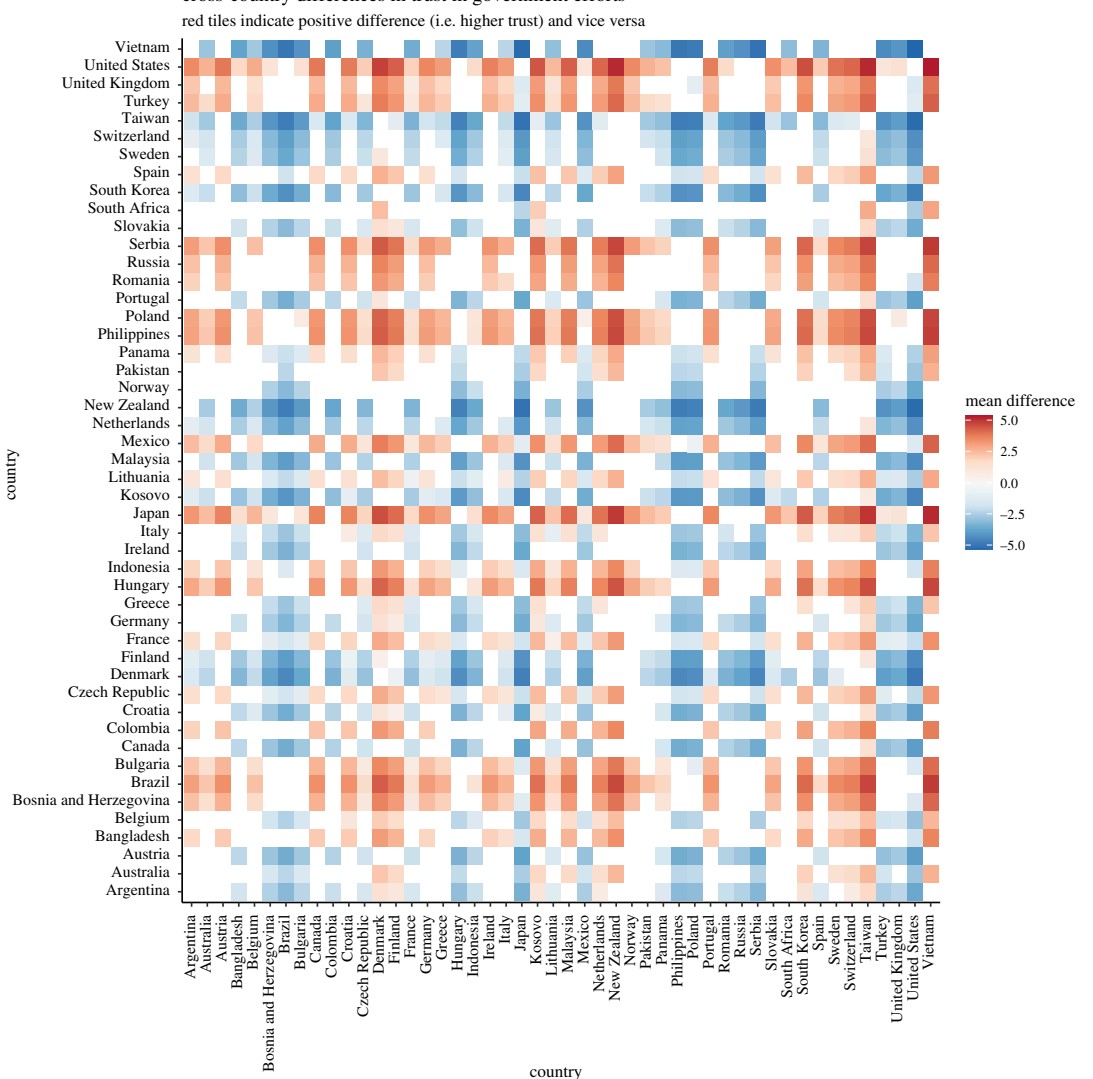

**Figure 3.** Cross-country differences in trust in government efforts (based on latent scores from alignment measurement invariance for between-country comparison; original scale: 0 = not at all, 10 = completely).

Model 3). Individual-level *concern* remained statistically significant ($B = 0.12$, s.e. $= 0.00$, $t_{100284} = 55.36$, $p < 0.001$) after controlling for country-level predictors. H2b was thus supported by the data.

In Model 5, the random slope of *concern* over coronavirus was statistically significant ($B = 0.00$, s.e. $= 0.00$, Wald $Z = 3.99$, $p < 0.001$), suggesting that the association between concern and compliance significantly varied across countries; the result of likelihood ratio test showed that the inclusion of *concern* random slope significantly increased the model fit, $\chi^2_2 = 418.40$, $p < 0.001$.

The result from the Bayesian MLM also supported H2b. The BF10 = infinite result indicated that evidence very strongly supported H2b against the null hypothesis (see OSF for the full report generated by brms; link: https://osf.io/7g8ej/). The calculated effect size of concern was Cohen's $D = 0.36$, suggesting the presence of a small to medium effect.

## 11.4. Trust in government efforts and perceived stress

Hypothesis 3a tested if *trust* in government efforts negatively predicted perceived *stress*. In Model 3, the individual-level variables together accounted for 12.33% and 51.83% of the level-1 within-country and level-2 between-country variance, respectively (0.85 and 0.04 versus 0.97 and 0.07 in Model 2; see electronic supplementary material, table S8). *Trust* in government efforts to slow the spread of coronavirus was negatively associated with *stress* ($B = -0.06$, s.e. $= 0.00$, $t_{98431} = 49.35$, $p < 0.001$), such that individuals with a higher level of trust in government efforts showed a lower level of stress. *Age*,

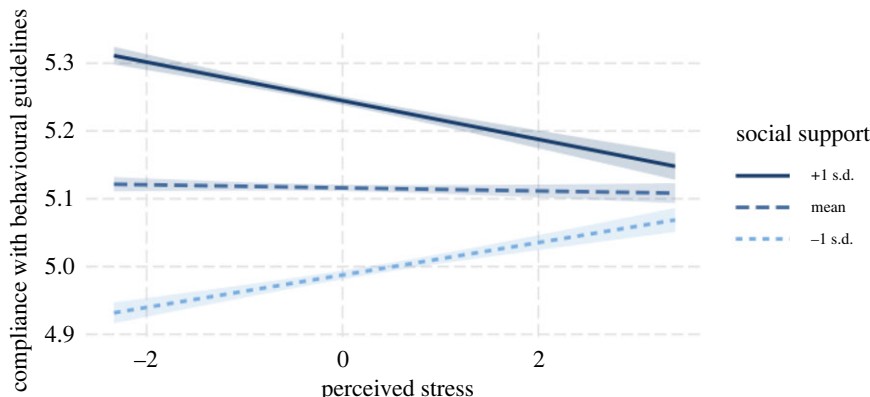

**Figure 4.** Cross country differences in compliance with behavioural guidelines (based on latent scores from alignment measurement invariance for between-country comparison; original scale: 1 = strongly disagree, 6 = strongly agree).

**Figure 5.** Availability of social provision moderating the relationship between perceived stress and compliance with behavioural guidelines (based on latent scores for social support and perceived stress from alignment measurement invariance).

*gender* and *education* significantly predicted perceived *stress*: individuals with younger *age* ($B = -0.02$, s.e. = 0.00, $t_{100589} = 83.68$, $p < 0.001$), female ($B = 0.34$, s.e. = 0.01, $t_{101012} = 50.19$, $p < 0.001$) or with lower *education* level ($B = -0.07$, s.e. = 0.02, $t_{101000} = 4.62$, $p < 0.001$; $B = -0.14$, s.e. = 0.02, $t_{101008} = 9.47$, $p < 0.001$) tended to report greater perceived *stress*. In Model 4, inclusion of country-level variables

accounted for 8.66% of the level-2 between-country variance (0.03 versus 0.04 in Model 3). The negative relationship between *trust* in government efforts and *stress* ($B = -0.06$, s.e. = 0.00, $t_{100969} = 49.14$, $p < 0.001$) remained statistically significant after country-level variables were controlled for. Hence, H3a was supported by the data.

In Model 5, the random slope for *trust* in government efforts was statistically significant ($B = 0.00$, s.e. = 0.00, Wald $Z = 3.36$, $p = 0.001$), suggesting that the association between government *trust* and *stress* significantly varied across countries. Inclusion of the random slopes significantly improved the model fit, $\chi_2^2 = 166.29$, $p < 0.001$.

The Bayesian MLM showed that evidence very strongly supported the presence of the non-zero effect of trust in government efforts, BF10 = infinite (see OSF for the full report generated by brms; link: https://osf.io/7g8ej/). The estimated effect size of trust in terms of Cohen's $D$ was $-0.31$, indicating the presence of a small to medium effect.

## 11.5. Trust in government efforts and concern over the coronavirus

Hypothesis 3b tested if *trust* in government efforts negatively predicted *concern* over the coronavirus. In Model 3, the individual-level variables together accounted for 2.91% of the level-1 within-country variance (1.52 versus 1.57 in Model 2; see electronic supplementary material, table S9). *Trust* in government efforts to slow the spread of coronavirus was significantly negatively associated with *concern* ($B = -0.01$, s.e. = 0.00, $t_{100644} = 5.36$, $p < 0.001$). In Model 4, the negative relationship between *trust* in government efforts and *concern* ($B = -0.01$, s.e. = 0.00, $t_{100685} = 5.29$, $p < 0.001$) was still statistically significant after country-level variables were controlled for. The inclusion of country-level variables accounted for 31.55% of the level-2 between-country variance (0.16 versus 0.24 in Model 3). However, in Model 5, when the random slope of government *trust* was entered in the model, the relationship between government *trust* and *concern* was no longer statistically significant, $t_{46} = 1.08$, $p = 0.29$. Analysis of random effects showed that random slope of government *trust* was statistically significant ($B = 0.00$, s.e. = 0.00, Wald $Z = 4.08$, $p < 0.001$) and adding this random effect significantly improved the model fit, $\chi_2^2 = 657.70$, $p < 0.001$. These results suggested that although government *trust* did not significantly predict *concern* when averaging all countries, the relationship between *trust* in government efforts and *concern* varied across countries to a great degree. H3b was therefore not supported.

The result of the Bayesian MLM also suggested that H3b could not be supported by the data. The calculated BF10 = 0.07 indicates that the null hypothesis was very strongly supported by evidence instead of H3b (see OSF for the full report generated by brms; link: https://osf.io/7g8ej/). The estimated effect size of trust, Cohen's $D = -0.04$, suggested the presence of a very small effect.

## 11.6. Concern as a predictor of stress

H4 tested the relationship between *concern* over the coronavirus and perceived *stress*. In Model 3, the individual-level variables accounted for 14.55% and 36.02% of the level-1 within-country and level-2 between-country variance, respectively (0.83 and 0.06 versus 0.97 and 0.07 in Model 2; see electronic supplementary material, table S10). *Concern* over the coronavirus was positively associated with perceived *stress* ($B = 0.18$, s.e. = 0.00, $t_{102284} = 77.92$, $p < 0.001$). In Model 4, the positive relationship between *concern* and *stress* ($B = 0.18$, s.e. = 0.00, $t_{102385} = 77.88$, $p < 0.001$) was still statistically significant after country-level variables were controlled for. The inclusion of country-level variables accounted for 13.83% of the level-2 between-country variance (0.04 versus 0.05 in Model 3). H4 was thus supported by the data.

In Model 5, random slopes of *concern* were statistically significant ($B = 0.00$, s.e. = 0.00, Wald $Z = 4.08$, $p < 0.001$), suggesting that the association between *concern* and *stress* significantly varied across countries. Inclusion of the random slopes significantly improved the model fit, $\chi_2^2 = 647.10$, $p < 0.001$.

The Bayesian MLM also supported H4. According to the Bayesian MLM result, evidence was found to support H4 very strongly against the null hypothesis, BF10 = infinite (see OSF for the full report generated by brms; link: https://osf.io/7g8ej/). The calculated effect size of stress, Cohen's $D = 0.49$, suggested the existence of a nearly medium effect.

## 11.7. Availability to social provisions as moderator

H5 examined if *social support* moderated the relationship between *stress* and *compliance* identified in H2a. At the first step, both individual-level and country-level *social support* was added in the Model 5 from

H2a (see Model 6 in electronic supplementary material, table S11). Individual-level *social support* significantly predicted compliance ($B = 0.10$, s.e. $= 0.00$, $t_{85745} = 37.16$, $p < 0.001$), so did country-level *social support* ($B = 0.15$, s.e. $= 0.06$, $t_{29} = 2.54$, $p = 0.017$), meaning that greater individual-level *social support* was associated with higher levels of compliance and that living in countries with greater *social support* also predicted higher levels of compliance. Subsequently, in Model 7, where *social support* intercept, random slopes and the covariates were introduced to the model, the random slopes of *social support* were statistically significant ($B = 0.00$, s.e. $= 0.00$, Wald $Z = 2.74$, $p = 0.006$), such that the relationship between *social support* and *compliance* varied significantly across countries; the result of likelihood ratio test showed that the inclusion of the random effects related to *social support* significantly increased the model fit ($\chi_3^2 = 137.34$, $p < 0.001$; electronic supplementary material, table S11F).

Next, in Model 8, both individual-level and country-level *stress × social support* interaction effects were entered in the model. As expected, there was a statistically significant individual-level *stress × social support* interaction, which suggests that *social support* moderated the relationship between *stress* and *compliance* ($B = -0.02$, s.e. $= 0.00$, $t_{6144} = 10.09$, $p < 0.001$).

The Bayesian MLM also supported H5. The BF10 $=$ infinite results indicated that evidence very strongly supported the presence of the non-zero interaction effect (see OSF for the full report generated by brms; link: https://osf.io/7g8ej/). The calculated effect size of the main effect of *social support*, Cohen's $D = 0.26$, suggested its effect was small to medium. The effect size of the interaction effect was very small, Cohen's $D = -0.07$.

Simple slope analyses were conducted to separately examine the strength of the *stress-compliance* association by high ($+1$ s.d.) versus low ($-1$ s.d.) levels of *social support* [93] (figure 5). When *social support* was high, perceived *stress* was found to negatively predict *compliance* ($B = -0.03$, s.e. $= 0.01$, $t_{38} = 4.31$, $p < 0.001$); however, when *social support* was low, perceived *stress* positively predicted *compliance* ($B = 0.03$, s.e. $= 0.01$, $t_{37} = 4.55$, $p < 0.001$). Taken together, and contrary to our hypothesis that greater social support would reduce the negative association between *stress* and *compliance*, we found that the relationship between *stress* and *compliance* in fact became more strongly negative among individuals with greater (and not lower) levels of *social support*.

## 11.8. Post hoc analyses

To examine why the association between *trust* in government was not negatively associated with *concern* over the coronavirus as we had hypothesized (H3b), we conducted additional exploratory analyses. As changes in institutional and political trust have been found to rise in the aftermath of lockdowns instituted in Western Europe [94] and New Zealand [2], and levels of trust have been shown to predict differing behavioural responses under varying levels of stringency in governmental policy responses to COVID-19 [95], one possibility is that the stringency of lockdowns themselves may have had additional effects on *trust*, and consequently in how *trust* relates to experienced *concern* over the coronavirus. Therefore, in this additional analysis, we explored whether the differences in the measures implemented in different countries were related to the association between people' *trust* in their governments and *concern* over the coronavirus.

We used the 2020 *Oxford Government Response Stringency Index* to quantify the degree to which each government's preventive measure was strict (see https://ourworldindata.org/grapher/covid-stringency-index). We extracted the index data dated Monday, 6 April 2020. Then, we performed one additional MLM and BMLM after modifying the model for H3b. For this exploratory analysis, we added the interaction effect of the individual-level *trust* in government efforts × country-level *stringency index*, and the country-level *stringency index* as a covariate to the initial model. We examined whether the interaction effect was statistically significant in predicting concern over the coronavirus based on the resultant *p*-value and BF10 by applying the same interpretation criteria (table 2).

The result of the *post hoc* analysis demonstrated that there was a statistically significant interaction effect of the individual-level *trust* in government efforts × country-level *stringency index* on *concern* over the coronavirus. Results from both the frequentist MLM ($B = 0.07$, s.e. $= 0.00$, $t_{36.18} = 3.49$, $p = 0.001$) and Bayesian MLM (BF10 $= 9.10$) supported this. Although not very strongly, the evidence positively supported the presence of an effect (BF $\geq 3$, table 2). The estimated effect size was very large, Cohen's $D = 1.16$. As shown in figure 6, we found that in countries where the government response measures were relatively strict (*stringency index* $\geq +1$ s.d.), the relationship between *trust* in government and *concern* over the coronavirus was positive while that was not the case in countries where the government response measures were less strict.

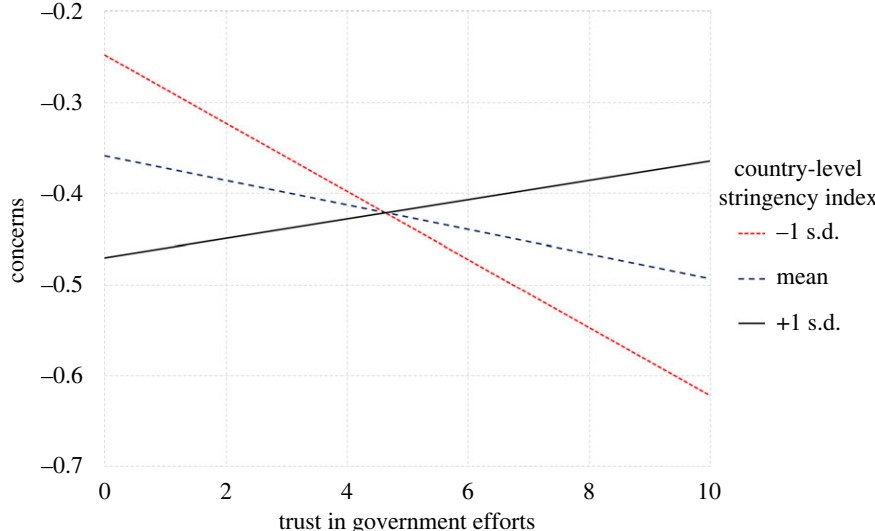

**Figure 6.** Country-level stringency index moderating the relationship between *trust* in government efforts and *concern* over the coronavirus (based on latent scores for concern over the coronavirus).

## 12. Discussion

Employing data from participants in 48 countries, we explored early psychological and behavioural responses to the 2020 COVID-19 pandemic, and how differences in psychological and behavioural responses influenced reactions to government efforts to curb the spread of the coronavirus and associated COVID-19 disease. There are five major findings discussed in turn.

First, the results revealed that there were marked differences between countries in the early months of the global COVID-19 crisis. Participants from Western European nations were more concerned over COVID-19, more stressed and had lower levels of trust in the government's efforts, compared with participants from other parts of the world. In their research comparing countries in Europe, Asia and North America, Dryhurst *et al*. [96] found that levels of risk perception were high among all countries for the period between March and April, with the highest levels being reported in the UK and Spain. In terms of compliance with local behavioural guidelines to slow the spread of coronavirus, there was no clear distinction in the geographical pattern in our data, with the exception of Japanese participants who reported particularly high levels of compliance (figure 4). In line with existing research showing cross-country differences in psychological responses to pandemics (e.g. [52,53,54]), the COVIDiSTRESS global survey finds that this was indeed also the case for early psychological responses to COVID-19. Our findings contribute to the research undertaken by other cross-country surveys highlighting the importance of the psychological underpinnings of public responses to the pandemic (e.g. Winton Centre for Risk and Evidence Communication; [96]).

Second, we found that concern over the coronavirus was positively associated with stress. The more concerned people were, the more stress they experienced. While this relationship between concern and stress seems to be a general pattern during pandemics (e.g. [22,24]), there were notable differences between countries, and concerns about the coronavirus itself did not account for all of the variances in experienced stress during the early months of coronavirus restrictions. As such, we found that while concern about the disease itself was a source of mental distress and perceived pressure, other factors including, for example, working conditions and children's education during lockdown must also be taken into consideration to fully understand the psychological impact of COVID-19 [97]. Our findings add to the existing findings from studies that have highlighted the negative economic impact of the pandemic and its association with anxiety or depression [98], as well as the overall negative psychological effects of the pandemic on parents [99].

Third, we found that the more stressed people felt, the less compliant they were with behavioural guidelines; but also that more concerned individuals tended to display more behavioural compliance. These associations between compliance with local behavioural guidelines, and the psychological factors of perceived stress and concern, respectively, differed across countries. Again, the early COVID-19 period conformed to previous research showing that perceived stress and concern over a pandemic related to people's compliance (e.g. [17–19,31–33]). However, our analyses also contribute

further, by providing a detailed cross-country comparison for relationships between early psychological and behavioural responses. While the full breadth of between-country differences is too numerous to discuss, interested readers can study these comparisons in figures 1–4. Specific country-level information can also be found in the electronic supplementary material and online datasets (https:// osf.io/z39us/ and [59]).

Fourth, the results showed that the more trust people had in government efforts to slow the spread of coronavirus, the less stress they experienced. Again, cross-country differences could be rooted in a number of factors ranging from differences in local cultures and political history, to variations in behavioural guidelines, government intervention stringency and the impact of COVID-19 at the time. Contrary to the relationship between trust in government efforts and perceived stress, we did not find support for the hypothesis that trust in government efforts to slow the spread of the coronavirus would be negatively correlated with concern over the coronavirus. It would have seemed intuitive that trusting that appropriate measures were being taken by authorities would lessen concern. Instead, our results revealed considerable variation in associations between trust in government efforts and concern across different countries. In an effort to examine why this could be the case, we conducted additional exploratory analyses. Here, we found a significant positive interaction effect of the individual-level trust in government with the country-level *Oxford Government Response Stringency Index*. The emerging patterns indicate that the relationship between trust and concern became positive only in countries where the measures were objectively strict in comparison with other countries. In other words, most people were concerned over the early COVID-19 situation, but this perception was accompanied by trust in governmental measures only if such measures were relatively strict.

There could be multiple explanations for this exploratory finding. Strict measures may instil a belief that all possible courses of action are being employed to alleviate the situation. We also know from H2 that stress is negatively associated with compliance—but given our findings from H3, namely the negative correlation between perceived stress and trust in government efforts, it could be that better government intervention reduces stress *along with* the perceived need for compliance, if things seem well in hand. Further, it is plausible that governments will be more likely to implement stronger measures when they perceive a mandate from the populace *combined with* great concern. Factors like cohesion within society and the ability of each country to withstand downstream economic consequences will probably also play a role in this dynamic. Casualties could thus run in multiple directions, and are likely to interact. Research on the lockdowns implemented in Western Europe in March and April concurrently found an association between lockdowns and greater trust in government, political support for the governing party, and democratic satisfaction [94], suggesting that lockdowns are indeed generally popular among citizens in well-functioning democracies and may feel reassuring to citizens concerned over the progression of the disease; similar effects on trust were also observed in New Zealand [2]. However, countries in which less stringent lockdowns are applied may have experienced a reverse effect, where those most likely to be satisfied by a light government response may also be those who are least concerned with COVID-19 to begin with. Along these lines, a recent cross-country study by Fetzer *et al.* [100] found that timely and stringent government measures were associated with better mental well-being among citizens, while negative evaluations of the government's response to the pandemic were related to lower mental well-being and a lack of trust in government efforts to tackle the pandemic. Another analysis of mobility in European countries during the early weeks of the pandemic revealed gaps in compliance between populations that had high or low political trust as the stringency of lockdown increased [95], perhaps in part due to varying concern levels, as we also found support for in H2b.

Fifth, greater availability of social support moderated the negative association between stress and compliance. Among people with higher levels of social support, stress was associated with less compliance with local preventive measures, while among people with lower levels of social support, stress was associated with more compliance. These outcomes run counter to our initial hypotheses, which might suggest that complying with society-level preventive measures was an especially attractive coping strategy for individuals who did not have the luxury of being looked out for by friends, family and neighbours. It might also be that those placing more stock in near relations for coping underestimated the importance and legitimacy of country-level restrictions. There might be several reasons why the evidence contradicted H5. Chang & Sivam [101] reported that the perceived stress and SARS-related fears were positively associated with preventive behaviours among Singaporians during the SARS outbreak. Similarly, Tang *et al.*'s [102] study showed that the perceived stress positively predicted the tendency of social distancing in early 2020 among participants in China and Hong Kong, places where the COVID-19 was more severe compared with other countries during

the period. These previous studies would suggest that people tend to adopt behavioural measures to prevent the spread of the virus if they experience significant stress at the epicentres of pandemics, but our findings add the nuance that the psychological buffering role of social support may actually counteract the role of stress as a driver of cautious behaviour in the public sphere. Rather than looking at this phenomenon as a function of individual stress levels, this buffering mechanism could come into play when groups oppose government restrictions for political or practical reasons, or when younger adults converge on the need to retain their social life in the face of pandemic curfews.

A notable limitation of the present study was the sampling method. While other studies (e.g. [3]) have collected survey data using paid representative respondent pools, the goal of the COVIDiSTRESS collaboration was to collect data as quickly and conveniently as possible, with researchers in each country deciding their own recruitment strategies. As such, although spanning across a large number of countries, our sample is not representative of the various national populations. For instance, the COVIDiSTRESS dataset has more female respondents, and, presumably due to the use of online media for recruitment, a younger demographic. Full dataset details can be found in the electronic supplementary material, and on the Open Science Framework.

## 12.1. Practical implications

The present analysis of the COVIDiSTRESS dataset contributes to an understanding of the complexities at play in early psychological effects of the COVID-19 pandemic. Our findings lead us to suggest several implications for practitioners such as health communicators, government officials and policy makers. Foremost, we find that although there are general patterns, between-countries differences in reactions to the crisis and compliance with behavioural guidelines show that interventions should be context sensitive. The actions already taken by governments, as well as the stress and stressors pervading a society, can at any given time lead to very different reactions to the same disease, the same restrictions, or the same recommendations from health or government authorities.

Lowering stress in the population may be needed to create optimal conditions for compliance with behavioural guidelines. Establishing trust in government efforts by taking decisive and visible action, and addressing stressful issues such as childcare, lockdown predictability or job security during a pandemic, may for instance lead to better compliance by way of lessening the stresses experienced by individuals. However, care must be taken not to lessen concerns over the gravity of the health situation at the same time, as our data suggest that lessened personal concern over the virus, or reliance on close relations rather than collective societal action to handle everyday challenges, can run counter to accepting the need for protective behaviours in society at large. Taken together, this implies that governments should be as transparent and objective as possible in providing guidelines and updates of the most current situation in terms of the number of positive and recovered cases as well as deaths. Initial downplaying or hiding actual numbers of positive cases or deaths can have detrimental effects on public health as well as on institutional trust and levels of compliance as the pandemic surges. In addition, clarity in communicating governmental guidelines to reduce transmission has also the potential to alleviate perceived stress.

Interventions to foster compliance with behavioural guidelines can thus be made more effective in their content and targeting, when based on knowledge of the target groups' current and baseline level of *perceived stress*, *concern over COVID-19* and *trust in government efforts*. These three psychological constructs are antecedents of compliance with behavioural guidelines to slow the spread of COVID-19 and vary considerably between different countries. As shown, however, the relationship between these constructs is quite complex and varies according to numerous local factors, including the stringency of government interventions in each country, and tracking their relationship over time, and in different locales and social groups, could thus be an important tool for continuously adapting social, psychological and communicative strategies when responding to global pandemics. Further, in order to successfully communicate and implement protective behavioural measures, it appears important to base messages and channels on whether individuals in a target audience are oriented towards the measures in society more broadly, or the personal support of social groups, where the latter may reduce stress and compliance both (i.e. [103]).

Even though general trends can be detected across the globe, we found notable between-countries variations in all our analyses. With a dataset comparing so many countries, it is infeasible to unpack all of these specific differences in the text properly. Instead, these must be considered individually though scrutiny of the graphs and electronic supplementary material, and possibly explained by future dives into the large open data pool combined with other sources of information. For instance,

the timeframe for outbreaks has varied greatly across the planet, as has the capacity of health systems to treat the seriously afflicted, and measures to curtail the spread of the coronavirus. Also, while the behavioural guidelines of some governments and health authorities were presented as mere recommendations, other countries, regions and cities enforced strict closing of businesses, schools and public spaces, as well as curfews enforced by local police and military personnel. Communication and execution of these guidelines have also changed over time, and as many such guidelines may have shifted from being perceived as temporary emergency measures to a longer term 'new normal' by summer 2020 [103], psychological responses to the same measures may shift in turn. Further, factors like cultural norms, distribution of wealth, living conditions, proximity to neighbours, typical work life and trust in authorities all varied before the pandemic in myriad ways, which must be taken into account when interpreting the difference between any two countries as they emerge in the results from H1a to H1d (figures 1–4). For this reason, we invite detailed scrutiny and discussion of any interesting between-country differences using the open science online resources made available as code, data, electronic supplementary material and dynamic visualizations [59].

## 12.2. Further research

After-effects of the COVID-19 pandemic are to be expected, and just as the emergence of the global pandemic presented new personal and societal challenges, re-opening and getting 'back to normal' will vary according to how each country opts to face these challenges [103]. As such, follow-up data collection will be important to assess the whole of the impact of the COVID-19 pandemic, as well as how attitudes and psychological reactions changed from 2020 and into the coming years. Conducting a further and deeper between-countries analysis taking into account the preventive and communicative strategies employed by each government, as well as the seriousness of the various consequences from the medical to the economic spheres is thus necessary to understand the effects of the pandemic more broadly, over time. We have included the stringency of intervention in the *post hoc* work of the present pre-registered analysis, but many more such variables must be considered in the future.

Finally, we believe that it is important to discuss the broad scientific climate of which the COVIDiSTRESS global survey is only a small part. As the global pandemic gained speed, researchers from a plethora of fields felt compelled to send out a barrage of surveys on issues ranging from conspiracy theories about the disease to experiences of parenting and working/learning from home. Indeed, multiple repositories and consortia also cropped up to support collaboration and avoid repetitive research questions in favour of a coordinated global undertaking. While these many research initiatives are bound to vary in quality, and contents will overlap, the period was an amazing example of the scientific community coming together to address a global crisis, and it is likely that the coronavirus pandemic and the lockdowns that ensued will be the most well-studied social event in human history from a perspective of real-time data collection. As such, future reviews and meta-analyses taking measures, sampling and overall study quality into account will lend a much deeper understanding. As a consortium, we believe that especially studies using the open data, pre-registration and publishing in the registered reports format will be decisive in the post-COVID-19 discussion of scientific quality and integrity during global crises. As the surge of new cases during the winter of 2020–2021 demonstrated, the pandemic had the potential to create lasting consequences beyond the measures implemented in the spring of 2020 and their after-effects. As such, longitudinal studies tracing the impact of protective measures on mental health would be particularly welcome. In a similar vein, it would be equally important to differentiate how infected versus uninfected individuals are influenced by the pandemic and related measures to contain it. Emerging research shows that individuals who recover continue to experience increased levels of anxiety, worry and poorer mental health [104]. Thus, future research must focus on comparative studies that investigate coping and stress-related maladaptive strategies and psychological recovery across infected and uninfected populations.

Ethics. Due to the pressing nature of COVID-19 research, a waiver to proceed has been acquired from the Aarhus University board of research ethics office, stating that the study is in accordance with the IRBs standards for acceptable research in human subjects. To further ensure the participant's interests and compliance with GDPR-standards, all data are anonymous, and no medical information is collected.

Data accessibility. The COVIDiSTRESS global survey is a collaborative open science project. The data used here, and all other data including raw data and advice on cleaning/coding are made available at the Open Science Framework https://osf.io/z39us/.

Authors' contributions. A.L. did stage 1 and 2 writing, review and editing; consortium organization; survey design and methodology; data management. S.-Y.L. did stage 1 and 2 writing, review and editing; stage 2 data analysis/visualization. S.S. did stage 1 and 2 writing, review and editing; stage 2 data analysis/visualizations. H.H. did stage 1 and 2 writing, review and editing; stage 2 data analysis/visualization. M.K. did stage 1 and 2 writing, review and editing; stage 2 data analysis/visualization; data management. R.G. did stage 1 and 2 writing, review and editing. S.C. did stage 1 and 2 writing, review and editing. T.P.T. did stage 1 and 2 writing, review and editing; stage 2 data analysis/visualization. A.J. did stage 1 and 2 writing, review and editing. J.R. did stage 1 writing, review and editing; data management. H.C. did stage 1 and 2 writing, review and editing. T.L.M. did stage 2 writing, review and editing; stage 2 data analysis/visualization. Contributions of individual COVIDiSTRESS consortium co-authors are specified in the electronic supplementary material. Authors and consortium co-authors have read and approved the manuscript before submission.

Competing interests. Dr Taciano Milfont acted as a referee in the first two rounds of review of the stage one registered report. Upon discussion of the appropriateness of particular analysis methods in his field of expertize, and with the consent of RSOS editors, it has been agreed that he will join the list of authors when the RR proceeds to stage two instead of acting as referee. No competing interests are declared.

Funding. We received no funding for this study.

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
