## [Peer Review File · Royal Society Open Science]

Review History

RSOS-200589.R0 (Original submission)

Review form: Reviewer 1

Do you have any ethical concerns with this paper?

No

Recommendation?

Major revision

Comments to the Author(s)

Review of: The impact of stress and worry from the coronavirus pandemic on trust and compliance across 45 countries - data from the COVIDiSTRESS global survey (Manuscript # RSOS-200589)

This is a Registered Report describing how data COVIDiSTRESS global survey will be used to investigate on psychological and behavioural responses to the COVID-19 pandemic across over 40 countries. First, I would like to thank the Editor for allowing me to contribute with this study as a reviewer, and to congratulate the authors for co-ordinating the COVIDiSTRESS and for this planned manuscript. I believe this will be a landmark project/study within psychology. (For disclosure, I am not involved with the COVIDiSTRESS project.)

In my view, the scientific validity of the research questions are strong and timely, there is clarity and methodological detail to allow future replication of the analytical approach (if the scripts are made available), and the level of detail provided reduces researchers degrees of freedom. However, I have noted issues that question the scientific validity of the research questions and proposed hypotheses, as detailed below.

Measurement Invariance

Overall, I believe the planned study and analyses are sound but there is a main issue that compromise all hypotheses and analytical plan: no testing of measurement invariance. I recommend Major Revision because this is a serious omission.

As we wrote in the abstract of an article on this topic, "When comparing groups an assumption is made that the instrument measures the same psychological construct in all groups. If this assumption holds, the comparisons are valid and differences/similarities between groups can be meaningfully interpreted. If this assumption does not hold, comparisons and interpretations are not fully meaningful. The establishment of measurement invariance is a prerequisite for meaningful comparisons across groups." (Milfont & Fischer, 2010).

There are three main levels of invariance testing: configural (underlying factor model identical across groups), metric (item loadings identical across groups), and interval (item intercepts identical across groups). The specialised literature argues that metric invariance is needed for examining associations between constructs across groups, and that mean comparisons across groups are only meaningful when interval invariance is achieved. All hypotheses in the Registered Report focus on mean comparisons or associations between constructs across cultural groups but no invariance testing is proposed. This needs to be addressed! Without testing and confirming the multi-item measures are equivalent across the groups, the testing of all proposed hypotheses are meaningless. For example, if participants across the considered nations do not respond to items of the Perceived Stress Scale similarly, then comparing stress means across countries (H1a) can't be meaningfully interpreted.

When revising the Registered Report to include measurement invariance, I suggest the authors to explore the alignment method, which is a much easier implementation of invariance testing – particularly for large groups as in this case. I believe alignment can already be implemented in R. I recommend two main publication on the topic: Asparouhov and Muthén (2014) and Marsh et al. (2018); and we have also implemented this approach in a recent article (see Milfont et al. 2018) and have discussed measurement invariance for replicating findings across cultural groups (see Milfont & Klein, 2018). I am sure the authors would know measurement invariance experts in their networks but I can make myself available to provide some guidance if needed (and if the Editor and authors feels this is appropriate).

Issues with Hypotheses

Beyond the main issue regarding measurement invariance noted above, I noted issues with the proposed hypotheses.

1. I feel the authors should make it explicit from the outset that they will employ a multilevel approach.
2. The authors propose a list of country-level variables and use them as control. This is a reasonable list of variables but there is no justification of their selection. This list can be much longer, especially if the authors decide to include psychological dimensions of cultural variability such as individualism/collectivism, tightness/looseness and relational mobility. It is thus important to provide some rationale for selecting only non-psychological country-level variables and for selecting these specific ones.
3. Related to the point above, the country-level variables are merely used as control variables in the analyses. It would be great if these were included as predictors, at least for the first four hypotheses. Once invariance is established and meaningful comparisons can be made (see above), merely showing variability in stress, concern, trust and compliance might not tell much, or might be difficult for the readers and government officials to interpret the observed similarities/differences. It would be more interesting to show whether stress, concern, trust and/or compliance is greater/weaker in wealthier/poorer countries, for example. (Again, the list of variables could be very large and such analysis would not be attainable; by addressing the point above and providing justification for a small list of variables selected, the authors could provide such addition to the first hypotheses.)
4. H3b reads "Across countries, trust in government efforts to slow the spread of coronavirus will negatively predict concern over coronavirus, when controlling for individual- and country-level covariates" but this negative prediction is not obvious to me. Greater concern should lead to stronger trust, no? Or alternatively, if I don't trust the government efforts (take delayed and denialistic actions by the USA and Brazil for example), I would be more concerned about the pandemic, no?
5. I found H5 unclear. Do you mean that provisions would reduce/buffer the expected negative effect of stress on compliance (H2a)? You need to spell out. Given the relations between stress and concern (see H2a, H2b and H4), I believe another moderation hypothesis is reasonable and could be included by examining the extent to which provisions increases/enhances the expected positive effect of concern on compliance.

Minor Points

I have made comments in the original PDF (Appendix A). I have highlighted sentences that are unclear of miss words, and have added missing words too (e.g., provide clear direction for some of the hypothesis).

I wish you the authors good luck with the project. Congratulations again for leading such an impressive and important research project!

Taciano L Milfont

<https://orcid.org/0000-0001-6838-6307>

References

Asparouhov, T., & Muthén, B. (2014). Multiple-group factor analysis alignment. *Structural Equation Modeling: A Multidisciplinary Journal*, 21, 495–508.

Marsh, H. W., Guo, J., Parker, P. D., Nagengast, B., Asparouhov, T., Muthén, B., & Dicke, T. (2018). What to do when scalar invariance fails: The extended alignment method for multi-group factor analysis comparison of latent means across many groups. *Psychological Methods*, 23, 524–545.

Milfont, T. L., Bain, P. G., Kashima, Y., Corral-Verdugo, V., Pasquali, C., Johansson, L.-O., Guan, Y., Gouveia, V. V., Garðarsdóttir, R. B., Doron, G., Bilewicz, M., Utsugi, A., Aragones, J. I., Steg, L., Soland, M., Park, J., Otto, S., Demarque, C., Wagner, C., Madsen, O. J., Lebedeva, N., González, R., Schultz, P. W., Saiz, J. L., Kurz, T., Gifford, R., Akotia, C. S., Saviolidis, N. M., & Einarsdóttir, G. (2018). On the relation between social dominance orientation and environmentalism: A 25-nation study. *Social Psychological and Personality Science*, 9, 802–814.

Milfont, T. L., & Fischer, R. (2010). Testing measurement invariance across groups: Applications in cross-cultural research. *International Journal of Psychological Research*, 3, 112–131.

Milfont, T. L., & Klein, R. A. (2018). Replication and reproducibility in cross-cultural psychology. *Journal of Cross-Cultural Psychology*, 49, 735–750.

Review form: Reviewer 2

Do you have any ethical concerns with this paper?

No

Recommendation?

Major revision

Comments to the Author(s)

Please see attached (Appendix B).

Review form: Reviewer 3

Do you have any ethical concerns with this paper?

No

Recommendation?

Accept in principle

Comments to the Author(s)

This is a well-documented and thought-out study. The topic is highly relevant and interesting. I have no major concerns. I have some smaller points, which the authors might want to consider.

It is understandable that with a study of events as they are unfolding, it is difficult to have very specific predictions. The predictions listed by the authors are relatively general but I find this appropriate in the current context. I suspect that the descriptive statistics might be very informative in their own right.

In the comparison across countries, it might be useful to have some measure of the stage of the pandemic that each country is in at the time of the survey, as this is likely to impact the responses. For example, if the situation in a country has stabilized somewhat (at least optically) people are likely to be less anxious than people who live in countries that are in earlier stages of the pandemic.

The exclusion criteria seem reasonable, both at the country level and at the respondent level. The authors might consider including an attention check, of which there are many examples in the literature.

The proposed analyses use both frequentist and Bayesian statistics. The authors specify the manner in which they will interpret these statistics, both separately and in conjunction.

Review form: Reviewer 4

Do you have any ethical concerns with this paper?

No

Recommendation?

Accept in principle

Comments to the Author(s)

This could be a great paper! Unfortunately I did not see the survey being spread around, though I am in a very research-intensive part of the world. I'd suggest increasing its visibility across Europe.

Review form: Reviewer 5 (Katherine S. Corker)

Do you have any ethical concerns with this paper?

Yes

Recommendation?

Major revision

Comments to the Author(s)

The current paper is a rapid registered report examining associations between self-reported variables in different countries during the COVID-19 pandemic. I considered the study proposal itself along with the data/materials shared on OSF. Data collection is already in progress, and we are asked to review the study's methods (not changeable) and its analysis plan (changeable) for this RR. Overall, there are several things to like about this paper. First, in spite of having been rapidly assembled, the authors appear to be taking steps to ensure the quality of the resultant data. I have several suggestions for improvement of the project.

1. The biggest issue is that the authors do not plan to test for measurement invariance before proceeding to their focal analyses. A lack of measurement invariance can compromise the validity of conclusions. I recommend the use of multiple group models using SEM to test invariance. There are several resources I can recommend to learn about this technique if necessary: Chen (2008, JPSP) and the Kline SEM textbook are good introductions.

2. The authors state “Continuous variables are centred on grand means, or converted to a z-score for the convenience of interpretation” (p. 10). Centering is a good idea; standardizing (converting to a z-score) is a bad idea in multi-level models (because you disturb the variance structure across levels). Avoid standardizing data in multilevel analyses. Rely on conversions to quasi-r-squared (or quasi-r) to interpret effect sizes. Effect sizes should absolutely be interpreted alongside p-values.

3. The authors have clear associations that they plan to test. At least as the introduction is currently written, these tested associations do not appear to stem from any particular theory or model. I don’t necessarily have an issue with this, but the tested models are not confirmatory in the sense that they test a theory-driven hypothesis. The introduction could benefit from a model or framework if there is one. I.e., why test these associations and not other models?

4. The authors intend to model a variety of demographic country level variables. I wonder if there are other pandemic-specific country level variables that should be included – e.g., rate of country-level hospital bed or protective equipment shortages, date of lockdown onset (or presence of lockdown), previous pandemic incidence. Further, I wonder if time of data collection should be explicitly modeled (e.g., days from date of survey to onset of lockdown). I don’t have strong feelings about this, but wanted to make the suggestion.

5. My preference would be to see all models with and without controls included. The authors could present their preferred specification (as presented here) and contrast it with models without level 1 and level 2 controls.

6. Hypotheses 1a to 1d should be tested with multi-level models (not ANOVAs). The reason is that country should be a random intercept, not a fixed effect. Hypotheses 2-5 should include and test random slopes for the predicted effects. For all hypotheses, I recommend a series of nested models: a null model (no predictors, no random effects), a random intercept only model (country as random intercept), a model with fixed effects of predictors, a model adding control variables, a model with random slopes for predictors. Each model can be compared to its predecessor via a likelihood ratio test. Strive to specify maximal models, but trim random effects (usually random slopes) if there are convergence issues.

7. Plan to report descriptive statistics for all variables (means, standard deviations, alpha or omega reliability, intercorrelations between variables). I don’t know these variables well enough to anticipate whether there might be problems with multi-collinearity of predictors in the planned models. Anticipate and plan for dealing with multi-collinearity if necessary.

8. It’s not clear in the proposal how country level variables will be scaled. That is, will they be treated as continuous? Contrast coded categories? For individual level predictors, I do wonder if the authors should plan to collapse educational categories (perhaps to just three: did not finish compulsory schooling in that country; finished secondary schooling; university degree or higher). If you choose to do this, I would contrast code the categories rather than treat them as continuous.

9. It’s not clear why for hypothesis 1c, the focus is on OECD_Institutions item 1, rather than all OECD_Institutions items. Should these items be combined into a total score if possible? Either way, would be good to have more reasoning behind this choice.

10. Some variables are measured with single items that may have limited reliability/validity. This is unavoidable given that we cannot alter the design at this point. Just be aware that this may further compromise statistical power of some analyses.

11. Regarding power, the authors state “we plan to recruit at least 30 participants per country so as to detect both the effects of individual- and country-level predictors” (p. 9). They cite Arend and Schäfer (2019)’s Table 8 as justification. They note that in the ESS database, medium ICCs, small to medium effects of level 1 and level 2 predictors, and small cross-level interactions are observed. They expect to collect data from 44 countries (level 2 units). The level 1 sample size (individuals per country) could vary. Of the 10 hypotheses, four focus on the ICCs themselves, five focus on level 1 predictors, and one contains an interaction between two level 1 variables. In six of the hypotheses, level 2 control variables are included. Based on Arend and Schäfer (2019)’s Table 8, with 44 countries and a sample size of 30 per country, small level 1 effects are underpowered (medium and large level 1 effects are adequately powered), regardless of the ICC. Only large effects of level 2 variables are adequately powered (regardless of ICC); small and medium effects are underpowered. Only large cross-level interactions under conditions of large random slopes are adequately powered; most random slopes are underpowered. Adding additional participants per country seems to matter less than adding more countries (or more broadly, level 2 units). Thus, it seems unavoidable that the current attempt will be underpowered in some ways, mostly because there is likely not an easy way to recruit more countries into the study. Having 30 participants per country seems reasonable based on the power tables, but the authors may run into issues with convergence with this size of sample per country. In particular, the multi-group invariance models tend to do better with larger per group sizes. I don’t see a strong reason to cut data collection short; it would seem prudent to get as much data as possible while the pandemic is happening. The results of the study are not themselves time sensitive (no government is going to base decision making on what these results show). Therefore, I would err on the side of sampling strategy that allows the collection of as much data as possible.

12. What is the rationale for the 15% cutoff for identifying floor/ceiling effects? Why not 10 or 20? Seems arbitrary. The exclusion rules for fast response time and Mahalonabis distance seemed appropriate and a good practice to improve data quality.

13. Causal language: There is causal language in the title of the paper (“the impact of”) and sprinkled throughout the paper itself. Causal language is inappropriate for this study’s methods.

Signed,
Katherine S. Corker

Decision letter (RSOS-200589.R0)

14-Apr-2020

Dear Dr Lieberoth,

The Editors assigned to your Stage 1 Registered Report (“The impact of stress and worry from the coronavirus pandemic on trust and compliance across 45 countries- data from the COVIDiSTRESS global survey”) have now received comments from reviewers. We would like you to revise your paper in accordance with the referee and editors suggestions which can be found below (not including confidential reports to the Editor). Please note this decision does not guarantee eventual acceptance.

To revise your manuscript, log into <http://mc.manuscriptcentral.com/rsos> and enter your Author Centre, where you will find your manuscript title listed under “Manuscripts with Decisions.” Under “Actions,” click on “Create a Revision.” Your manuscript number has been

appended to denote a revision. Revise your manuscript and upload a new version through your Author Centre.

When submitting your revised manuscript, you must respond to the comments made by the referees and upload a file "Response to Referees" in "Section 2 - File Upload". Please use this to document how you have responded to the comments, and the adjustments you have made. In order to expedite the processing of the revised manuscript, please be as specific as possible in your response.

Kind regards,
Andrew Dunn
Royal Society Open Science
openscience@royalsociety.org

on behalf of Professor Chris Chambers (Registered Reports Editor, Royal Society Open Science)
openscience@royalsociety.org

Associate Editor Comments to Author (Professor Chris Chambers):

Associate Editor: 1

Comments to the Author:

Five expert reviewers have now assessed the protocol, and I have read it myself, and before going further I want to extend my sincere gratitude to the reviewers for contributing such extraordinarily detailed, insightful and constructive assessments within 48 hours of accepting the review request. Of the more than 200 Registered Reports I have edited, I cannot think of a submission that has received a more attentive and helpful set of reviews.

As you will see, the reviewers are broadly positive about the proposal, praising the value and validity of the research question and overall strength of the design. However, collectively they also note shortcomings in the rationale, design and statistical plans that span the full range of the Stage 1 RR criteria at RSOS and will need to be thoroughly resolved to achieve Stage 1 in-principle acceptance. Given the detail in the reviews, I want to offer the authors some guidance in addressing the key concerns.

First, Reviewer 1 notes a number of design weaknesses, but most notably the lack of consideration of measurement invariance to ensure the validity of cross-group comparisons (a point also echoed by Reviewer 5). Reviewer 1 also critiques the clarity and rationale of the hypotheses and analysis plans, and helpfully supplies an annotated PDF (attached) highlighting typographical errors and additional points of concern.

Reviewer 2 notes a lack of coherence and clarity in the introduction and questions the assumptions underlying the hypotheses as well as the use of causal language (a point also noted by Reviewer 5), justification of H5 especially, and potential concerns with the limited sample size per country and lack of multiple comparisons corrections. The reviewer also points to a lack of positive controls (as does Reviewer 3), and insufficient detail about specific methodological components (such as recruitment).

Reviewer 3 is very positive and does not ask for major revisions, but recommends consideration of the stage of the pandemic in any cross-country comparisons. The reviewer also suggests the

inclusion of an attention check to address lack of positive controls, which is a point that occurred to me also in reading the proposal.

Reviewer 4 is similarly very positive while also questioning the rationale and justification of the hypotheses. The reviewer also raises the interesting point about validation of surveys in different languages and the limitations arising.

Finally, Reviewer 5 provides a detailed and constructive critique that echoes many points raised by the other reviewers, including the consideration of measurement invariance, pandemic-specific variables, lack of theoretical justification of the hypotheses, and concerns about sample size and moderation of causal language. Among the reviewer's chief recommendations is that the authors use multi-level models in place of the ANOVAs for several hypotheses.

On top of these headline issues, the reviews contain a rich set of more detailed recommendations that are too extensive to summarise. Although the reviews are critical, I want the authors to take encouragement from their positive tenor, and while substantial work is needed to reach IPA, the goal is certainly within reach provided the authors feel equipped to address the points raised. On this basis, a Major Revision is recommended.

Comments to Author:

Reviewer: 1

Comments to the Author(s)

Review of: The impact of stress and worry from the coronavirus pandemic on trust and compliance across 45 countries - data from the COVIDiSTRESS global survey (Manuscript # RSOS-200589)

This is a Registered Report describing how data COVIDiSTRESS global survey will be used to investigate on psychological and behavioural responses to the COVID-19 pandemic across over 40 countries. First, I would like to thank the Editor for allowing me to contribute with this study as a reviewer, and to congratulate the authors for co-ordinating the COVIDiSTRESS and for this planned manuscript. I believe this will be a landmark project/study within psychology. (For disclosure, I am not involved with the COVIDiSTRESS project.)

In my view, the scientific validity of the research questions are strong and timely, there is clarity and methodological detail to allow future replication of the analytical approach (if the scripts are made available), and the level of detail provided reduces researchers degrees of freedom. However, I have noted issues that question the scientific validity of the research questions and proposed hypotheses, as detailed below.

Measurement Invariance

Overall, I believe the planned study and analyses are sound but there is a main issue that compromise all hypotheses and analytical plan: no testing of measurement invariance. I recommend Major Revision because this is a serious omission.

As we wrote in the abstract of an article on this topic, "When comparing groups an assumption is made that the instrument measures the same psychological construct in all groups. If this assumption holds, the comparisons are valid and differences/similarities between groups can be meaningfully interpreted. If this assumption does not hold, comparisons and interpretations are not fully meaningful. The establishment of measurement invariance is a prerequisite for meaningful comparisons across groups." (Milfont & Fischer, 2010).

There are three main levels of invariance testing: configural (underlying factor model identical across groups), metric (item loadings identical across groups), and interval (item intercepts identical across groups). The specialised literature argues that metric invariance is needed for examining associations between constructs across groups, and that mean comparisons across groups are only meaningful when interval invariance is achieved. All hypotheses in the Registered Report focus on mean comparisons or associations between constructs across cultural groups but no invariance testing is proposed. This needs to be addressed! Without testing and confirming the multi-item measures are equivalent across the groups, the testing of all proposed hypotheses are meaningless. For example, if participants across the considered nations do not respond to items of the Perceived Stress Scale similarly, then comparing stress means across countries (H1a) can't be meaningfully interpreted.

When revising the Registered Report to include measurement invariance, I suggest the authors to explore the alignment method, which is a much easier implementation of invariance testing – particularly for large groups as in this case. I believe alignment can already be implemented in R. I recommend two main publications on the topic: Asparouhov and Muthén (2014) and Marsh et al. (2018); and we have also implemented this approach in a recent article (see Milfont et al. 2018) and have discussed measurement invariance for replicating findings across cultural groups (see Milfont & Klein, 2018). I am sure the authors would know measurement invariance experts in their networks but I can make myself available to provide some guidance if needed (and if the Editor and authors feels this is appropriate).

Issues with Hypotheses

Beyond the main issue regarding measurement invariance noted above, I noted issues with the proposed hypotheses.

1. I feel the authors should make it explicit from the outset that they will employ a multilevel approach.
2. The authors propose a list of country-level variables and use them as control. This is a reasonable list of variables but there is no justification of their selection. This list can be much longer, especially if the authors decide to include psychological dimensions of cultural variability such as individualism/collectivism, tightness/looseness and relational mobility. It is thus important to provide some rationale for selecting only non-psychological country-level variables and for selecting these specific ones.
3. Related to the point above, the country-level variables are merely used as control variables in the analyses. It would be great if these were included as predictors, at least for the first four hypotheses. Once invariance is established and meaningful comparisons can be made (see above), merely showing variability in stress, concern, trust and compliance might not tell much, or might be difficult for the readers and government officials to interpret the observed similarities/differences. It would be more interesting to show whether stress, concern, trust and/or compliance is greater/weaker in wealthier/poorer countries, for example. (Again, the list of variables could be very large and such analysis would not be attainable; by addressing the point above and providing justification for a small list of variables selected, the authors could provide such addition to the first hypotheses.)
4. H3b reads "Across countries, trust in government efforts to slow the spread of coronavirus will negatively predict concern over coronavirus, when controlling for individual- and country-level covariates" but this negative prediction is not obvious to me. Greater concern should lead to stronger trust, no? Or alternatively, if I don't trust the government efforts (take delayed and

denialistic actions by the USA and Brazil for example), I would be more concerned about the pandemic, no?

5. I found H5 unclear. Do you mean that provisions would reduce/buffer the expected negative effect of stress on compliance (H2a)? You need to spell out. Given the relations between stress and concern (see H2a, H2b and H4), I believe another moderation hypothesis is reasonable and could be included by examining the extent to which provisions increases/enhances the expected positive effect of concern on compliance.

Minor Points

I have made comments in the original PDF. I have highlighted sentences that are unclear of miss words, and have added missing words too (e.g., provide clear direction for some of the hypothesis).

I wish you the authors good luck with the project. Congratulations again for leading such an impressive and important research project!

Taciano L Milfont

<https://orcid.org/0000-0001-6838-6307>

References

Asparouhov, T., & Muthén, B. (2014). Multiple-group factor analysis alignment. *Structural Equation Modeling: A Multidisciplinary Journal*, 21, 495–508.

Marsh, H. W., Guo, J., Parker, P. D., Nagengast, B., Asparouhov, T., Muthén, B., & Dicke, T. (2018). What to do when scalar invariance fails: The extended alignment method for multi-group factor analysis comparison of latent means across many groups. *Psychological Methods*, 23, 524–545.

Milfont, T. L., Bain, P. G., Kashima, Y., Corral-Verdugo, V., Pasquali, C., Johansson, L.-O., Guan, Y., Gouveia, V. V., Garðarsdóttir, R. B., Doron, G., Bilewicz, M., Utsugi, A., Aragones, J. I., Steg, L., Soland, M., Park, J., Otto, S., Demarque, C., Wagner, C., Madsen, O. J., Lebedeva, N., González, R., Schultz, P. W., Saiz, J. L., Kurz, T., Gifford, R., Akotia, C. S., Saviolidis, N. M., & Einarsdóttir, G. (2018). On the relation between social dominance orientation and environmentalism: A 25-nation study. *Social Psychological and Personality Science*, 9, 802–814.

Milfont, T. L., & Fischer, R. (2010). Testing measurement invariance across groups: Applications in cross-cultural research. *International Journal of Psychological Research*, 3, 112–131.

Milfont, T. L., & Klein, R. A. (2018). Replication and reproducibility in cross-cultural psychology. *Journal of Cross-Cultural Psychology*, 49, 735–750.

Reviewer: 2

Comments to the Author(s)

Please see attached.

Reviewer: 3

Comments to the Author(s)

This is a well-documented and thought-out study. The topic is highly relevant and interesting. I have no major concerns. I have some smaller points, which the authors might want to consider.

It is understandable that with a study of events as they are unfolding, it is difficult to have very specific predictions. The predictions listed by the authors are relatively general but I find this appropriate in the current context. I suspect that the descriptive statistics might be very informative in their own right.

In the comparison across countries, it might be useful to have some measure of the stage of the pandemic that each country is in at the time of the survey, as this is likely to impact the responses. For example, if the situation in a country has stabilized somewhat (at least optically) people are likely to be less anxious than people who live in countries that are in earlier stages of the pandemic.

The exclusion criteria seem reasonable, both at the country level and at the respondent level. The authors might consider including an attention check, of which there are many examples in the literature.

The proposed analyses use both frequentist and Bayesian statistics. The authors specify the manner in which they will interpret these statistics, both separately and in conjunction.

Reviewer: 4

Comments to the Author(s)

This could be a great paper! Unfortunately I did not see the survey being spread around, though I am in a very research-intensive part of the world. I'd suggest increasing its visibility across Europe.

[see attached review for detail comments]

Reviewer: 5

Comments to the Author(s)

The current paper is a rapid registered report examining associations between self-reported variables in different countries during the COVID-19 pandemic. I considered the study proposal itself along with the data/materials shared on OSF. Data collection is already in progress, and we are asked to review the study's methods (not changeable) and its analysis plan (changeable) for this RR. Overall, there are several things to like about this paper. First, in spite of having been rapidly assembled, the authors appear to be taking steps to ensure the quality of the resultant data. I have several suggestions for improvement of the project.

1. The biggest issue is that the authors do not plan to test for measurement invariance before proceeding to their focal analyses. A lack of measurement invariance can compromise the validity of conclusions. I recommend the use of multiple group models using SEM to test invariance. There are several resources I can recommend to learn about this technique if necessary: Chen (2008, JPSP) and the Kline SEM textbook are good introductions.

2. The authors state "Continuous variables are centred on grand means, or converted to a z-score for the convenience of interpretation" (p. 10). Centering is a good idea; standardizing (converting to a z-score) is a bad idea in multi-level models (because you disturb the variance structure across levels). Avoid standardizing data in multilevel analyses. Rely on conversions to quasi-r-squared (or quasi-r) to interpret effect sizes. Effect sizes should absolutely be interpreted alongside p-values.

3. The authors have clear associations that they plan to test. At least as the introduction is currently written, these tested associations do not appear to stem from any particular theory or

model. I don't necessarily have an issue with this, but the tested models are not confirmatory in the sense that they test a theory-driven hypothesis. The introduction could benefit from a model or framework if there is one. I.e., why test these associations and not other models?

4. The authors intend to model a variety of demographic country level variables. I wonder if there are other pandemic-specific country level variables that should be included – e.g., rate of country-level hospital bed or protective equipment shortages, date of lockdown onset (or presence of lockdown), previous pandemic incidence. Further, I wonder if time of data collection should be explicitly modeled (e.g., days from date of survey to onset of lockdown). I don't have strong feelings about this, but wanted to make the suggestion.

5. My preference would be to see all models with and without controls included. The authors could present their preferred specification (as presented here) and contrast it with models without level 1 and level 2 controls.

6. Hypotheses 1a to 1d should be tested with multi-level models (not ANOVAs). The reason is that country should be a random intercept, not a fixed effect. Hypotheses 2-5 should include and test random slopes for the predicted effects. For all hypotheses, I recommend a series of nested models: a null model (no predictors, no random effects), a random intercept only model (country as random intercept), a model with fixed effects of predictors, a model adding control variables, a model with random slopes for predictors. Each model can be compared to its predecessor via a likelihood ratio test. Strive to specify maximal models, but trim random effects (usually random slopes) if there are convergence issues.

7. Plan to report descriptive statistics for all variables (means, standard deviations, alpha or omega reliability, intercorrelations between variables). I don't know these variables well enough to anticipate whether there might be problems with multi-collinearity of predictors in the planned models. Anticipate and plan for dealing with multi-collinearity if necessary.

8. It's not clear in the proposal how country level variables will be scaled. That is, will they be treated as continuous? Contrast coded categories? For individual level predictors, I do wonder if the authors should plan to collapse educational categories (perhaps to just three: did not finish compulsory schooling in that country; finished secondary schooling; university degree or higher). If you choose to do this, I would contrast code the categories rather than treat them as continuous.

9. It's not clear why for hypothesis 1c, the focus is on OECD_Institutions item 1, rather than all OECD_Institutions items. Should these items be combined into a total score if possible? Either way, would be good to have more reasoning behind this choice.

10. Some variables are measured with single items that may have limited reliability/validity. This is unavoidable given that we cannot alter the design at this point. Just be aware that this may further compromise statistical power of some analyses.

11. Regarding power, the authors state "we plan to recruit at least 30 participants per country so as to detect both the effects of individual- and country-level predictors" (p. 9). They cite Arend and Schäfer (2019)'s Table 8 as justification. They note that in the ESS database, medium ICCs, small to medium effects of level 1 and level 2 predictors, and small cross-level interactions are observed. They expect to collect data from 44 countries (level 2 units). The level 1 sample size (individuals per country) could vary. Of the 10 hypotheses, four focus on the ICCs themselves, five focus on level 1 predictors, and one contains an interaction between two level 1 variables. In six of the hypotheses, level 2 control variables are included. Based on Arend and Schäfer (2019)'s Table 8, with 44 countries and a sample size of 30 per country, small level 1 effects are

underpowered (medium and large level 1 effects are adequately powered), regardless of the ICC. Only large effects of level 2 variables are adequately powered (regardless of ICC); small and medium effects are underpowered. Only large cross-level interactions under conditions of large random slopes are adequately powered; most random slopes are underpowered. Adding additional participants per country seems to matter less than adding more countries (or more broadly, level 2 units). Thus, it seems unavoidable that the current attempt will be underpowered in some ways, mostly because there is likely not an easy way to recruit more countries into the study. Having 30 participants per country seems reasonable based on the power tables, but the authors may run into issues with convergence with this size of sample per country. In particular, the multi-group invariance models tend to do better with larger per group sizes. I don't see a strong reason to cut data collection short; it would seem prudent to get as much data as possible while the pandemic is happening. The results of the study are not themselves time sensitive (no government is going to base decision making on what these results show). Therefore, I would err on the side of sampling strategy that allows the collection of as much data as possible.

12. What is the rationale for the 15% cutoff for identifying floor/ceiling effects? Why not 10 or 20? Seems arbitrary. The exclusion rules for fast response time and Mahalanobis distance seemed appropriate and a good practice to improve data quality.

13. Causal language: There is causal language in the title of the paper ("the impact of") and sprinkled throughout the paper itself. Causal language is inappropriate for this study's methods.

Signed,
Katherine S. Corker

Author's Response to Decision Letter for (RSOS-200589.R0)

See Appendix C.

RSOS-200589.R1 (Revision)

Review form: Reviewer 1

Do you have any ethical concerns with this paper?

Yes

Recommendation?

Accept in principle

Comments to the Author(s)

I congratulate the authors for undertaking and providing a solid revision of the original manuscript. I have only minor suggestions that could be easily completed when uploading the final file

1. The new section on measurement invariance testing is very good. The authors elected to follow a traditional multigroup confirmatory factor analysis approach, but practical applications with cross-cultural data indicate it is hard to achieve scalar invariance with complex measures and

several groups, which is the case in this planned study. For this reason, I recommend the authors to mention the alignment method as an alternative if scalar invariance is not achieved (this comparison article might be informative:

<https://journals.sagepub.com/doi/full/10.1177/0022022117697844>). Or the authors could mention the new approach proposed by Marsh et al. (2018; the alignment-within-CFA approach) that combines both: <https://www.ncbi.nlm.nih.gov/pubmed/28080078>.

2. I recommend mentioning the measurement invariance tests briefly in the analytical plan (section 10).

3. There is a recently published review piece the authors should cite:

<https://www.nature.com/articles/s41562-020-0884-z>. We have also a COVID-19 manuscript under the second round of revisions in *American Psychologist* the authors could consider citing; the pre-print is available here: <https://psyarxiv.com/cx6qa>

4. I have made small suggestions in the PDF file (Appendix D), uploaded to the system. Once again I wish you the authors good luck with the project!

Taciano L Milfont

<https://orcid.org/0000-0001-6838-6307>

Review form: Reviewer 2

Do you have any ethical concerns with this paper?

No

Recommendation?

Accept in principle

Comments to the Author(s)

I reviewed RSOS-200589 (Stage 1 RR, Revision 1). My concerns from the first round are sufficiently addressed. I think the fit is slightly awkward for an RR because some of the proposed analyses are partially unspecified and data collection has already begun. RRs are easier to evaluate when they focus on known confirmatory tests. But, I think the rapid review and revision process have served this project well, it sounds like the quantitative requests from the other authors don't require changes to the data collection, and I'd support moving to Stage 2 when the reviewer consensus allows.

One weakness of the current draft, although not a condition for acceptance for me, is how the sampling is oversold, e.g., "...leverage fast organic sampling, which favours a varied reach over predetermined population criteria" and in the author response: "organic sampling rather than e.g. purchased responses from predetermined subject pools". These sentences are more marketing copy than clear scientific writing, particularly "organic". I assume the sampling strategy is mostly due to convenience (lower cost), and that is okay and even okay to write. Beyond that, if there are advantages to the sampling strategy, then please make them explicit and stated clearly in separate sentences. For example, two implied problems are participants being paid and participants being fatigued. Please justify how either might influence the specific hypotheses here before claiming that the other strategy is better. Otherwise, I assume the loss of population representativeness and therefore generalizability compared to paid, national samples is the main difference between strategies. This can also be stated directly (Discussion would be okay).

Next, “Colleagues from the consortium hope that causal relationships can be studied further down the line, as longitudinal data becomes available from the dataset.” This reflects a lack of planning. It is possible to design those tests now. This is not a condition for acceptance.

Also, “We are experiencing an unprecedented outbreak of a new pandemic.” might not read as well in 2022 or 2023. I suggest revision.

Use spellcheck again, e.g., “complex.Worry”

Thanks to the authors for redrafting their article and pointing out their rationale on other points in the author letter.

Review form: Reviewer 3

Do you have any ethical concerns with this paper?

No

Recommendation?

Accept in principle

Comments to the Author(s)

I am happy with the way the authors have taken on board the reviewers' concerns and suggestions. I therefore recommend in principle acceptance. I look forward to seeing the results.

Review form: Reviewer 5 (Katherine S. Corker)

Do you have any ethical concerns with this paper?

No

Recommendation?

Accept with minor revision

Comments to the Author(s)

Summary: Revisions are needed regarding my previous points 1, 2, and 6.

Point 1: Corona concerns also needs to be tested for invariance, in a similar way as perceived stress. The five items are implicitly assumed to reflect one underlying factor. SPS is also a multi-item scale that should be tested in this way.

Point 2: I did not understand the authors' response to this concern. I no longer find any mention of centering in the manuscript, and the noted response in Section 6 does not appear to be there. I believe this concern stands as previously noted.

Point 3: This concern is addressed.

Point 4: I disagree that severity of infection in the country in question would not influence individual behavior. It seems obvious that people would change their behavior more if the

perceived threat is higher. Nonetheless, the authors are free to disagree, so this concern is addressed.

Point 5: This concern is addressed.

Point 6: This is not an optional change. The model is misspecified if the authors do not include a random intercept for country. My recommendation remains to test the models using the nested models previously described. Post hoc between country comparisons can still be done if there are significant random intercepts. Perhaps it would help the authors conceptualize it to think of the analysis as a mixed ANOVA rather than a multi-level model (although these are equivalent statistically).

Point 7: This concern is addressed.

Point 8: This concern is addressed.

Point 9: This concern is addressed.

Point 10: This concern is addressed.

Point 11: The authors do not appear to have modified their sampling strategy and intend to focus only on the data collected through April 6. I continue to think they should use all available data at the time of Stage 1 RR acceptance.

Point 12: This concern is addressed.

Point 13: This concern is addressed.

Decision letter (RSOS-200589.R1)

Dear Dr Lieberoth,

On behalf of the Editors, I am pleased to inform you that your Manuscript RSOS-200589.R1 entitled "Trust and compliance with preventive measures in the 2020 coronavirus pandemic: relationships to trust, stress and worry" has been accepted in principle for publication in Royal Society Open Science subject to minor revision in accordance with the referee and editor suggestions. Please find their comments at the end of this email.

The reviewers and handling editors have recommended publication, but also suggest some minor revisions to your manuscript. Therefore, I invite you to respond to the comments and revise your manuscript.

When submitting your revised manuscript, you will be able to respond to the comments made by the referees and you should upload a file "Response to Referees". You can use this to document any changes you make to the original manuscript. In order to expedite the processing of the revised manuscript, please be as specific as possible in your response to the referees.

Full author guidelines can be found here <https://royalsocietypublishing.org/rsos/registered-reports>.

on behalf of Chris Chambers (Subject Editor, Royal Society Open Science)
openscience@royalsociety.org

Associate Editor Comments to Author (Professor Chris Chambers):

Associate Editor: 1

Comments to the Author:

The revised manuscript was returned to four of the original reviewers. The good news is that the manuscript is now substantially closer to IPA, however some there remain some wrinkles to iron out, especially in addressing the points raised by Reviewer 5 (including the concerns in point 1, 2, 6 and 11, all of which must be satisfied). Reviewer 1 and 2 also offer a range of constructive suggestions for improving clarity, avoiding unnecessary spin, and citing additional literature.

Reviewer comments to Author:

Reviewer: 1

Comments to the Author(s)

I congratulate the authors for undertaking and providing a solid revision of the original manuscript. I have only minor suggestions that could be easily completed when uploading the final file

1. The new section on measurement invariance testing is very good. The authors elected to follow a traditional multigroup confirmatory factor analysis approach, but practical applications with cross-cultural data indicate it is hard to achieve scalar invariance with complex measures and several groups, which is the case in this planned study. For this reason, I recommend the authors to mention the alignment method as an alternative if scalar invariance is not achieved (this comparison article might be informative:

<https://journals.sagepub.com/doi/full/10.1177/0022022117697844>). Or the authors could mention the new approach proposed by Marsh et al. (2018; the alignment-within-CFA approach) that combines both: <https://www.ncbi.nlm.nih.gov/pubmed/28080078>.

2. I recommend mentioning the measurement invariance tests briefly in the analytical plan (section 10).

3. There is a recently published review piece the authors should cite:
<https://www.nature.com/articles/s41562-020-0884-z>. We have also a COVID-19 manuscript under the second round of revisions in *American Psychologist* the authors could consider citing; the pre-print is available here: <https://psyarxiv.com/cx6qa>
4. I have made small suggestions in the PDF file, uploaded to the system.
 Once again I wish you the authors good luck with the project!

Taciano L Milfont
<https://orcid.org/0000-0001-6838-6307>

Reviewer: 2

Comments to the Author(s)

I reviewed RSOS-200589 (Stage 1 RR, Revision 1). My concerns from the first round are sufficiently addressed. I think the fit is slightly awkward for an RR because some of the proposed analyses are partially unspecified and data collection has already begun. RRs are easier to evaluate when they focus on known confirmatory tests. But, I think the rapid review and revision process have served this project well, it sounds like the quantitative requests from the other authors don't require changes to the data collection, and I'd support moving to Stage 2 when the reviewer consensus allows.

One weakness of the current draft, although not a condition for acceptance for me, is how the sampling is oversold, e.g., "...leverage fast organic sampling, which favours a varied reach over predetermined population criteria" and in the author response: "organic sampling rather than e.g. purchased responses from predetermined subject pools". These sentences are more marketing copy than clear scientific writing, particularly "organic". I assume the sampling strategy is mostly due to convenience (lower cost), and that is okay and even okay to write. Beyond that, if there are advantages to the sampling strategy, then please make them explicit and stated clearly in separate sentences. For example, two implied problems are participants being paid and participants being fatigued. Please justify how either might influence the specific hypotheses here before claiming that the other strategy is better. Otherwise, I assume the loss of population representativeness and therefore generalizability compared to paid, national samples is the main difference between strategies. This can also be stated directly (Discussion would be okay).

Next, "Colleagues from the consortium hope that causal relationships can be studied further down the line, as longitudinal data becomes available from the dataset." This reflects a lack of planning. It is possible to design those tests now. This is not a condition for acceptance.

Also, "We are experiencing an unprecedented outbreak of a new pandemic." might not read as well in 2022 or 2023. I suggest revision.

Use spellcheck again, e.g., "complex.Worry"

Thanks to the authors for redrafting their article and pointing out their rationale on other points in the author letter.

Reviewer: 3

Comments to the Author(s)

I am happy with the way the authors have taken on board the reviewers' concerns and suggestions. I therefore recommend in principle acceptance. I look forward to seeing the results.

Reviewer: 5

Comments to the Author(s)

Summary: Revisions are needed regarding my previous points 1, 2, and 6.

Point 1: Corona concerns also needs to be tested for invariance, in a similar way as perceived stress. The five items are implicitly assumed to reflect one underlying factor. SPS is also a multi-item scale that should be tested in this way.

Point 2: I did not understand the authors' response to this concern. I no longer find any mention of centering in the manuscript, and the noted response in Section 6 does not appear to be there. I believe this concern stands as previously noted.

Point 3: This concern is addressed.

Point 4: I disagree that severity of infection in the country in question would not influence individual behavior. It seems obvious that people would change their behavior more if the perceived threat is higher. Nonetheless, the authors are free to disagree, so this concern is addressed.

Point 5: This concern is addressed.

Point 6: This is not an optional change. The model is misspecified if the authors do not include a random intercept for country. My recommendation remains to test the models using the nested models previously described. Post hoc between country comparisons can still be done if there are significant random intercepts. Perhaps it would help the authors conceptualize it to think of the analysis as a mixed ANOVA rather than a multi-level model (although these are equivalent statistically).

Point 7: This concern is addressed.

Point 8: This concern is addressed.

Point 9: This concern is addressed.

Point 10: This concern is addressed.

Point 11: The authors do not appear to have modified their sampling strategy and intend to focus only on the data collected through April 6. I continue to think they should use all available data at the time of Stage 1 RR acceptance.

Point 12: This concern is addressed.

Point 13: This concern is addressed.

Author's Response to Decision Letter for (RSOS-200589.R1)

See Appendix E.

RSOS-200589.R2 (Revision)

Review form: Reviewer 5 (Katherine S. Corker)

Do you have any ethical concerns with this paper?

No

Recommendation?

Accept in principle

Comments to the Author(s)

NA

Decision letter (RSOS-200589.R2)

Dear Dr Lieberoth

On behalf of the Editor, I am pleased to inform you that your Manuscript RSOS-200589.R2 entitled "Stress and worry in the 2020 coronavirus pandemic: Relationships to trust and compliance with preventive measures across 45* countries" has been accepted in principle for publication in Royal Society Open Science.

Your revised manuscript was assessed by one of the original reviewers in consultation with the Registered Reports editor, whose comments you can find below.

Please read the following email carefully

Your accepted Stage 1 manuscript has been publicly registered at:

<https://doi.org/10.17605/OSF.IO/YTBCS>

Following completion of your study, we invite you to resubmit your paper for peer review as a Stage 2 Registered Report. Please note that your manuscript can still be rejected for publication at Stage 2 if the editors consider any of the following conditions to be met:

- The results were unable to test the authors' proposed hypotheses by failing to meet any approved outcome-neutral criteria.
- The authors altered the Introduction, rationale, or hypotheses, as approved in the Stage 1 submission.
- The authors failed to adhere closely to the registered experimental procedures. Please note that any deviations from the approved experimental procedures must be communicated to the editor immediately for approval, and prior to the completion of data collection. Failure to do so can result in revocation of in-principle acceptance and rejection at Stage 2 (see complete guidelines for further information).

- Any post-hoc (unregistered) analyses were either unjustified, insufficiently caveated, or overly dominant in shaping the authors' conclusions.
- The authors' conclusions were not justified given the data obtained.

We encourage you to read the complete guidelines for authors concerning Stage 2 submissions at <https://royalsocietypublishing.org/rsos/registered-reports#ReviewerGuideRegRep>. Please especially note the requirements for data sharing, reporting the URL of the independently registered protocol, and that withdrawing your manuscript will result in publication of a Withdrawn Registration.

Once again, thank you for submitting your manuscript to Royal Society Open Science and we look forward to receiving your Stage 2 submission. If you have any questions at all, please do not hesitate to get in touch. We look forward to hearing from you shortly with the anticipated submission date for your stage two manuscript.

on behalf of Professor Chris Chambers (Registered Reports Editor, Royal Society Open Science)
openscience@royalsociety.org

Associate Editor Comments to Author (Professor Chris Chambers):

Associate Editor: 1

Comments to the Author:

I have now consulted again with Reviewer 5 and read the manuscript again myself, and I want to commend the authors for responding so comprehensively and constructively to all the reviews. Concerning the one remaining issue (point 11 raised by Reviewer 5), there is no obvious a priori solution but I agree with the reviewer that including more data is preferable for the preregistered analyses. That said, I also recognise the potential value (and potential increase in sensitivity) in the authors' proposed alternative. I am therefore issuing Stage 1 IPA for the latest manuscript as submitted, and I would encourage the authors to include additional post hoc analyses at Stage 2 using a limited temporal window (or windows), and/or including time as a variable of interest as previously suggested by Reviewer 5. There is no need to preregister these additional analyses as they can be included at Stage 2 as exploratory analyses (e.g. as robustness or sensitivity checks).

Author's Response to Decision Letter for (RSOS-200589.R2)

See Appendix F.

RSOS-200589.R3 (Revision)

Review form: Reviewer 2

Is the manuscript scientifically sound in its present form?

Yes

Are the interpretations and conclusions justified by the results?

Yes

Is the language acceptable?

Yes

Do you have any ethical concerns with this paper?

No

Have you any concerns about statistical analyses in this paper?

No

Recommendation?

Major revision

Comments to the Author(s)

See attached (Appendix G).

Decision letter (RSOS-200589.R3)

This year has been very difficult for everyone, and we want to take the opportunity to thank you for your continued support in 2020.

The Royal Society Open Science editorial office will be closed from the evening of Friday 18 December 2020 until Monday 4 January 2021. We will not be responding during this time. If you have received a deadline within this time period, please contact us as soon as possible to allow us to extend the deadline. If you receive any automated messages during this time asking you to meet a deadline, we offer apologies and invite you to respond after the festive period or during normal working hours.

With our best for a peaceful festive period and New Year, and we look forward to working with you in 2021.

Dear Dr Lieberoth:

On behalf of the Editor, I am pleased to inform you that your Stage 2 Registered Report RSOS-200589.R3 entitled "Stress and worry in the 2020 coronavirus pandemic: Relationships to trust and compliance with preventive measures across 48 countries" has been deemed suitable for publication in Royal Society Open Science subject to minor revision in accordance with the referee suggestions. Please find the referees' comments at the end of this email.

The reviewers and Subject Editor have recommended publication, but also suggest some minor revisions to your manuscript. Therefore, I invite you to respond to the comments and revise your manuscript.

Please also ensure that all the below editorial sections are included where appropriate -- if any section is not applicable to your manuscript, please can we ask you to nevertheless include the heading, but explicitly state that the heading is inapplicable. An example of these sections is attached with this email.

- Ethics statement

- Data accessibility

[http://datadryad.org/submit?journalID=RSOS&manu=\(Document not available\)](http://datadryad.org/submit?journalID=RSOS&manu=(Document not available))

- Competing interests

- Authors' contributions

- Acknowledgements

- Funding statement

Because the schedule for publication is very tight, it is a condition of publication that you submit the revised version of your manuscript within 30 days. If you do not think you will be able to meet this date please let me know immediately.

on behalf of Professor Chris Chambers
(Registered Reports Editor, Royal Society Open Science)
openscience@royalsociety.org

Associate Editor Comments to Author (Professor Chris Chambers):

Associate Editor: 1

Comments to the Author:

One of the original Stage 1 reviewers was available for re-review, likely reflecting the intense pressures on academics as this extraordinary year draws to a close. Based on the intensity of the

Stage 1 review process, the high quality of the Stage 2 review provided, and my own assessment of the Stage 2 manuscript, I have decided to proceed with an editorial decision on the basis of this single review.

As you will see, the reviewer raises a number of significant issues to address in finalising the Stage 2 manuscript, ranging from deviations from protocol, to clarity in the presentation (and OSF archiving) of the data/results, to the quality of synthesis and contextualisation in the Discussion. Concerning the reviewer's comment that "the results section would benefit from a summary table of the hypotheses and whether they were met" -- I agree and please either add such a table (abbreviated form of Table 3) or augment the existing Table 3 to include the results in an additional column (which would require moving Table 3 to the main text).

Comments to Author:

Reviewer: 2

Comments to the Author(s)

see attached

Author's Response to Decision Letter for (RSOS-200589.R3)

See Appendix H.

Decision letter (RSOS-200589.R4)

Dear Dr Lieberoth:

It is a pleasure to accept your Stage 2 Registered Report entitled "Stress and worry in the 2020 coronavirus pandemic: Relationships to trust and compliance with preventive measures across 48 countries" in its current form for publication in Royal Society Open Science.

COVID-19 rapid publication process:

We are taking steps to expedite the publication of research relevant to the pandemic. If you wish, you can opt to have your paper published as soon as it is ready, rather than waiting for it to be published the scheduled Wednesday.

This means your paper will not be included in the weekly media round-up which the Society sends to journalists ahead of publication. However, it will still appear in the COVID-19 Publishing Collection which journalists will be directed to each week (<https://royalsocietypublishing.org/topic/special-collections/novel-coronavirus-outbreak>).

If you wish to have your paper considered for immediate publication, or to discuss further, please notify openscience_proofs@royalsociety.org and press@royalsociety.org when you respond to this email.

You can expect to receive a proof of your article in the near future. Please contact the editorial office (openscience@royalsociety.org) and the production office (openscience_proofs@royalsociety.org) to let us know if you are likely to be away from e-mail contact – if you are going to be away, please nominate a co-author (if available) to manage the proofing process, and ensure they are copied into your email to the journal.

Best regards,

on behalf of Professor Chris Chambers (Subject Editor)
openscience@royalsociety.org

Appendix A**ROYAL SOCIETY
OPEN SCIENCE****The impact of stress and worry from the coronavirus
pandemic on trust and compliance across 45 countries- data
from the COVIDiSTRESS global survey**

Journal:	Royal Society Open Science
Manuscript ID	RSOS-200589
Article Type:	Registered Report - Stage 1
Date Submitted by the Author:	07-Apr-2020
Complete List of Authors:	Lieberoth, Andreas; Aarhus University, School of Culture and Society (Interacting Minds Center); Aarhus University, Danish School of Education (DPU) Han, Hyemin; University of Alabama Lin, Shiang-Yi; Hong Kong Institute of Education Stöckli, Sabrina; University of Bern
Subject:	health and disease and epidemiology < BIOLOGY, psychology < BIOLOGY
Keywords:	COVID-19, preventive measures, worry, stress, compliance behavior
Subject Category:	Psychology and cognitive neuroscience

Author-supplied statements

Relevant information will appear here if provided.

Ethics

Does your article include research that required ethical approval or permits?:

Yes

Statement (if applicable):

Due to the pressing nature of COVID-19 research, a waiver to proceed has been acquired from the Aarhus University board of research ethics office, stating that the study is in accordance with the IRBs standards for acceptable research in human subjects.

To further ensure the participant's interests and compliance with GDPR-standards, all data are anonymous, and no medical information is collected.

Data

It is a condition of publication that data, code and materials supporting your paper are made publicly available. Does your paper present new data?:

Yes

Statement (if applicable):

The COVIDiSTRESS global survey is a collaborative open science project. The data used here, and all other data including raw data and advice on cleaning/coding is made available at the Open Science Framework <https://osf.io/z39us/>

Conflict of interest

I/We declare we have no competing interests

Statement (if applicable):

CUST_STATE_CONFLICT :No data available.

Authors' contributions

This paper has multiple authors and our individual contributions were as below

Statement (if applicable):

The COVIDiSTRESS global survey is a large consortium. Due to the time sensitive nature of the RSOS COVID-19 rapid response initiative, we will specify the contributions of all participants in the data collection and analysis, if and when acceptance in principle is received

See the full list of participants as co-signees in the cover letter.

***The impact of stress and worry from the coronavirus pandemic on trust,***
***behaviour and compliance with preventive measures across 45* countries -***
***data from the COVIDiSTRESS global survey***

**STAGE ONE REGISTERED REPORT**
**Structured using the template for the COVID-19 rapid response**

This Registered Report under the COVID-19 rapid response initiative presents the
COVIDiSTRESS global survey, which collects open data on psychological and behavioural
responses to the COVID-19 pandemic (caused by the coronavirus SARS-CoV-2) in 45*
countries. This collaborative study was co-designed by researchers from numerous
universities across the world to investigate how psychological and behavioural responses to
the COVID-19 pandemic differ across countries and cultures, and how this has impacted
social behaviour and trust in government efforts to slow the spread of the coronavirus.
Contrary to paid national studies, COVIDiSTRESS aims not at generating nationally
representative data, but collecting large amounts of human experiences for rapid access to the
research community.

The present analysis of COVIDiSTRESS data is of pressing importance during the COVID-
19 crisis, **in order to understand how different government strategies succeed or fail.** By
generating data rapidly, and disseminating them continuously as open data on the Open
Science Framework (OSF), the international research network behind the COVIDiSTRESS
project aims to yield insights for the use of practitioners such as health communicators,
government officials and policy-makers at all levels as well as members of the academic
community. **The present RR is the first detailed test of hypotheses with the data.**

[*RR stage one note to reviewers: We will update this projected number based on final number of
valid responses according to inclusion/exclusion criteria.
This RR concerns data analysis for central variables from the COVIDiSTRESS global survey, as data
collection has already commenced]

**1. Background and Research Questions**

1. What is the impact of psychological stress on trust in and compliance with efforts to prevent
COVID-19 when responses are compared across countries?
- 2. To what degree is psychological stress related to concerns over COVID-19 itself, and what
proportion of the variance could be explained by other factors?

This Registered Report analyses data from a swiftly executed global survey study, containing a
number of variables meant to allow both hypothesis testing and exploratory cross-country
comparisons. As an open science project, the COVIDiSTRESS effort as a whole intends to supply
data to the public, allowing researchers and other stakeholders to independently conduct their own

analyses. As such, the present Registered Report investigates 10 central hypotheses, focusing on
peoples' experiences and behaviours during the COVID-19 crisis.

We are experiencing an unprecedented outbreak of a new pandemic. To control the spread of the
coronavirus, governments are imposing a range of measures, which include but are not restricted to
quarantines and limits to civil liberties. Inevitably, these changes have generated a variety of
psychological responses which in turn shape the level of compliance with preventive efforts. In fact,
extant research on the factors that shape willingness to comply with such public health efforts aimed
to prevent or slow the spread of epidemics has highlighted the importance of psychological responses
(Taylor et al., 2009; Leung et al. 2003) and trust in the state authorities (Seligman, 1997; Capelos et
al., 2016) in compliance with the guidelines and restrictions. Yet, even though their impact is
projected to be significant (Brooks et al., 2020), the consequences of these varied psychological
responses and differing levels of trust on compliance with measures imposed by different
governments is not yet known.

In addition to their effects on public health, pandemic outbreaks breed misinformation and fear of
contagion as well as uncertainty and panic in the course of their spread (Khan & Huremovic, 2019).
Both concern for the severity of a disease and the perceived reliability of a government's information
and belief in the efficacy of preventive measures in turn can influence the intention to comply and
self-reported preventive behaviour (Bults et al., 2011). Thus, the extent of compliance is influenced
by the level of trust in one's sources of information about a pandemic, as well as the perceived gravity
of the disease. Indeed, communicating about the necessary measures to prevent the spread of a
pandemic such as COVID-19 may be most effective when a population feels "optimistic anxiety"
(Petersen, 2020)—appropriate concern over the disease but also enough hope that one believes that
acting in accordance with the preventive measures suggested by the authorities will make a
difference. Indeed, high risk perception in combination with moderate anxiety can enable compliance
with protective measures against a disease (e.g. in SARS, Leung et al., 2003). However, concern over
one's perceived risk during a pandemic can also be a source of ongoing worry and anxiety as well as
stress (e.g. in H1N1, Bults et al., 2011; in MERS, Ro et al., 2017), and these effects can be
exacerbated by the fact that even some of the most efficient methods of slowing down the spread of
the disease, such as self-isolation and quarantine, also take an immediate and potentially long-lasting
psychological toll among affected populations (Huremovic, 2019; Maunder, 2009; Brooks et al.,
2020). Worry and other situational stressors thus seem to interact, and together influence behavior
during pandemics. Surveys of quarantined healthcare workers (Reynolds et al., 2008) and members of
the general population (Hawryluck et al., 2004) following the 2003 SARS outbreak found evidence of
both acute stress responses and post-traumatic stress disorder, especially in cases with a longer
duration of quarantine and greater perceived difficulty in compliance. This as well as confusion over
an appropriate quarantine protocol due to the lack of timely information and resources from public
health systems (Cava et al., 2005) may have contributed to lower levels of compliance. The
perception of openness and reliability of governments and health organisations (Bish & Michie,
2010), levels of trust in media and medical authorities (Prati et al. 2011; Gilles et al. 2011) as well as
perceptions of disease's severity and the efficacy of one's actions (Bish & Michie, 2010; Fung &
Cairncross, 2007; Lau et al., 2004) play a role in compliance with recommendations for preventive
behaviour.

As such, both the medical situation and the psychological effects of isolation and confinement
(Oliver, 1991; Palinkas & Suedfeld, 2008) need to be taken into consideration when prolonged
periods of quarantine are implemented. A subset of negative effects colloquially summarised under
"cabin fever" includes responses varying from anxiety and depression (Kehoe & Abbot, 1975) to
impaired cognitive ability and hostility (Gunderson, 1974, Palinkas & Suedfeld, 2008). Efforts such
as closing down schools and workplaces, and calls for people to self-isolate in their homes, are likely
to constitute a source of both existential and practical stress unrelated to the fear of contacting the
disease such measures are put in place to stop. For instance, stress is known to result from inability to
participate in work (Lynch, 1999) or interpersonal problems in the family (ibid.; Neighbours, 1997).
Compliance with medical guidelines has been shown to decrease not just as a result of higher stress
levels (Karvinen, Murray, Arastu, & Allison, 2013), but also of minor everyday stressors such as

workplace conflict or household responsibilities (Hitchcock, Brantley, Jones, & McKnight, 1992). Prolonged states of emergency and the chronic psychological, social and economic stressors related to them (Huremovic, 2019, Brooks et al, 2020) may decrease compliance with set behavioural objectives during pandemics. Further, social support from groups such as one's family, friends, and colleagues moderate the effect of concern for the disease or other sources of stress on one's psychological well-being (Mak et al., 2009; Wang et al., 2014). According to the stress-buffering hypothesis (Cohen, 2004), this social support may buffer people from other negative consequences of distressing events. For example, such support could promote less threatening interpretations of stressful situations (ibid.; Cohen & Wills, 1985), and thus possibly more positive appraisals of institutional and governmental efforts to stop the spread of a pandemic. Sources of stress and worry may thus have direct implications not simply for well-being, but also for the effectiveness of collective efforts and the degree of public compliance with self-isolation measures.

So far, little is known about the differences in reactions across countries to preventive measures during a pandemic, but as cross-cultural differences have been observed in coping with and responding to stressors (Kuo, 2011; Wahid, 2017) and in acceptance of various countermeasures against pandemic outbreaks (Bennett & Carney, 2010), optimal institutional responses to the COVID-19 crisis may require culturally and situationally appropriate public health interventions both to alleviate human distress and to communicate preventive measures in an effective manner. Understanding the psychological and behavioural implications of human experiences of the current pandemic in various countries will help us identify the psychological risk factors for negative outcomes under the conditions of a pandemic and aid governments and other organisations devise effective interventions structured around the stressful consequences of enacting self-isolation measures.

2. Methods

A global survey study is currently being conducted with the purpose of mapping the psychological impact of COVID-19 across countries, and its relationship to behavior and attitudes to preventive measures while the virus is still spreading. As data collection is urgent, this stage one Registered Report is submitted as the survey has already been launched, and the first wave of data extraction has been completed. Although the first data extraction has been performed, no data analysis has been initiated as of yet by our group.

The COVIDiSTRESS survey (COVIDiSTRESS global survey network, 2020 - available and pre-registered at <https://osf.io/z39us/>) was constructed for wide and rapid data collection on human reactions to the coronavirus-situation across the globe, without a focus on necessarily creating nationally representative samples from the outset. The survey has two parts. The first part collects general demographic data including proximate effects of the COVID-19 pandemic (e.g., isolation status, first-hand experience, attenuated risk), personality traits, and self-reported variables such as loneliness and perceived stress (Cohen, Karmarck & Mermelstein, 1983), interpersonal and in institutions, daily behaviours including compliance with general and social preventive measures. The second part contains sets of more specific items related to people's experiences of distress and worry during the ongoing outbreak of coronavirus (e.g. access to amenities, loss of work, adapting work, education and social interactions to digital platforms, the social stresses of confinement with adults and children) as well as the factors people experience as soothing and positive to coping (e.g. social contact, staying informed, dedicating oneself to preparation, hobbies, religion) and the Social Provisions Scale (SPS; Steigen & Bergh, 2019). Finally, increases in media behaviours, including news-sources and social media, are reported. Participants supply this information as self-reported "agree-disagree" answers to how the range of options apply to their situations, as well as in text boxes to add other factors. Validated short versions of established measures are used whenever possible, and also where available in local languages. In order to protect participants' data and avoid sensitive

information, participants are not asked about COVID-19 symptoms or other data with direct medical implications or that would allow readers to identify them.

[*RR stage one note to reviewers: Due to the rapid nature of data collection envisioned in this project, the content of the survey cannot be changed at this stage. The use of variables from the survey can.]

Translation has been completed for 46 languages and localisations (Afrikaans, Albanian, Arabic, Bangla, Indonesian, Bosnian, Bulgarian, Chinese [Simplified and Traditional], Croatian, Czech, Danish, Dutch [Belgium, Netherlands], English, Spanish [Argentina, Colombia, Cuba, Mexico, Spain], Filipino, Finnish, French, German, Greek, Hebrew, Hindi, Hungarian, isiXhosa, isiZulu, Italian, Japanese, Korean, Lithuanian, Nepali, Persian, Polish, Portuguese [Brazil, Portugal], Romanian, Russian, Slovakian, Serbian, Turkish, Urdu, Vietnamese) with the possibility for more translations in future waves. Translations were completed by a forward translator from the original English version, and then validated through both panel and back-translation process by separate translators when possible.

Given the urgent call for COVID-19 research, the survey received a waiver to commence data collection from the IRB office at Aarhus University, Denmark. The survey and data extraction schedule was pre-registered at the Open Science Framework (COVIDiSTRESS global survey network, 2020). The first wave of data extraction took place on April 6th. Before the in principle acceptance of the current registered report, we will not initiate any data analysis

[note for stage 2: the dataset will also be submitted for publication in *Scientific Data*]

3. Key Independent and Dependent Variables

The COVIDiSTRESS global survey contains a number of variables (full survey at <https://osf.io/z39us/>, COVIDiSTRESS global survey network, 2020). This urgent call Registered Report focuses on 10 hypotheses (see section 1 and 4) and comparisons between countries. The following table gives an overview of all variables that are relevant for our hypotheses.

Variable name	Description	Measurement	Remarks
Independent and dependent variables			
[Scale_PSS10]	Perceived stress for the past week	Perceived Stress Scale (PSS-10). 10 items. 5-Point Likert Scale (1 =never, 5 = very often). Validated language versions where available. Back-translated where necessary. Mean score will be computed.	Cohen, Kamarck, & Mermelstein (1983)

[OECD_ insititutions]	Trust in country’s government and health system	6 questions: On a scale from 0-10 (0=not at all, 10=completely), how much you personally trust each of the institutions: (1) parliament/government?, (2) police, (3) civil service, (4) health system (added), (5) WHO (added), (6) government’s effort to handle coronavirus (added).	OECD guidelines on measuring institutional trust (OECD, 2017)
---	--	---

[Corona_concerns]	Concern over coronavirus	5 self-reported items to capture concerns about coronavirus consequences on a 6-Point Likert Scale (1=strongly disagree, 6=strongly agree): Concern about consequences of the coronavirus (1)...for yourself, (2)...for your family, (3)...for your close friends, (4)...for your country, (5)...for other countries across the globe. Mean score will be computed.	-
-------------------	--------------------------	---	---

[Compliance]	Overall compliance with local prevention guidelines	The item “I have done everything I could possibly do as an individual, to reduce the spread of coronavirus”, out of 6 self-reported items to capture behaviors and reactions, including compliance with local prevention guidelines on a 6-Point Likert Scale (1=strongly disagree, 6=strongly agree).	- --------------	---	--	---

Control variables (covariates)

[age]	Participants’ age	-	-
[gender]	Participants’ gender	0=male, 1=female, 2=other/would rather not say	-
[education]	Participants’ education	1=PhD / Doctorate, 2=College degree, 3=Some College or equivalent, 4=Up to 12 years of school, 5=Up to 9 years of school, 6=Up to 6 years of school, 7=None	-
[generalised_trust]	Participants’ trust in most of people in the country of residence	10-point Likert scale from 1 (not at all) to 10 (completely)	
[country]	Country of residence	List of all countries where the survey is disseminated/spread.	
[SPS]	Available social provisions in critical/distressing situations	Social Provisions Scale short form SPS-10. 10 items. 6-Point Likert Scale (1 =strongly disagree, 6 = strongly agree). Validated language versions where available. Back-translated where necessary. Mean scores will be computed.	Steigen & Bergh (2019)

[population_size]	Country data	- Population size	Country
[GDP]		- GDP per capita	demographic
[edu_attainment]		- Education attainment	variables from
[unemployment]		- Unemployment rate	OECD
[gini_coefficient]		- Income inequality (GINI coefficients)	(OECD, 2020)
		- Life expectancy	
[life_expectancy]			

[*RR stage one note to reviewers: Choice of variables is changeable. This includes country-level variables not collected with the COVIDiSTRESS global survey **itself Before** the in principle acceptance of the current registered report, we will not initiate any data analysis.]

4. Hypotheses

First, this Registered Report is intended to draw a map of differences in the psychological impact of COVID-19, and the trust in government measures and compliance with preventive guidelines in the countries represented in the first data extraction on Monday April 6 2020 (see inclusion criteria below). We expect that the *COVIDiSTRESS global survey* will show significant differences between countries.

- H1a Perceived stress will differ between countries.
- H1b Concern over the coronavirus will differ between countries.
- H1c Trust in government efforts to slow the spread of coronavirus will differ between countries.
- H1d Compliance with behavioural guidelines to slow the spread of coronavirus will differ between countries.

Second, the present analysis focuses on how both stress in general and **concerns endemic** to the COVID-19 pandemic (e.g. isolation, changes in public life and worry over the virus itself) together affect compliance with preventive guidelines and trust in government efforts intended to reduce the spread of coronavirus.

Compliance with behavioural guidelines

- H2a: Across countries, perceived stress will negatively predict compliance with behavioural guidelines to slow the spread of coronavirus, when controlling for individual-level covariates (age, gender, education, and generalised trust) and country-level covariates (population size, GDP per capita, education attainment, unemployment rate, income inequality and life expectancy).
- H2b: Across countries, concern over coronavirus will predict compliance with behavioural guidelines to slow the spread of coronavirus, when controlling for individual- and country-level covariates.

Trust in government efforts

- H3a: Across countries, trust in government efforts to slow the spread of coronavirus will negatively predict perceived stress, when controlling for individual- and country-level covariates.
- H3b: Across countries, trust in government efforts to slow the spread of coronavirus will negatively predict concern over coronavirus, when controlling for individual- and country-level covariates.

Third, we analyse the sources of stress during the global COVID-19 crisis. We assume that the direct threat and uncontrollable nature of coronavirus will be a direct stressor, but the COVIDiSTRESS survey also seeks to allow researchers to explore other sources of stress endemic to the situation in the short and long term.

- H4: Across countries, concern over coronavirus will predict perceived stress, when controlling for individual- and country-level covariates.

Lastly, we analyse the effects of social support on outcomes. While stressful situations may be unavoidable during a pandemic, the availability of support structures may promote less threatening interpretations of events as they unfold, and thus more positive appraisal of efforts to combat the coronavirus.

- H5: Across countries, availability of social provisions will negatively moderate the effect of perceived stress over coronavirus on compliance with behavioural guidelines to slow the spread of coronavirus, when controlling for individual- and country-level covariates.

Exploratory cross-country comparisons and descriptive statistics will be provided in stage two of this Registered Report 2.

5. Conditions of assignment

No direct manipulation or assignment.

6. Number of observations to be collected and rule of termination

Given that this large-scale survey will be spread by numerous researchers all over the world, we have limited control over how many respondents we will collect in total and per country. Thus, we define a set of stopping rules that are practically feasible:

Previous research confirms that statistical power in multilevel design is rather complex and greatly depends on the nested structure of the data (Arend & Schäfer, 2019). As such, a general rule of thumb (i.e., 30-30 rule; Kreft & de Leeuw, 1998) is unlikely to be applicable to all different data structures. Therefore, we sought to refer to a more specific guideline to plan sample sizes according to the estimates we acquired from pre-existing databases.

Due to a lack of previous research that could be used for reasonable power analyses, we ran a planned multilevel analysis for H5 (specified in Section 10), as there was a suitable data set at hand. We used the 2018 European Social Survey (ESS) database to estimate the effect sizes of fixed and random effects as well as intraclass coefficients (ICCs). The ESS database consists of 43,215 respondents from 23 European countries (in average 1878 respondents per country). Although the variables available in the ESS database are not identical to those in our survey, given their similarity in the underlying constructs, we used the health condition variable (a higher value indicating poorer health condition) and satisfaction about the government in the ESS data as proxies for stress, and compliance and trust in government efforts for our original variables of interest, respectively.

The ICC from the observed ESS data is .26, indicating that 26% of variability comes from between-country differences. Standardized estimates of fixed effects of the individual-level predictor (individual's health condition: $\beta = .09$; social provisions: $\beta = .21$) and its country-level predictor (averaged health condition in each country: $\beta = -.35$) are deemed as having small and medium effect sizes, respectively. Their cross-level interaction effects had only small effect sizes ($\beta_s = .03 - .10$).

Subsequently, we refer to the guideline from Arend and Schäfer (2019), which provide a fast and frugal power estimation for each combination of effect size, the value of ICC, as well as the size of random slope variance. By looking up Arend and Schäfer's Table 8 of power simulation results that

give a required sample size and group size (in our case, the number of countries) to detect such effects
with 80% statistical power (p. 17), in the present study, we plan to recruit at least **30 participants per**
**country** so as to detect both the effects of individual- and country-level predictors.

[*RR stage one note to reviewers: This can be changed based on review. Before the in principle
acceptance of the current registered report, we will not initiate any data analysis.]

11 12 **7. Inclusion criteria** 13 14

All participating researchers received the same task of starting to distribute the survey in their
language on March 30 2020 (exception: Denmark started on March 26 2020, see section 8). All
possible channels were allowed, e.g. social media, panels, e-mails to friends or organizations and use
of media contacts. Researchers were asked to register a protocol on recruiting activities. Further, the
survey itself has two calls to action to help us share the survey, providing them a suggested
standardised text and link, to spread the survey on social media.

In order to be considered for the present analysis, a country needs at least 30 respondents at the time
of extraction (i.e. April 6 2020). **In order to be considered as a valid response**
**, a respondent in the COVIDiSTRESS global survey must have reported their country of residence,**
**and submitted valid responses to the variables treated in each analysis (also see section 8).**

[*RR stage one note to reviewers: These criteria can be changed. Before the in principle acceptance of
the current registered report, we will not initiate any data analysis.]

33 **8. Exclusion criteria** 34 35

We define exclusion criteria on the country and on the respondent level.

***Country level***

Active dissemination of the survey and calls for participation are carried out via online and traditional
media platforms in Afghanistan, Argentina, Australia, Austria, Bangladesh, Belgium, Bosnia, Brazil,
Bulgaria, Canada, China, Colombia, Croatia, Czech Republic, Denmark, Finland, France, Germany,
Greece, India, Indonesia, Israel, Italy, Japan, South Korea, Lithuania, Malaysia, Mexico, Netherlands,
Pakistan, Philippines, Poland, Portugal, Russia, Serbia, Singapore, South Africa, Spain, Switzerland,
Taiwan, Turkey, United Kingdom, United States of America, and Vietnam*. The number of
participating countries may be increased in the future as researchers and others interested in sharing
the COVIDiSTRESS global survey join the project.

If fewer than 50% of all participating countries have failed to generate at least 30 valid responses by
first extraction on April 6 2020, we extend the data collection for one week, that is, we utilize the
weekly data extracted on April 13 2020. All countries that fail to generate 30 respondents by April 13
2020 will not be included in the final data set. For a justification for the benchmark of 30 respondents
52 per country see section 6 and 10 (sample size justification).

***Respondent level***

On the respondents level, the rather long survey can lead some participants to not ~~simply skipping~~
questions, but also give repetitive or unrepresentative answers (e.g., Krosnick & Alwin, 1987),
leading to misclassification of participants and responses that do not reflect real experience (Egleston,

Miller, & Meropol, 2011). We employ the following exclusion measures to guard against these
threats.

- 1. First, the predicted duration of the survey in Qualtrics is 22 minutes if all questions are answered
and free text boxes are filled out. On that basis, we exclude all responses who completed the
whole survey in less than 2 minutes and 12 seconds, equivalent to one-tenth of the estimated time.
- 2. Second, we use Mahalanobis distance to detect multivariate outliers due to random or carelessly
invalid responses (Curan, 2016; Dupuis et al. 2019). Participants with a $p < .001$ in the chi-square
test will be excluded.

[*RR stage one note to reviewers: These criteria and procedures can be changed. Before the in
principle acceptance of the current registered report, we will not initiate any data analysis.]

9. Quality checks

Regarding quality checks, we will check for floor / ceiling effects within the following scales / item
batteries: [Scale_PSS10], [OECD_institutions], [Corona_concerns], [Compliance], [SPS]. We will
evaluate floor / ceiling effects on the base of the percentage of the respondents with the minimum /
maximum scores. Therefore, we will provide the percentage (%) and n for respondents with the
minimum / maximum scores. Floor / ceiling effects will be considered as present if minimum /
maximum scores occur in 15% or more of the respondents.

[*RR stage one note to reviewers: Before the in principle acceptance of the current registered report,
we will not initiate any data analysis.]

10. Analysis plan

[*RR stage one note to reviewers: This section can be updated as per reviewers' suggestions. Before
the in principle acceptance of the current registered report, we will not initiate any data analysis.]

Data analysis

- • The analysis uses the tidyverse, BayesFactor and brms R packages
- • Multilevel models are run using SAS PROC MIXED with Restricted Maximum likelihood
(REML) and Kenward-Roger denominator degrees of freedom
- • For frequentist analyses, 95% confidence intervals and the conventional 5% significance level (p
$< .05$) are used for H0 significance testing. Continuous variables are centred on grand means, or
converted to a z-score for the convenience of interpretation. Aggregated variables for individual-
level predictors (country-mean variables) are computed and centred on grand means in order to
separate variance of country part from variance of individual part (see Hox, 2010).
- • For multilevel modeling, the significance of fixed effects is examined using Wald tests with
degrees of freedom adjusted with Kenward-Roger method; random effects are tested via
likelihood ratio tests ($-2\Delta LL$ with degrees of freedom equal to the number of new random effects
variance and covariance. Effect size for fixed effects are examined via pseudo-R² for the
proportion reduction in each variance component, along with the change in total R², i.e., the
squared correlation between actual outcome and predicted outcome by the predictive models
(Hoffman, 2015).
- • For Bayesian analyses, Bayes Factor ≥ 10 ($BF_{10} \geq 10$) is employed, which indicates the presence
of strong evidence supporting H1 against H0 for Bayesian testing in general (Han et al., 2018;
Wagenmakers et al., 2018).

Question/Hypothesis	Sampling plan (e.g. power analysis)	Analysis	Interpretation given different outcomes
H1a Perceived stress will differ between countries	Given that the analysis that requires the greatest sample size is the planned MLM (see H5) due to its complexity, we intend to follow the sample size estimation for the MLM (see section 6 for further details). In addition, we intend to examine the resultant Bayes Factor ≥ 10 to see whether the corrected evidence is sufficient for hypothesis testing.	We will perform one-way ANOVA to examine the international differences. DV: Perceived stress ([Scale_PSS10]) IV: Country ([country]) IF $p < .05$ is reported, THEN we will perform a post-hoc test with Scheffe's method for exploration. In addition, we will perform the same one-way ANOVA with Bayesian inference with anovaBF function implemented in BayesFactor R package. We will use the non-informative Cauchy prior ($d = .00$, scale = .707) that was proposed by Wagenmakers et al. (2018). We will examine whether the resultant Bayes Factor (BF10) ≥ 10, which indicates the strength of evidence that strongly supports the non-zero international difference (Han et al., 2018).	We will check if $p < .05$, for both results from frequentist (with "Anova" in R) and Bayesian ANOVAs. We intend to primarily refer to the result from Bayesian ANOVA for interpretation, but intend to provide the results from frequentist ANOVA for reasons of completeness. For the Bayesian analysis, we will first examine whether the resultant BF10 ≥ 10. IF BF10 ≥ 10, THEN we would conclude that evidence strongly supports the non-zero international difference. IF $3 \leq \text{BF10} < 10$, THEN it would indicate that evidence positively supports the non-zero difference, but not strongly. IF BF10 < 3, THEN it would indicate the lack of evidence to make any decision. IF BF10 ≤ -3, THEN the null hypothesis would be supported in lieu of H1a.
H1b Concern over the coronavirus will differ between countries	See descriptions provided in H1a and section 6.	We will perform one-way ANOVA to examine the international differences. DV: Concern over the coronavirus ([Corona_concerns]) IV: Country ([country]) IF $p < .05$ is reported, THEN we will perform a post-hoc test with Scheffe's method for exploration. As for H1a, we will also perform the same one-way ANOVA with Bayesian inference (see descriptions provided in H1a).	See descriptions provided in H1a. We will apply the same criteria to examine whether the effect of country is non-zero (H1b).
H1c Trust in government efforts to slow the spread of coronavirus will differ between countries	See descriptions provided in H1a and section 6.	We will perform one-way ANOVA to examine the international differences. DV: Trust in country's government efforts ([OECD_institutions Q1])	See descriptions provided in H1a. We will apply the same criteria to examine whether the effect of

		IV: Country ([country]) IF $p < .05$ is reported, THEN we will perform a post-hoc test with Scheffe's method for exploration. As for H1a, we will also perform the same one-way ANOVA with Bayesian inference (see descriptions provided in H1a).	country is non-zero (H1c).
H1d Compliance with behavioural guidelines to slow the spread of coronavirus will differ between countries	See descriptions provided in H1a and section 6.	We will perform one-way ANOVA to examine the international differences. DV: Compliance with behavioural guidelines to slow the spread of coronavirus ([Compliance]) IV: Country ([country]) IF $p < .05$ is reported, THEN we will perform a post-hoc test with Scheffe's method for exploration. As for H1a, we will also perform the same one-way ANOVA with Bayesian inference (see descriptions provided in H1a).	See descriptions provided in H1a. We will apply the same criteria to examine whether the effect of country is non-zero.
H2a Across countries, perceived stress will negatively predict compliance with behavioural guidelines to slow the spread of coronavirus, when controlling for individual-level covariates (age, gender, education, and generalised trust) and country-level covariates (population size, GDP per capita, education attainment, unemployment rate, income inequality and life expectancy).(see the model specified below)	See descriptions provided in H1a and section 6.	We will examine a MLM without interaction effects to test the relationship between stress and compliance. DV: Compliance with behavioural guidelines to slow the spread of coronavirus [Compliance] Level 1 IVs: Perceived stress ([Scale_PSS10]), individual demographic variables ([age], [gender], [edu], [generalised_trust]) Level 2 IVs: Country-level indicators and country demographic variables ([population_size], [GDP per capita], [edu_attainment],[unemployment], [gini_coefficient],[life_expectancy]) We include [country] as a random intercept in the model. In addition, we will perform the same Bayesian MLM with brms R package. We will examine Bayes Factor of the estimated B for perceived stress at Level 1 is 10 or greater ($BF_{10} \geq 10$). Because we intend to test whether B is non-zero (H0), we will use non-informative	We will examine both the p-value and Bayes Factor of the B for the Level 1 stress for completeness of our analysis. We will use $p < .05$ and $BF \geq 10$ as the indicators for the presence of the non-zero effect of perceived stress. IF $BF_{10} \geq 10$ and $p < .05$, THEN we would conclude that evidence strongly supports the non-zero effect of perceived stress. IF $3 \leq BF_{10} < 10$ and $p < .05$, THEN they would indicate that evidence positively supports the non-zero effect, but not strongly. IF $BF_{10} < 3$ and/or $p \geq .05$, THEN they would indicate the lack of evidence to make any decision. IF $BF_{10} \leq -3$, THEN the null hypothesis would be supported in lieu of H2a.

		priors centred around zero for brms and Bayes Factor calculation. Following Rouder and Morey's (2012) suggestions on Bayesian multivariate regression analysis, we will use a Cauchy prior ($d = .00$, scale = 1.00) for Level 1 coefficients. For other indicators, we will use default priors set by brms.	
H2b Across countries, concern over coronavirus will predict compliance with behavioural guidelines to slow the spread of coronavirus, when controlling for individual- and country-level covariates (see the model specified below)	See descriptions provided in H1a and section 6.	We will examine a MLM without interaction effects to test the relationship between concern over coronavirus and compliance. DV: Compliance with behavioural guidelines to slow the spread of coronavirus [Compliance] Level 1 IVs: Concern over coronavirus ([Corona_concerns]), individual demographic variables ([age], [gender], [edu], [generalised_trust]) Level 2 IVs: Country-level indicators and country demographic variables ([population_size], [GDP], [edu_attainment],[unemployment], [gini_coefficient],[life_expectancy]) We include [country] as a random intercept in the model. In addition, as for H2a, we will perform the same Bayesian MLM with brms R package. (see descriptions provided in H2a).	See descriptions provided in H2a. We will apply the same criteria to examine whether the effect of concern over the coronavirus is non-zero (H2b).
H3a Across countries, trust in government efforts to slow the spread of coronavirus will negatively predict perceived stress, when controlling for individual- and country-level covariates	See descriptions provided in H1a and section 6.	We will examine a MLM without interaction effects to test the relationship between trust in government efforts to slow the spread of coronavirus and perceived stress. DV: Perceived stress ([Scale_PSS10]) Level 1 IVs: Trust in government efforts to slow the spread of coronavirus ([OECD_institutions Q1]), individual demographic variables ([age], [gender], [edu], [generalised_trust]) Level 2 IVs: Country-level indicators and country demographic variables ([population_size], [GDP], [edu_attainment],[unemployment], [gini_coefficient],[life_expectancy]) We include [country] as a random intercept in the model.	See descriptions provided in H2a. We will apply the same criteria to examine whether the effect of trust is non-zero (H3a).

		In addition, as for H2a, we will perform the same Bayesian MLM with brms R package. (see descriptions provided in H2a).	
H3b Across countries, trust in government efforts to slow the spread of coronavirus will negatively predict concern over coronavirus, when controlling for individual- and country-level covariates	See descriptions provided in H1a and section 6.	We will examine a MLM without interaction effects to test the relationship between concern over coronavirus and trust in government efforts to slow the spread of coronavirus. DV: Concern over coronavirus ([Corona_concerns]) Level 1 IVs: Trust in government efforts to slow the spread of coronavirus ([OECD_institutions Q1]), individual demographic variables ([age], [gender], [edu], [generalised_trust]) Level 2 IVs: country-level indicators and country demographic variables ([population_size], [GDP], [edu_attainment],[unemployment], [gini_coefficient],[life_expectancy]) We include [country] as a random intercept in the model. In addition, as for H2a, we will perform the same Bayesian MLM with brms R package. (see descriptions provided in H2a).	See descriptions provided in H2a. We will apply the same criteria to examine whether the effect of trust is non-zero (H3b).
H4 Across countries, concern over coronavirus will predict perceived stress, when controlling for individual- and country-level covariates	See descriptions provided in H1a and section 6.	We will examine a MLM without interaction effects to test the relationship between concern over coronavirus and stress. DV: Perceived stress ([Scale_PSS10]) Level 1 IVs: Concern over coronavirus ([Corona_concerns]), individual demographic variables ([age], [gender], [edu], [generalised_trust]) Level 2 IVs: country-level indicators and country demographic variables ([population_size], [GDP], [edu_attainment],[unemployment], [gini_coefficient],[life_expectancy]) We include [country] as a random intercept in the model. In addition, as for H2a, we will perform the same Bayesian MLM with brms R package. (see descriptions provided in H2a).	See descriptions provided in H2a. We will apply the same criteria to examine whether the effect of concern over coronavirus is non-zero (H4).

H5 Across countries, availability of social provisions will negatively moderate the effect of perceived stress on compliance with behavioural guidelines to slow the spread of coronavirus, when controlling for individual- and country-level covariates (see the model specified below)	See descriptions provided in H1a and section 6.	We will examine a MLM with the intended moderator. DV: Compliance with behavioural guidelines to slow the spread of coronavirus [Compliance] Level 1 IVs: Perceived stress ([Scale_PSS10]), social provisions ([SPS]), perceived stress x social provisions, individual demographic variables ([age], [gender], [education], [generalised_trust]) Level 2 IVs: Country-level indicators, aggregated variables (country-level stress), and country demographic variables ([population_size], [GDP], [edu_attainment],[unemployment], [gini_coefficient],[life_expectancy]) We include [country] as a random intercept and a random slope of perceived stress in the model. The planned multilevel models will be run step by step to make sure that each random effect is added by order (i.e., country random intercept, the random slope of perceived stress), so that the model reaches convergence. We will keep a more parsimonious model and drop the random effect in the case of non-convergence. In addition, as for H2a, we will perform the same Bayesian MLM with brms R package. (see descriptions provided in H2a). We will add the aforementioned interaction effects in the brms model.	To test the moderation effect, we will examine both the p-value and Bayes Factor of the B for the moderator of interest. We will use $p < .05$ and $BF \geq 10$ as the indicators for the presence of the non-zero effect of the moderator. IF $BF_{10} \geq 10$ and $p < .05$, THEN we would conclude that evidence strongly supports the non-zero moderation. IF $3 \leq BF_{10} < 10$ and $p < .05$, THEN they would indicate that evidence positively supports the non-zero moderation, but not strongly. IF $BF_{10} < 3$ and/or $p \geq .05$, THEN they would indicate the lack of evidence to make any decision. IF $BF_{10} \leq -3$, THEN the null hypothesis would be supported in lieu of H5.
---	--	---	---

Model Specification for Multilevel Model

In the designed model, the outcome, compliance with behavioral guidelines, is a function of fixed and random intercepts (B00 and R0), perceived stress (B70 and R1) and availability of social provisions (B80), when controlling for both individual demographic variables (B10 - B60) and country demographic variables (B01 - B06). A contextual predictor, country-level stress (B08), is aggregated from individual-level stress, to refer to the average stress level of a given country. Two interaction terms are created to probe into the moderating role of social provisions in the relationship between individual-level stress and compliance (B90) and in the relationship between country-level stress and compliance (B81). Country-level stress will be tested as a moderator for the relationship between individual-level stress and compliance (B71).

Level 1 (Individual):

$$Y = P_0 + P_1*(Gender) + P_2*(Age) + P_3*(Education) + P_4*(GeneralisedTrust) + \dots + P_7*(Stress) + P_8*(SocialProvisions) + P_9*(Stress \times SocialProvisions) + E$$

Level 2 (Country):

$$P_0 = B_{00} + B_{01}*(PopulationSize) + B_{02}*(GDP) + \dots + B_{08}*(CStress) + R_0$$

$$P1 = B10, P2 = B20, P3 = B30 \dots P6 = B60$$

$$P7 = B70 + B71*(CStress) + R1$$

$$P8 = B80 + B81*(CStress)$$

$$P9 = B90$$

Compositional Model for Multilevel Analysis:

$$\begin{aligned} \text{Compliance} = & B00 + B10*(\text{Gender}) + B20*(\text{Age}) + B30*(\text{Education}) + \\ & B40*(\text{GeneralisedTrust}) \dots B01*(\text{PopulationSize}) + B02*(\text{GDP}) \dots + B08*(\text{CStress}) + (B70 + \\ & R1)*(Stress) + B71*(\text{CStress}*Stress) + B80*(\text{SocialProvisions}) + B81*(\text{CStress}*SocialProvisions) + \\ & B90*(\text{Stress} \times \text{SocialProvisions}) + R0 + E \end{aligned}$$

10. New data or analysis of existing data

We collect new data. The survey has been pre-registered at the Open Science Framework on March 30th 2020 (COVIDiSTRESS global survey network, 2020), and scheduled for weekly data extractions for public access, to be made available there. The data collection started on March 30th 2020 in all participating countries, with a pre-launch in Denmark on March 26 2020 to test if the survey works in practice and get a first impression on the response rate. Prior to submitting this Registered Report, we have not accessed or analysed any data, except following response rates to give all COVIDiSTRESS participant researchers feedback on the recruiting progress.

As this Registered Report is submitted under the COVID-19 rapid response call for papers, upon acceptance of the stage one protocol, we will analyze the newest weekly data extraction that fit minimum criteria (see above) for the present analysis. While we intend to stop data collection at the latest on April 13 2020 for the data analysis for this RR, we will continue with the data collection until May 30th 2020. The data collected is made available to any other researcher for further hypotheses testing.

11. Results

To be added in the second stage Registered Report.

12. Discussion

To be added in the second stage Registered Report.

References

- Arend, M. G., & Schäfer, T. (2019). Statistical power in two-level models: A tutorial based on Monte Carlo simulation. *Psychological Methods*, 24(1), 1–19.
- Bish, A., & Michie, S. (2010). Demographic and attitudinal determinants of protective behaviours during a pandemic: a review. *British Journal of Health Psychology*, 15(4), 797–824.
- Brooks, S. K., Webster, R. K., Smith, L. E., Woodland, L., Wessely, S., Greenberg, N., & Rubin, G. J. (2020). The psychological impact of quarantine and how to reduce it: rapid review of the evidence. *The Lancet*, 395(10227), 912–920. [https://doi.org/10.1016/S0140-6736\(20\)30460-8](https://doi.org/10.1016/S0140-6736(20)30460-8)
- Bults, M., Beaujean, D.J., de Zwart, O., Kok, G., van Empelen, P., van Steenbergen, J. E., Richardus, J. H., & Voeten, H. A. C. M. (2011). Perceived risk, anxiety, and behavioural responses of the general

1
2
3 public during the early phase of the Influenza A (H1N1) pandemic in the Netherlands: Results of
4 three consecutive online surveys. *BMC Public Health*, 11(1), 2. <https://doi.org/10.1186/1471-2458-11-2>

Capelos, T., Provost, C., Parouti, M., Barnett, J., Chenoweth, J., Fife-Schaw, C. & Kelay, T. (2016)
Ingredients of institutional reputations and citizen engagement with regulators. *Regulation &*
*Governance*, 10(4), 350–376. <https://doi.org/10.1111/rego.12097>

Cava, M. A., Fay, K. E., Beanlands, H. J., McCay, E. A., & Wignall, R. (2005). The Experience of
Quarantine for Individuals Affected by SARS in Toronto. *Public Health Nursing*, 22(5), 398–406.
<https://doi.org/10.1111/j.0737-1209.2005.220504.x>

Cheng, C. and Ng, A.-K. (2006), Psychosocial Factors Predicting SARS-Preventive Behaviors in Four
Major SARS-Affected Regions. *Journal of Applied Social Psychology*, 36, 222–247.
<https://doi.org/10.1111/j.0021-9029.2006.00059.x>

Cho, Y. J. (2008). *Culture, sex-role, mutual social support and adult attachment as predictors of*
*Korean couples' relationship satisfaction* (Doctoral dissertation, University of Missouri-Columbia).

Cohen, S. (2004). Social Relationships and Health. *American Psychologist*, 59(8), 676–684.

Cohen, S., Kamarck, T., & Mermelstein, R. (1983). A Global Measure of Perceived Stress. *Journal of*
*Health and Social Behavior*, 24(4), 385–396.

Cohen, S., & Wills, T. A. (1985). Stress, social support, and the buffering hypothesis. *Psychological*
*Bulletin*, 98(2), 310.

Curran, P. G. (2016). Methods for the detection of carelessly invalid responses in survey data. *Journal*
*of Experimental Social Psychology*, 66, 4–19. <https://doi.org/10.1016/j.jesp.2015.07.006>

Dar, A. Wahid. (2017). Cross Cultural Understanding of Stress and Coping Mechanisms.
*International Journal of Research in Engineering, IT and Social Sciences*, 7(10),75–80. Retrieved
from:
[https://www.researchgate.net/publication/331372626_Cross_Cultural_Understanding_of_Stress_and](https://www.researchgate.net/publication/331372626_Cross_Cultural_Understanding_of_Stress_and_Coping_Mechanisms)
[Coping_Mechanisms](https://www.researchgate.net/publication/331372626_Cross_Cultural_Understanding_of_Stress_and_Coping_Mechanisms) [last accessed 05.04.2020].

Dupuis, M., Meier, E., & Cuneo, F. (2019). Detecting computer-generated random responding in
questionnaire-based data: A comparison of seven indices. *Behavior Research Methods*, 51(5), 2228–
2237. <https://doi.org/10.3758/s13428-018-1103-y>

Egleston, B. L., Miller, S. M., & Meropol, N. J. (2011). The impact of misclassification due to survey
response fatigue on estimation and identifiability of treatment effects. *Statistics in Medicine*, 30(30),
3560–3572.

Fung, I. C. H., & Cairncross, S. (2007). How often do you wash your hands? A review of studies of
hand-washing practices in the community during and after the SARS outbreak in 2003. *International*
*Journal of Environmental Health Research*, 17(3), 161–183.

Gilles, I., Bangerter, A., Clemence, A., Green, G. T. E., Krings, F., Staerke, C. & Wagner-Egger P.
(2011). Trust in medical organizations predicts pandemic (H1N1) 2009 vaccination behavior and
perceived efficacy of protection measures in the Swiss public. *European Journal of Epidemiology*,
26(3), 203–2011

Goldman, M., Gier, J. A., & Smith, D. E. (1981). Compliance as affected by task difficulty and order
of tasks. *The Journal of Social Psychology*, 114(1), 75–83.

Gunderson, E. E. (1974). Psychological studies in Antarctica. *Human Adaptability in Antarctic*
*conditions, Antarctic Research Series, Vol. 22*, 115–131.

Han, H., Park, J., & Thoma, S. J. (2018). Why do we need to employ Bayesian statistics and how can
we employ it in studies of moral education?: With practical guidelines to use JASP for educators and
researchers. *Journal of Moral Education, 47*(4), 519–537.

Hawryluck, L., Gold, W. L., Robinson, S., Pogorski, S., Galea, S., & Styra, R. (2004). SARS Control
and Psychological Effects of Quarantine, Toronto, Canada. *Emerging Infectious Diseases, 10*(7),
1206–1212. <https://doi.org/10.3201/eid1007.030703>

Hitchcock, P. B., Brantley, P. J., Jones, G. N., & McKnight, G. T. (1992). Stress and social support as
predictors of dietary compliance in hemodialysis patients. *Behavioral Medicine, 18*(1), pp. 13–20.

Hoffman, L. (2015). *Longitudinal analysis: Modeling within-person fluctuation and change*. New
York, NY: Routledge Press.

Hox, J. (2010). *Multilevel Analysis: Techniques and Applications*, Mahwah, NJ: Lawrence Erlbaum
Associates.

Huremović, D. (Ed.). (2019). *Psychiatry of Pandemics: A Mental Health Response to Infection*
*Outbreak*. Springer Nature Switzerland.

Karvinen, K. H., Murray, N. P., Arastu, H., & Allison, R. R. (2013). Stress Reactivity, Health
Behaviors, and Compliance to Medical Care in Breast Cancer Survivors. *Oncology Nursing Forum,*
*40*(2), 149–156. <https://doi.org/10.1188/13.ONF.149-156>

Kehoe, J. P., & Abbott, A. P. (1975). Suicide and attempted suicide in the Yukon territory. *Canadian*
*Psychiatric Association Journal, 20*, 15–23.

Khan, S. & Huremović, D. (Ed.). (2019). Psychology of the Pandemic. In Huremović, D. (Ed.)
*Psychiatry of Pandemics: A Mental Health Response to Infection Outbreak*. (pp. 37–44). Springer
Nature Switzerland.

Kim, S-Y., Kim, J-M., Yoo, J-A., Bae, K-Y., Kim, S-W., Yang, S-J., Shin, I-S., & Yoon, J-S. (2010).
Standardization and validation of Big Five Inventory-Korean Version(BFI-K) in elders. *Korean*
*Journal of Biological Psychiatry, 17*(1), 15–25.

Klein, R. A., Ratliff, K. A., Vianello, M., Adams Jr., R. B., Bahník, Š., Bernstein, M. J., ... Nosek, B.
43 A. (2014). Investigating variation in replicability: A “many labs” replication project. *Social*
*Psychology, 45*(3), 142–152. <https://doi.org/10.1027/1864-9335/a000178>

Krosnick, J. A., & Alwin, D. F. (1987). An evaluation of a cognitive theory of response-order effects
in survey measurement. *Public Opinion Quarterly, 51*(2), 201–219.

Lau, J. T., Yang, X., Tsui, H., Pang, E., & Kim, J. H. (2004). SARS preventive and risk behaviours of
Hong Kong air travellers. *Epidemiology & Infection, 132*(4), 727–736.

Lee, J-H., Shin, C-M., Ko, Y-H., Lim, J-H., Joe, S-H., Kim, S-H., Jung, I-K., Han, C-S. (2012). The
reliability and validity studies of the Korean version of the Perceived Stress Scale. *Korean Journal of*
*Psychosomatic Medicine, 20*(2), 127–134.

Leung, C. M., Lam, T-H., Ho L-M., Ho S-Y., Chan B. H. Y., Wong I., O., L. & Hedley, A. J. (2003).
The impact of community psychological responses on outbreak control for severe acute respiratory
syndrome in Hong Kong. *Journal of Epidemiological Community Health*, 57, 857–863.

Lynch, D. A. (1999). Sources of stress, perceived stress levels, and levels of self-esteem among career
oriented mothers who are employed full-time, part-time, and those who opted to stay home full-time
(Doctoral dissertation, Seton Hall University, NJ, 1999). *Dissertation Abstracts International*, 60(3-
B), 1307.

Mak, W. W., Law, R. W., Woo, J., Cheung, F. M., & Lee, D. (2009). Social support and
psychological adjustment to SARS: The mediating role of self-care self-efficacy. *Psychology &*
*Health*, 24(2), 161–174. <https://doi.org/10.1080/08870440701447649>

Marjanovic, Z., Greenglass, E.R., Coffey S. (2007). The relevance of psychosocial variables and
working conditions in predicting nurses' coping strategies during the SARS crisis: an online
questionnaire survey. *International Journal of Nursing Studies*, 44, 991–998.

Maunder, R. G. (2009). Was SARS a mental health catastrophe? [Editorial]. *General Hospital*
*Psychiatry*, 31(4), 316–317. <https://doi.org/10.1016/j.genhosppsy.2009.04.004>

Mellins, C. A., & Ehrhardt, A. A. (1994). Families affected by pediatric acquired immunodeficiency
syndrome: Sources of stress and coping. *Journal of Developmental and Behavioral Pediatrics*, 15(3),
54–60.

Neighbors, H. W. (1997). Husbands, wives, family, and friends: Sources of stress, sources of support.
In R. J. Taylor, J. S. Jackson & L. M. Chatters (Eds.), *Family Life in Black America* (pp. 277–292).
Thousand Oaks, CA: Sage Publications.

O'Connor, D. B., Ferguson, E., & O'Connor, R. C. (2005). Intentions to use hormonal male
contraception: The role of message framing, attitudes and stress appraisals. *British Journal of*
*Psychology*, 96(3), 351–369.

Organisation for Economic Co-operation and Development (2017). *OECD guidelines on measuring*
*trust*. OECD Publishing. <http://dx.doi.org/10.1787/9789264278219-en>

Oliver, D. C. (1991). Psychological effects of isolation and confinement of a winter-over group at
McMurdo Station, Antarctica. In *From Antarctica to Outer Space* (pp. 217–227). Springer, New
York, NY.

Palinkas, L. A., & Suedfeld, P. (2008). Psychological effects of polar expeditions. *The Lancet*,
371(9607), 153–163.

Petersen, M.B. (2020, March 9). The unpleasant truth is the best protection against coronavirus (M. B.
Petersen, Trans.) *Politiken*. Retrieved from
https://pure.au.dk/portal/files/181464339/The_unpleasant_truth_is_the_best_protection_against_coronavirus_Michael_Bang_Petersen.pdf [last accessed 05.04.2020]

Prati, G., Pietrantonio, L. & Zani B. (2011). Compliance with recommendations for pandemic influenza
H1N1 2009: the role of trust and personal beliefs. *Health Education Research*, 26(5), 761–769.

Raue, M., Streicher, B., Lerner, E., & Frey, D. (2015). How far does it feel? Construal level and
decisions under risk. *Journal of Applied Research in Memory and Cognition*, 4(3), 256–264.

Reynolds, D. L., Garay, J. R., Deamond, S. L., Moran, M. K., Gold, W., & Styra, R. (2008).
Understanding, compliance and psychological impact of the SARS quarantine experience.
*Epidemiology and Infection*, 136(7), 997–1007. <https://doi.org/10.1017/S0950268807009156>
- Ro, J-S., Lee, J-S., Kang, S-C., Jung, H-M. (2017). Worry experienced during the 2015 Middle East
Respiratory Syndrome (MERS) pandemic in Korea. *PLoS ONE*, 12(3),
e0173234. <https://doi.org/10.1371/journal.pone.0173234>
- Rouder, J. N., & Morey, R. D. (2012). Default Bayes factors for model selection in regression.
*Multivariate Behavioral Research*, 47(6), 877–903.
- Seligman, A. B. (1997). *The Problem of Trust*. Princeton, NJ: Princeton University Press
- Steigen, A. M., & Bergh, D. (2019). The Social Provisions Scale: psychometric properties of the SPS-
10 among participants in nature-based services. *Disability and Rehabilitation*, 41(14), 1690–1698.
<https://doi.org/10.1080/09638288.2018.1434689>
- The COVIDiSTRESS global survey network (2020). *COVIDiSTRESS global survey pre-registration*.
<https://doi.org/10.17605/osf.io/z39us>.
- Trautmann, S. T., & van de Kuilen, G. (2012). Prospect theory or construal level theory?:
Diminishing sensitivity vs. psychological distance in risky decisions. *Acta Psychologica*, 139(1), 254–
260.
- Trope, Y., & Liberman, N. (2010). Construal-level theory of psychological distance. *Psychological*
*Review*, 117(2), 440.
- Van't Riet, J., Cox, A. D., Cox, D., Zimet, G. D., De Bruijn, G., Van den Putte, B., . . . Ruiter, R. A.
C. (2014). Does perceived risk influence the effects of message framing? A new investigation of a
widely held notion. *Psychology & Health*, 29(8), 933–949.
- Wagenmakers, E.-J., Love, J., Marsman, M., Jamil, T., Ly, A., Verhagen, A. J., Selker, R., Gronau, Q.,
Dropmann, D., Boutin, B., Meerhoff, F., Knight, P., Raj, A., Kesteren, E.-J., Van Doorn, J., Šmíra, M.,
Epskamp, S., Etz, A., Matzke, D. & Morey, R. (2017). Bayesian inference for psychology. Part II:
Example applications with JASP. *Psychonomic Bulletin & Review*, 25, 1–19.
<https://doi.org/10.3758/s13423-017-1323-7>
- Wang, X., Cai, L., Qian, J., & Peng, J. (2014). Social support moderates stress effects on depression.
*International Journal of Mental Health Systems*, 8(1), 41. <https://doi.org/10.1186/1752-4458-8-41>
- Wong, T. W., Gao, Y., & Tam, W. W. S. (2007). Anxiety among university students during the SARS
epidemic in Hong Kong. *Stress and Health: Journal of the International Society for the Investigation*
*of Stress*, 23(1), 31–35.

Appendix B

I reviewed RSOS-200589 (Stage 1 RR). Overall, this proposal is promising in part because of the wide network, timeliness, and care in planning and setting the analytic pipeline. However, it seems incomplete. I hope my comments are useful in quickly revising the project.

The scientific validity of the research question(s)

The introduction does a good job of setting the overall context of the work and why it is important, but a less good job of why these particular variables are being measured and what they will enable tests of. It is also unclear why the data is being collected over time: the planned analyses didn't seem to include time.

The introduction is a little disorganized; for example, the narrative started with how this research could inform policy making and behavior change communications, and then pivoted to mental health outcomes, and then back to behavior change.

There are many similar survey projects being launched right now. I've heard of several consortia. I was surprised to see no mention of this similar research, e.g.: <https://wintoncentre.maths.cam.ac.uk/news/how-different-countries-are-reacting-to-the-covid-19-risk-and-their-governments-responses/> data: <https://osf.io/jnu74/>. At the least, it would be useful to articulate how the current effort differs from existing results and from ongoing, related surveys.

To the extent that the intro focuses on behavior change, it would be useful to spend more time explaining the context of previous interventions (e.g., but not limited to nudges). Also, specifically on COVID-19 this report examines behavior change to different messages: <https://www.bi.team/blogs/bright-infographics-and-minimal-text-make-handwashing-posters-most-effective/>

This tracker of cross-governmental responses (e.g., lockdown severity) may be useful to integrate: <https://www.bsg.ox.ac.uk/research/research-projects/oxford-covid-19-government-response-tracker>

The logic, rationale, and plausibility of the proposed hypotheses

RQ1/H2A/B assume the causal direction of (+) trust  (-) stress / (-) concern. Those are plausible, but so would be the reversed arrows. Unless the longitudinal data will be used with, e.g., HLM to test for both causal directions, the causal language should be softened and this caveat stated explicitly for all cross-sectional data. The same point applies to H4.

For H5, it might be useful to connect this directly to the famous stress buffering hypothesis from health psychology, whether or not this particular ref (18000 citations??): <https://psycnet.apa.org/record/1986-01119-001>. I found this moderation underjustified theoretically, for example compared to an alternate claim like (+) social provisions acting as an independent, direct predictor of (+) compliance.

The soundness and feasibility of the methodology and analysis pipeline

I didn't deeply look into the estimated sample size needed per country, re: Arend and Schäfer, but $n = 30$ feels low.

The exclusion criteria may be too strict. Why discard partial responses?

I didn't see any discussion of how these countries were selected (e.g., feasibility; connections) nor of the representativeness of those countries.

Is $\alpha = .05$ appropriate given the number of inferential tests? I wasn't clear whether the hypotheses would be tested within each country. If so, some adjustment for multiple comparisons seems appropriate, either by lowering alpha or using Bonferonni-Holm, etc.

I am not trained adequately in Bayesian methods to review those analysis plans.

Whether the clarity and degree of methodological detail would be sufficient to replicate exactly the proposed experimental procedures and analysis pipeline

Yes, although not much detail was given about recruitment techniques (e.g., email; social media), materials (advertisements), or target populations (e.g., snowballing in student samples?).

Whether the authors have considered sufficient outcome-neutral conditions (e.g. positive controls) for ensuring that the results obtained are able to test the stated hypotheses

I think so. It is unclear how the authors would go about interpreting a null effect for one of the hypothesized relationships; there aren't many positive controls.

Whether the authors provide a sufficiently clear and detailed description of the methods to prevent undisclosed flexibility in the experimental procedures or analysis pipeline

Broadly, yes. It was not always clear to me how the MLM would be constructed, e.g., for which analyses the country level would be included.

Appendix C

Aarhus, May 2020

Response to Reviewers — RSOS-200589

Dear Prof. Chambers and Prof. Dunn,

It is with pleasure that we submit to you a revised version of our manuscript. We appreciate your feedback and the feedback from five reviewers. We believe that the revision is much stronger than the previous manuscript.

In the new version, we have updated the title and literature review to address reviewer comments, and adapted the hypotheses to a more correlational language. Also, to more systematically present how we will interpret findings based on both p -values and Bayes Factors, we summarized our decision-making process in a table (see Tables 2 and 3 in the RR).

As also stressed by Reviewer 3 and 4, cross-country descriptives and more exploratory analyses will constitute a large part of the finished article. Even though we cannot specify these in the stage one RR, we would like to underscore their centrality to the final manuscript, and the hypotheses as a route to creating this final broad overview.

As called for by reviewer two, we have softened the language in the title and hypotheses to one of correlation rather than unidirectional causality.

While several reviewers have suggested adding additional country level variables to our analyses, we stress that this is the first publication to come out of the COVIDiSTRESS global survey, which we intend to showcase data from as many participating countries as possible. However, because of the relatively small number of countries (approximately 50-60) we can include in a multilevel model, our study would be underpowered to detect both direct effects of country-level predictors and cross-level interactions. Therefore, we will focus on hypotheses examining individual-level relationships, as stated in our hypotheses 2-5. To this point, we appreciate the reviewers' comments that invite detailed exploratory analyses and illustrations of between-country differences.

Following your advice, we also addressed some methodological shortcomings and more explicitly discussed the methodological limitations of our research. Below, please find the detailed response to the reviewers' individual remarks.

Thank you very much in advance for considering the revision of our manuscript. We are looking forward to hearing from you.

Sincerely,

The COVIDiSTRESS global survey group

Comments to Reviewer 1

Thank you for evaluating our manuscript and providing detailed feedback.

As we wrote in the abstract of an article on this topic, “When comparing groups an assumption is made that the instrument measures the same psychological construct in all groups. If this assumption holds, the comparisons are valid and differences/similarities between groups can be meaningfully interpreted. If this assumption does not hold, comparisons and interpretations are not fully meaningful. The establishment of measurement invariance is a prerequisite for meaningful comparisons across groups.” (Milfont & Fischer, 2010).

There are three main levels of invariance testing: configural (underlying factor model identical across groups), metric (item loadings identical across groups), and interval (item intercepts identical across groups). The specialised literature argues that metric invariance is needed for examining associations between constructs across groups, and that mean comparisons across groups are only meaningful when interval invariance is achieved. All hypotheses in the Registered Report focus on mean comparisons or associations between constructs across cultural groups but no invariance testing is proposed. This needs to be addressed! Without testing and confirming the multi-item measures are equivalent across the groups, the testing of all proposed hypotheses are meaningless. For example, if participants across the considered nations do not respond to items of the Perceived Stress Scale similarly, then comparing stress means across countries (H1a) can't be meaningfully interpreted.

When revising the Registered Report to include measurement invariance, I suggest the authors to explore the alignment method, which is a much easier implementation of invariance testing – particularly for large groups as in this case. I believe alignment can already be implemented in R. I recommend two main publications on the topic: Asparouhov and Muthén (2014) and Marsh et al. (2018); and we have also implemented this approach in a recent article (see Milfont et al. 2018) and have discussed measurement invariance for replicating findings across cultural groups (see Milfont & Klein, 2018). I am sure the authors would know measurement invariance experts in their networks but I can make myself available to provide some guidance if needed (and if the Editor and authors feels this is appropriate).

1.1

First of all, we would welcome the Reviewer's collaboration, as, based on the suggested literature overview, we believe that the Reviewer could significantly contribute to the invariance analyses (if the Editor allows).

We have also extended the information about planned analyses of the equivalence of invariance in the Quality checks section. Based on the comment, we now intend to investigate the cross-national equivalence of the Perceived Stress Scale (PSS-10, see Cohen et al., 1983; Cohen & Williamson, 1988) prior to analysis. Using a multi-group factor analysis we will compare the models assuming the two-factor structure (positive and negative, with the latter consisting of reversed items; Chaaya et al., 2010; Roberti et al., 2006) across all countries (configural invariance), with a model with factor loadings and latent correlations constrained to be equal (metric invariance), and items' intercepts to be the same in all groups (scalar invariance). When evaluating the model fit, we will rely on the

usually applied criteria (Hu & Bentler, 1999), in which a Comparative Fit Index (CFI) and Tucker Lewis Index (TLI) above .90 would indicate adequate fit, whereas a standardized root mean square residual (SRMR) below .06, and a root mean square error of approximation (RMSEA) below .08 indicate no misfit. When evaluating the measurement equivalence, we will compare the configural invariance model with the metric invariance model, and then the metric invariance model with the scalar invariance model (Milfont & Fischer, 2010). As these models are characterized by a growing complexity (each subsequent model is nested within the previous one), while assessing models' superiority, we will rely on cut-off criteria recommended for testing measurement invariance: a change of *CFI* (ΔCFI) less than .01 ($\Delta CFI < .01$), a change of *RMSEA* of less than .015 ($\Delta RMSEA < .015$), and a change of *SRMR* less than .01 ($\Delta SRMR < .01$) indicating that two compared models do not differ in terms of model fit (Chen, 2008; Cheung & Rensvold, 2002). However, in case that the equivalence of invariance will not be achieved, based on recommendations indicating that partial invariance may allow for reasonable comparisons (see e.g., Byrne, Shavelson, & Muthén, 1989), we will also estimate models with partial invariance.

In addition, in the revised RR, we elaborated how we will conduct the planned analyses after performing the measurement invariance test and if some portion of the dataset should be excluded due to the failed measurement invariance test:

Given that the number of countries to be analysed directly influences the power of the planned multilevel modeling, we intend to conduct the multilevel modeling analyses as follows:

IF our planned analyses can be conducted without any issue (e.g., failed convergence) with the dataset that passed the measurement invariance test, THEN report the findings.

IF 1 failed, THEN conduct the planned analyses with the dataset that includes all countries with $N \geq 30$ (according to the country-level exclusion criteria).

If 1 failed and findings from 2 are reported, we will discuss limitations and caveats regarding the interpretation of the findings in the discussion section.

I feel the authors should make it explicit from the outset that they will employ a multilevel approach.

1.2

Thank you for the suggestion. We have added a paragraph in the introduction stressing that we use MLM analyses to test our hypotheses (see Section 1).

The authors propose a list of country-level variables and use them as control. This is a reasonable list of variables but there is no justification of their selection. This list can be much longer, especially if the authors decide to include psychological dimensions of cultural variability such as individualism/collectivism, tightness/looseness and relational mobility. It is

thus important to provide some rationale for selecting only non-psychological country-level variables and for selecting these specific ones.

1.3

I have added a paragraph explaining the justifications for the inclusion of both individual-level and country-level control variables, following the hypotheses (see Section 4).

As this is a very large and varied dataset, psychological between-countries differences are indeed planned to be the subject of a later analysis currently planned by other researchers in our group. This upcoming analysis is also intended to take differences in COVID-19 responses by each government into account.

We have updated H5, to test the relationship between individual stress, social provisions and compliance to behavior guidelines

Related to the point above, the country-level variables are merely used as control variables in the analyses. It would be great if these were included as predictors, at least for the first four hypotheses. Once invariance is established and meaningful comparisons can be made (see above), merely showing variability in stress, concern, trust and compliance might not tell much, or might be difficult for the readers and government officials to interpret the observed similarities/differences. It would be more interesting to show whether stress, concern, trust and/or compliance is greater/weaker in wealthier/poorer countries, for example. (Again, the list of variables could be very large and such analysis would not be attainable; by addressing the point above and providing justification for a small list of variables selected, the authors could provide such addition to the first hypotheses.)

1.4

Please see comment for 1.3

H3b reads "Across countries, trust in government efforts to slow the spread of coronavirus will negatively predict concern over coronavirus, when controlling for individual- and country-level covariates" but this negative prediction is not obvious to me. Greater concern should lead to stronger trust, no? Or alternatively, if I don't trust the government efforts (take delayed and denialistic actions by the USA and Brazil for example), I would be more concerned about the pandemic, no?

1.5

We see how the hypothesis is phrased a bit backwards. The idea is, that if you believe that the government measures will protect you, you will be less worried.

We have conducted further literature searches, and found some support for this quite intuitive notion (literature section updated), but interestingly not much has been written. It is going to be an interesting hypothesis to test.

I found H5 unclear. Do you mean that provisions would reduce/buffer the expected negative effect of stress on compliance (H2a)? You need to spell out. Given the relations between stress and concern (see H2a, H2b and H4), I believe another moderation hypothesis is reasonable and could be included by examining the extent to which provisions increases/enhances the expected positive effect of concern on compliance.

1.6

Thank you for the suggestion. We have strengthened the relevant arguments in the introduction section, and make it clear that we examine the buffering effect of social provisions on the negative impact of stress on compliance (see Page 3 in Section 1).

I have made comments in the original PDF. I have highlighted sentences that are unclear of miss words, and have added missing words too (e.g., provide clear direction for some of the hypothesis).

1.7

We would like to thank the Reviewer for their helpful comments. We have changed the sentences and added missing words.

Comments to Reviewer 2

Thank you for evaluating our manuscript and providing detailed feedback.

The introduction does a good job of setting the overall context of the work and why it is important, but a less good job of why these particular variables are being measured and what they will enable tests of. It is also unclear why the data is being collected over time: the planned analyses didn't seem to include time.

2.1

We have adjusted the introductory section accordingly. The text is now focused more on the specific variables we are planning to include in our analysis, and the complexities that emerge from that combined space. We believe that the link between the introductory part and the subsequent sections is now clearer. This paper is the first of several that will be coming out of our consortium. Presently we wish to take advantage of the rapid response scheme to share the current picture. Once more data has been collected, a time perspective will be forthcoming, but since it is difficult to project the future trajectory of COVID-19 related developments, and thus a pragmatic stop-rule for data collection overall, this analysis will be better suited as a retrospective analysis, using data from the coming months.

The introduction is a little disorganized; for example, the narrative started with how this research could inform policy making and behavior change communications, and then pivoted to mental health outcomes, and then back to behavior change.

2.2

We have edited the introduction to lead more clearly into the analysis.

There are many similar survey projects being launched right now. I've heard of several consortia. I was surprised to see no mention of this similar research, e.g.: <https://wintoncentre.maths.cam.ac.uk/news/how-different-countries-are-reacting-to-the-covid-19-risk-and-their-governments-responses/> data: <https://osf.io/jnu74/>. At the least, it would be useful to articulate how the current effort differs from existing results and from ongoing, related surveys.

2.3

Thank you for pointing this out. We now refer to other projects and emphasize that our project is unique in the sense that it is a global survey relying on organic sampling rather than e.g. purchased responses from predetermined subject pools (see page 1).

To the extent that the intro focuses on behavior change, it would be useful to spend more time explaining the context of previous interventions (e.g., but not limited to nudges). Also, specifically on COVID-19 this report examines behavior change to different messages: <https://www.bi.team/blogs/bright-infographics-and-minimal-text-make-handwashing-posters-most-effective/> This tracker of cross-governmental responses (e.g., lockdown severity) may be

useful to integrate:

<https://www.bsg.ox.ac.uk/research/research-projects/oxford-covid-19-governmentresponse-tracker>

2.4

This paper is the first of several. The tracker especially will be extremely useful in the future.

RQ1/H2A/B assume the causal direction of (+) trust  (-) stress / (-) concern. Those are plausible, but so would be the reversed arrows. Unless the longitudinal data will be used with, e.g., HLM to test for both causal directions, the causal language should be softened and this caveat stated explicitly for all cross-sectional data. The same point applies to H4.

2.5

We agree. The intention was not to suggest only one direction of causality. We have softened the language to one of correlation.

Colleagues from the consortium hope that causal relationships can be studied further down the line, as longitudinal data becomes available from the dataset.

For H5, it might be useful to connect this directly to the famous stress buffering hypothesis from health psychology, whether or not this particular ref (18000 citations??): <https://psycnet.apa.org/record/1986-01119-001>. I found this moderation underjustified theoretically, for example compared to an alternate claim like (+) social provisions acting as an independent, direct predictor of (+) compliance.

2.6

As per Comment 1.6, we have strengthened the rationale of the moderating effect of social provisions in the introduction section.

I didn't deeply look into the estimated sample size needed per country, re: Arend and Schäfer, but $n = 30$ feels low.

2.7

As we also thought that the sample size of 30 seems low at the first look, we have thoroughly searched pre-existing literature that looks at both the number of groups (countries) vs. the number of sample size. Maas and Hox (2005) and Snijders (2005) both give recommendations that the number of groups is way more important to increase the power of MLM than sample size. Both of the articles suggested that even if the sample size in a group is really small (< 5), that group should be still included in the analysis.

The exclusion criteria may be too strict. Why discard partial responses?

2.8

We agree with you that it is desirable to keep as many respondents in our data as possible. The fact that we exclude countries with less than 30 participants is a statistical decision (see section 6 and 10). Further, we decided to exclude participants with repetitive and unrepresentative answers to ensure quality. As we did not include attention checks, we think that this step is necessary. Overall, we are now more explicit in pointing out the need for these "strict exclusion criterias".

I didn't see any discussion of how these countries were selected (e.g., feasibility; connections) nor of the representativeness of those countries.

2.9

In section 7, we now provide more details about the selection and representativeness of the countries in our sample.

Is alpha = .05 appropriate given the number of inferential tests? I wasn't clear whether the hypotheses would be tested within each country. If so, some adjustment for multiple comparisons seems appropriate, either by lowering alpha or using Bonferonni-Holm, etc.

2.10

We intend to conduct MLM for all hypothesis tests. Thus, the multiple comparison correction is not necessary since we will conduct only one hypothesis test per hypothesis; hypotheses will not be tested within each country.

Yes, although not much detail was given about recruitment techniques (e.g., email; social media), materials (advertisements), or target populations (e.g., snowballing in student samples?).

2.11

We now give some more details about the recruitment in section 7.

I think so. It is unclear how the authors would go about interpreting a null effect for one of the hypothesized relationships; there aren't many positive controls.

2.12

Thank you very much for your comment on interpreting null results. Although we do not set any control condition in our study, as stated in the summarization table, we intend to use Bayesian inference in addition to frequentist inference to interpret results. According to Morey and Rouder (2011), it is possible to examine whether the null hypothesis is the case with the resultant Bayes Factor. We plan to use both p -values as well as Bayes Factors in our decision-making process, so by doing so will enable us to examine whether the null hypothesis is supported by evidence and is likely to be the case. More specifically, we intend to investigate whether the resultant Bayes Factor is smaller than $\frac{1}{3}$, which indicates evidence at least positively supports the null hypothesis (null effect) against the alternative hypothesis (presence of non-zero effect).

Broadly, yes. It was not always clear to me how the MLM would be constructed, e.g., for which analyses the country level would be included.

2.13

MLM will be conducted to test the hypotheses 2-5, and individual responses will be nested within each country, so that we can take into account the variability in each DV for both the individual level and country level.

Comments to Reviewer 3

Thank you for evaluating our manuscript and providing detailed feedback.

It is understandable that with a study of events as they are unfolding, it is difficult to have very specific predictions. The predictions listed by the authors are relatively general but I find this appropriate in the current context. I suspect that the descriptive statistics might be very informative in their own right.

3.1

Thank you for acknowledging this. We hope so, and view the descriptives as a central contribution.

In the comparison across countries, it might be useful to have some measure of the stage of the pandemic that each country is in at the time of the survey, as this is likely to impact the responses. For example, if the situation in a country has stabilized somewhat (at least optically) people are likely to be less anxious than people who live in countries that are in earlier stages of the pandemic.

3.2

We agree. This paper is the first of several that will be coming out of our consortium. Presently we wish to take advantage of the rapid response scheme to share a one to two week picture. Once more data has been collected, a time perspective will be forthcoming, which will also attempt to incorporate the various stages of COVID-19 related developments/interventions in different countries.

The exclusion criteria seem reasonable, both at the country level and at the respondent level. The authors might consider including an attention check, of which there are many examples in the literature.

3.4

We agree that it is important to ensure the quality of survey data with measures such as attention checks or time-related exclusion of respondents. As the survey was very long, we decided to not include attention checks, but to rely on other checks to ensure data quality (see section 9). Unfortunately, as the survey has already started, it is not possible to add attention checks.

The proposed analyses use both frequentist and Bayesian statistics. The authors specify the manner in which they will interpret these statistics, both separately and in conjunction.

3.5

We created a table (Table 2) that summarizes how we will interpret the resultant p-values and Bayes Factor values in conjunction. We also added a brief explanation about how those two indicators will be used to improve the completeness of our analyses.

Comments to Reviewer 4

Thank you for evaluating our manuscript and providing detailed feedback.

The proposed hypotheses are perfectly reasonable. My only point is that it is not entirely clear how the country-level stress variable, which is first mentioned in H5 and the model specification section on page 15, is (1) aggregated and (2) why it appears only in H5. My suggestion would be to also control for the country differences observed in hypothesis (H1), at least as an additional analysis/check. We need to see whether the country differences on stress levels might affect the overall effects of concern on compliance for example. If you already map group differences (in this case- country differences), if these differences exist, then you need to control for them.

4.1

Thank you very much for your suggestion. We recognise that the issue you raised is indeed important: our first attempt was also to model country-level stress and examine its contextual effects. However, due to the underpower issue of the present study to detect the effect of country-level predictors (see Reviewer 5's comment 11 and our responses for further details regarding the power issue), we decided to remove the country-level stress from the model in order to adopt a more parsimonious model that corresponds to the relatively small country size and to increase the consistency between the hypotheses and analytic plan. In addition, we include country-level factors, such as the population size, GDP, etc., in our models. These factors are included to control for differences in the aforementioned factors across different countries, and are employed to control for the variability residing at the country level.

Some of the questionnaires included in the surveys are validated in the respective languages and some not, some have been validated through back-translation and some not. This will need to be clearly acknowledged, together with the limitations in interpreting the results. Having said this, this is an inevitable downside in the current context. In addition, it benefits from being shared across the world and these shortages may be overcome by sheer statistical power.

4.2

We agree that your point is a limitation of our project. As you suggested, we will explicitly point out this limitation and its implications (e.g., interpretation of results) in our discussion.

My understanding is that the data will be open access, and therefore available for anyone to replicate. Ideally, the authors should provide their code.

4.3

Exactly! All our data and code will be publicly available, as will cleaning codebooks for raw data.

The authors provide a detailed description of the methods and analysis approach. They also mention “exploratory cross-country comparisons”. Given the extraordinary circumstances, I would personally support exploratory approaches too, alongside hypothesis driven ones.

4.4

Thank you for stressing this - we very much agree, and intend for descriptive comparisons to take up a good part of the final paper. Unfortunately following the RR stage 1 template, we couldn't not specify explorations. Only hypotheses.

This is one reason why we have selected relatively simple hypotheses from the outset.

I appreciate the power analysis which the authors conducted; however, the strength of this project will lie in its numbers, especially to allow for cross country comparisons. So, I would encourage that subsequent analyses include more cases. This is especially important when needing to control for a good number of confounds, including between-country differences on the DVs and IVs of interest. Having said this, I see that results are needed quickly to respond to current efforts and using a across-country approach will increase the study's power. Therefore, I would say that the authors have considered sufficient outcome-neutral conditions, although a clear acknowledgement of the interim nature of the analysis should be stated.

4.5

As stated earlier, we will try our best to include as many countries as possible, but we also acknowledge that the major limitation of current study lies in the insufficient number of countries to test the direct effects of country-level predictors.

P. 2. Line 21: “numerous universities” – requires a footnote, or a reference to the list of universities

4.6

Good point. We added a reference.

Line 32: “in order to understand how different government strategies succeed or fail” – we can tell if strategies succeed or fail if the number of cases drops. From this analysis, we will understand if the strategies are successful in achieving behaviour change (not whether strategies are successful in general – that's a different kind of study). Just a clarification needed here.

4.7

Thank you. We have clarified this in the new version of the manuscript, and focus on calling attention to the apparent complexity emerging from the literature.

Line 51: having completed the questionnaire myself, I'm wondering if “psychological stress” is the best choice of words; it looks like this should be deconstructed into the constructs that

the questionnaire is actually measuring: social support, attitudes, state anxiety, acute stress. Although I see the need to use an overarching concept, it would be good to have clarification on the constructs measured from the very beginning.

4.8

The main DV used for the hypotheses tested here is the Perceived Stress Scale (PSS-10), which is a state-level generalized measure. As this is only the first publication, other elements are intended for more specific analyses as the dataset matures. We have updated the title and literature section to make relationships more clear.

Line 57: "a number of variables" -> clarify briefly what kind of variables ("a number of variables measuring a, b, c, d")

4.9

We agree that if we write "a number of variables", we need to specify these. As it seems not to be necessary to specify all our variables in this paragraph, we re-worded the sentence in a way that we do no longer write about our set of variables (see section 3).

P. 3. Line 37: is "interact" the right word? Are the factors mediating or moderating the effect?

4.10

We revised the part using the term "interact." In fact, we intended to mean "association" between the factors not either "moderation" or "mediation." To avoid any unclarity, we changed the term so that it does not imply either moderation or mediation.

Line 43: was there a lower level of compliance? Also lower compared to what?

4.11

This no longer appears in the revised text.

Line 54: from the previous paragraph we already understand why this study is necessary; this paragraph goes into some detail that may be somewhat redundant and a bit far-fetched. For example: mentioning impaired cognitive activity and citing a couple of papers studying arctic explorers may not be appropriate (there are many other reasons for cognitive impairments there that do not apply to one isolating at home). Also the survey doesn't measure cognition.

4.12

We agree. The section has been re-written and e.g. the "cabin fever" elements etc. omitted.

Line 57-58: slight oversimplification of the complex causes of stress; this needs to be rephrased to say that some of the causes for stress which have been reported in these papers are related to profession and family.

4.13

We agree with this observation, and have changed the language to reflect this.

Last paragraph on page 3 – cont. page 4: needs to be rephrased and simplified to outline some of the key concepts (e.g. social support) that the survey is measuring and the evidence behind it that motivated inclusion of these questionnaires.

4.14

Sections updated to include SPS int. stress buffering effects.

P. 4 Line 16 – 21 – please break the sentence down for clarity of the message.

4.15

We have rephrased this sentence for clarity.

Line 23: “will help identify some of the psychological risk factors”

4.16

We have rephrased this during revisions.

Line 48-54 – please break the sentence down for clarity of the message.

4.17

We have rephrased this and re-organised the paragraphs for clarity.

Page 56 – “agree-disagree”: Likert? How many points?

4.18

Thank you for pointing this out. We removed the sentences about the questions concerning media consumption as it is not relevant for our research. As a result, we do not need to specify anymore with what kind of Likert Scale we were measuring.

Page 58-59: the paper will need to list where a validated measure was used or not. And acknowledge this limitation if appropriate.

4.19

We agree with you. The table of variables shows which measures are validated and which ones are not. Of course, we will acknowledge this limitation in our discussion.

P. 5 Line 17: can the validation data be included, including test-retest values in supplemental materials? Also, an acknowledgement of the surveys where this was not possible, and the consequent limitations to the interpretation of results will need to be clearly stated.

4.20

We will provide a translation and validation protocol as supplemental material. This will be done as soon as possible but latest with the publication of our project. We will also acknowledge related limitations.

Line 48: why mean and not sum? Same for page 6, line 22 (and other questionnaires where mean is chosen over sum);

4.21

Good point, we use sum instead of mean scores for those questionnaires where researchers commonly build sum instead of mean scores (see section 3).

P. 9 Line 4: If the authors will include 30 observations / country when potentially much more data will be collected – then this is an interim analysis, which should be acknowledged as such.

4.22

A paragraph is added in the Section 6

The authors have not mentioned the number of countries that they aim/hope to include. Is there a minimum? They mention that this depends on the inclusion/exclusion criteria, but can there be too few countries who meet these criteria?

4.23

We will follow the suggestions by Maas and Hox (2005) and Snijders (2005), and try to include as many countries as possible to increase the power of MLM. The number of countries at maximum may be 60.

Missing data: need to state if observations are excluded based on missingness and if yes, what is the cut-off on the number of items left blank within the questionnaire

4.24

In section 8 and 9 we now provide a detailed description on how we deal with respondents that have "a large number" of items left blank within the questionnaire (including our definition of "a large number"). For our analyses, we will include all respondents of the final data set where answers for the variables of interest are present.

Page 10, line 46 – explanation will be needed on how variables are aggregated

4.25

We are now more clear about our procedure and refer to section 3 for details about our variables.

Comments to Reviewer 5

Thank you for evaluating our manuscript and providing detailed feedback.

The biggest issue is that the authors do not plan to test for measurement invariance before proceeding to their focal analyses. A lack of measurement invariance can compromise the validity of conclusions. I recommend the use of multiple group models using SEM to test invariance. There are several resources I can recommend to learn about this technique if necessary: Chen (2008, JPSP) and the Kline SEM textbook are good introductions.

5.1

Based on other reviewer comments, we added a plan to perform measurement invariance tests by comparing configural invariance, metric invariance, and scalar invariance models in the quality check section (as per Milfont & Fischer, 2010). We also elaborate what we will do for our planned analyses once the equivalence of invariance is not achieved. We are aware that this is not equal to SEM but should give some of the same insights.

The authors state “Continuous variables are centred on grand means, or converted to a z-score for the convenience of interpretation” (p. 10). Centering is a good idea; standardizing (converting to a z-score) is a bad idea in multi-level models (because you disturb the variance structure across levels). Avoid standardizing data in multilevel analyses. Rely on conversions to quasi-r-squared (or quasi-r) to interpret effect sizes. Effect sizes should absolutely be interpreted alongside p-values.

5.2

As we are aware of the issue you raised, the MLM analyses for power analysis were run twice with grand-mean centering vs. standardisation of the continuous variables. Only estimated fixed effects were reported based on the modeling with standardised variables, whereas the random effects were estimated with grand-mean centering. We added a note about this rationale in Section 6

The authors have clear associations that they plan to test. At least as the introduction is currently written, these tested associations do not appear to stem from any particular theory or model. I don't necessarily have an issue with this, but the tested models are not confirmatory in the sense that they test a theory-driven hypothesis. The introduction could benefit from a model or framework if there is one. I.e., why test these associations and not other models?

5.3

This paper is the first of several that will be coming out of our consortium. Presently we wish to take advantage of the rapid response scheme to share the current picture of experienced stress and its - very probably not unidirectional - relationships to trust in preventive measures. As such, the intention behind the hypotheses is more exploratory than tied to any particular model. Following your advice we have rephrased several passages as well as the title to reflect this more clearly, and re-written the literature section.

The authors intend to model a variety of demographic country level variables. I wonder if there are other pandemic-specific country level variables that should be included – e.g., rate of country-level hospital bed or protective equipment shortages, date of lockdown onset (or presence of lockdown), previous pandemic incidence. Further, I wonder if time of data collection should be explicitly modeled (e.g., days from date of survey to onset of lockdown). I don't have strong feelings about this, but wanted to make the suggestion.

5.4

We acknowledge that the proposed ideas are indeed important for current pandemic situation. After we have thoroughly evaluated the goals of the present study and their compatibility with the proposed control variables, we decided to keep the original design, as we believe medical resources may have less to do with compliant behavior.

My preference would be to see all models with and without controls included. The authors could present their preferred specification (as presented here) and contrast it with models without level 1 and level 2 controls.

5.5

We appreciate your comment regarding the comparison of models with vs. without control variables. We intended to compare five different models (please see our response to your comment #6 below), so this point could also be addressed simultaneously. Once we compare Model 3 (model with fixed effects) vs. Model 4 (Model 3 + control variables), then we will be able to address the point that you raised.

Hypotheses 1a to 1d should be tested with multi-level models (not ANOVAs). The reason is that country should be a random intercept, not a fixed effect. Hypotheses 2-5 should include and test random slopes for the predicted effects. For all hypotheses, I recommend a series of nested models: a null model (no predictors, no random effects), a random intercept only model (country as random intercept), a model with fixed effects of predictors, a model adding control variables, a model with random slopes for predictors. Each model can be compared to its predecessor via a likelihood ratio test. Strive to specify maximal models, but trim random effects (usually random slopes) if there are convergence issues.

5.6

Thank you very much for your suggestion regarding the inclusion of control variables in MLM. Unfortunately, we decided to perform ANOVAs to test H1. Because we are mainly interested in comparing the mentioned dependent variables between countries, we found that ANOVA is more theoretically relevant than MLM. Also, to perform post-hoc tests to examine international differences in the DVs, employing ANOVA would be necessary. Moreover, technically, adding countries as a fixed effect may create the difficulty to have models converged.

We appreciate your suggestion regarding the model comparison for H2-H5. We agree with you that comparing different nested models would be informative in identifying the best model to predict the outcome variables of interest. Thus, we revised our analysis plans so

that they include some procedures for the model comparison. Following your suggestion, we intend to compare these five models:

- Model 1: Null model (no predictors or random effects added).
- Model 2: Random intercept-only model (Model 1 + country as a random intercept).
- Model 3: Model with fixed effects of predictors (Model 2 + fixed effects).
- Model 4: Model with control variables (Model 3 + control variables).
- Model 5: Full model (Model 4 + random slopes for predictors).

We plan to perform a likelihood ratio (LR) test to examine whether the addition of random effects significantly improves the model (Model 1 vs. Model 2). For other comparisons, i.e., Models 2 vs. 3, Models 3 vs. 4, Models 4 vs. 5, we will employ other methods, such as the pseudo R^2 comparison and/or omnibus F test because the SAS macro allows the use of the LR test only for the comparison of models with vs. without random effects.

Plan to report descriptive statistics for all variables (means, standard deviations, alpha or omega reliability, intercorrelations between variables). I don't know these variables well enough to anticipate whether there might be problems with multi-collinearity of predictors in the planned models. Anticipate and plan for dealing with multi-collinearity if necessary.

5.7

We added that we will provide descriptive statistics for all variables. Although we do not expect to have problems with multi-collinearity, we will compute and check VIFs. We have added this procedure including our approach to deal with potential multi-collinearity in section 10.

It's not clear in the proposal how country level variables will be scaled. That is, will they be treated as continuous? Contrast coded categories? For individual level predictors, I do wonder if the authors should plan to collapse educational categories (perhaps to just three: did not finish compulsory schooling in that country; finished secondary schooling; university degree or higher). If you choose to do this, I would contrast code the categories rather than treat them as continuous.

5.8

We added the necessary information to the table in section 3.

It's not clear why for hypothesis 1c, the focus is on OECD_Institutions item 1, rather than all OECD_Institutions items. Should these items be combined into a total score if possible? Either way, would be good to have more reasoning behind this choice.

5.9

Good point! We now provide more details about the variable in section 3.

Some variables are measured with single items that may have limited reliability/validity. This is unavoidable given that we cannot alter the design at this point. Just be aware that this may further compromise statistical power of some analyses.

5.10

Thank you for pointing this out. We will keep this in mind and discuss it as limitation.

Regarding power, the authors state “we plan to recruit at least 30 participants per country so as to detect both the effects of individual- and country-level predictors” (p. 9). They cite Arend and Schäfer (2019)’s Table 8 as justification. They note that in the ESS database, medium ICCs, small to medium effects of level 1 and level 2 predictors, and small cross-level interactions are observed. They expect to collect data from 44 countries (level 2 units). The level 1 sample size (individuals per country) could vary. Of the 10 hypotheses, four focus on the ICCs themselves, five focus on level 1 predictors, and one contains an interaction between two level 1 variables. In six of the hypotheses, level 2 control variables are included. Based on Arend and Schäfer (2019)’s Table 8, with 44 countries and a sample size of 30 per country, small level 1 effects are underpowered (medium and large level 1 effects are adequately powered), regardless of the ICC. Only large effects of level 2 variables are adequately powered (regardless of ICC); small and medium effects are underpowered. Only large cross-level interactions under conditions of large random slopes are adequately powered; most random slopes are underpowered. Adding additional participants per country seems to matter less than adding more countries (or more broadly, level 2 units). Thus, it seems unavoidable that the current attempt will be underpowered in some ways, mostly because there is likely not an easy way to recruit more countries into the study. Having 30 participants per country seems reasonable based on the power tables, but the authors may run into issues with convergence with this size of sample per country. In particular, the multi-group invariance models tend to do better with larger per group sizes. I don’t see a strong reason to cut data collection short; it would seem prudent to get as much data as possible while the pandemic is happening. The results of the study are not themselves time sensitive (no government is going to base decision making on what these results show). Therefore, I would err on the side of sampling strategy that allows the collection of as much data as possible.

5.11

We agreed with your comment on the underpower issue. Therefore, the scope of the present study will focus on modeling the structure of individual responses nested within countries, and examining our proposed hypotheses with respect to psychological factors at the individual levels (rather than the country-level predictors). Furthermore, as we recognised the issue with convergence, we intend to conduct the multilevel modeling analyses as follows:

1. IF our planned analyses can be conducted without any issue (e.g., failed convergence) with the dataset that passed the measurement invariance test, THEN report the findings.
2. IF 1 failed, THEN conduct the planned analyses with the dataset that includes all countries with $N \geq 30$ (according to the country-level exclusion criteria).

If 1 failed and findings from 2 are reported, we will discuss limitations and caveats regarding the interpretation of the findings in the discussion section.

What is the rationale for the 15% cutoff for identifying floor/ceiling effects? Why not 10 or 20? Seems arbitrary. The exclusion rules for fast response time and Mahalanobis distance seemed appropriate and a good practice to improve data quality.

5.12

We agree that the cutoff is a bit arbitrary. Indeed, we had the very same discussion until we settled on 15% based on this work, that suggested the use of the 15% threshold:

McHorney, C. A., & Tarlov, A. R. (1995). Individual-patient monitoring in clinical practice: are available health status surveys adequate?. *Quality of Life Research*, 4(4), 293-307.

Causal language: There is causal language in the title of the paper (“the impact of”) and sprinkled throughout the paper itself. Causal language is inappropriate for this study’s methods.

5.13

We have softened the causal language in various places, including hypotheses and the title.

Appendix D**ROYAL SOCIETY
OPEN SCIENCE****Trust and compliance with preventive measures in the 2020
coronavirus pandemic: relationships to trust, stress and
worry**

Journal:	Royal Society Open Science
Manuscript ID	RSOS-200589.R1
Article Type:	Registered Report - Stage 1
Date Submitted by the Author:	05-May-2020
Complete List of Authors:	Lieberoth, Andreas; Aarhus University, School of Culture and Society (Interacting Minds Center); Aarhus University, Danish School of Education (DPU) Lin, Shiang-Yi; Hong Kong Institute of Education Stöckli, Sabrina; University of Bern Han, Hyemin; University of Alabama Chrona, Stavroula; King's College London Kowal, Marta; Wroclaw University Institute of Psychology, Gelphi, Rebekah; University of Toronto, Department of Psychology
Subject:	health and disease and epidemiology < BIOLOGY, psychology < BIOLOGY
Keywords:	COVID-19, preventive measures, worry, stress, compliance behavior
Subject Category:	Psychology and cognitive neuroscience

*Aarhus, May 2020*

**Response to Reviewers — RSOS-200589**

Dear Prof. Chambers and Prof. Dunn,

It is with pleasure that we submit to you a revised version of our manuscript. We appreciate
your feedback and the feedback from five reviewers. We believe that the revision is much
stronger than the previous manuscript.

In the new version, we have updated the title and literature review to address reviewer
comments, and adapted the hypotheses to a more correlational language. Also, to more
systematically present how we will interpret findings based on both p -values and Bayes
Factors, we summarized our decision-making process in a table (see Tables 2 and 3 in the
RR).

As also stressed by Reviewer 3 and 4, cross-country descriptives and more exploratory
analyses will constitute a large part of the finished article. Even though we cannot specify
these in the stage one RR, we would like to underscore their centrality to the final
manuscript, and the hypotheses as a route to creating this final broad overview.

As called for by reviewer two, we have softened the language in the title and hypotheses to
one of correlation rather than unidirectional causality.

While several reviewers have suggested adding additional country level variables to our
analyses, we stress that this is the first publication to come out of the COVIDiSTRESS global
survey, which we intend to showcase data from as many participating countries as possible.
However, because of the relatively small number of countries (approximately 50-60) we can
include in a multilevel model, our study would be underpowered to detect both direct effects
of country-level predictors and cross-level interactions. Therefore, we will focus on
hypotheses examining individual-level relationships, as stated in our hypotheses 2-5. To this
point, we appreciate the reviewers' comments that invite detailed exploratory analyses and
illustrations of between-country differences.

Following your advice, we also addressed some methodological shortcomings and more
explicitly discussed the methodological limitations of our research. Below, please find the
detailed response to the reviewers' individual remarks.

Thank you very much in advance for considering the revision of our manuscript. We are
looking forward to hearing from you.

Sincerely,

The COVIDiSTRESS global survey group

Comments to Reviewer 1

Thank you for evaluating our manuscript and providing detailed feedback.

As we wrote in the abstract of an article on this topic, “When comparing groups an assumption is made that the instrument measures the same psychological construct in all groups. If this assumption holds, the comparisons are valid and differences/similarities between groups can be meaningfully interpreted. If this assumption does not hold, comparisons and interpretations are not fully meaningful. The establishment of measurement invariance is a prerequisite for meaningful comparisons across groups.” (Milfont & Fischer, 2010).

There are three main levels of invariance testing: configural (underlying factor model identical across groups), metric (item loadings identical across groups), and interval (item intercepts identical across groups). The specialised literature argues that metric invariance is needed for examining associations between constructs across groups, and that mean comparisons across groups are only meaningful when interval invariance is achieved. All hypotheses in the Registered Report focus on mean comparisons or associations between constructs across cultural groups but no invariance testing is proposed. This needs to be addressed! Without testing and confirming the multi-item measures are equivalent across the groups, the testing of all proposed hypotheses are meaningless. For example, if participants across the considered nations do not respond to items of the Perceived Stress Scale similarly, then comparing stress means across countries (H1a) can't be meaningfully interpreted.

When revising the Registered Report to include measurement invariance, I suggest the authors to explore the alignment method, which is a much easier implementation of invariance testing – particularly for large groups as in this case. I believe alignment can already be implemented in R. I recommend two main publication on the topic: Asparouhov and Muthén (2014) and Marsh et al. (2018); and we have also implemented this approach in a recent article (see Milfont et al. 2018) and have discussed measurement invariance for replicating findings across cultural groups (see Milfont & Klein, 2018). I am sure the authors would know measurement invariance experts in their networks but I can make myself available to provide some guidance if needed (and if the Editor and authors feels this is appropriate).

1.1

First of all, we would welcome the Reviewer's collaboration, as, based on the suggested literature overview, we believe that the Reviewer could significantly contribute to the invariance analyses (if the Editor allows).

We have also extended the information about planned analyses of the equivalence of invariance in the Quality checks section. Based on the comment, we now intend to investigate the cross-national equivalence of the Perceived Stress Scale (PSS-10, see Cohen et al., 1983; Cohen & Williamson, 1988) prior to analysis. Using a multi-group factor analysis we will compare the models assuming the two-factor structure (positive and negative, with the latter consisting of reversed items; Chaaya et al., 2010; Roberti et al., 2006) across all countries (configural invariance), with a model with factor loadings and latent correlations constrained to be equal (metric invariance), and items' intercepts to be the same in all groups (scalar invariance). When evaluating the model fit, we will rely on the

usually applied criteria (Hu & Bentler, 1999), in which a Comparative Fit Index (CFI) and
Tucker Lewis Index (TLI) above .90 would indicate adequate fit, whereas a standardized root
mean square residual (SRMR) below .06, and a root mean square error of approximation
(RMSEA) below .08 indicate no misfit. When evaluating the measurement equivalence, we
will compare the configural invariance model with the metric invariance model, and then the
metric invariance model with the scalar invariance model (Milfont & Fischer, 2010). As these
models are characterized by a growing complexity (each subsequent model is nested within
the previous one), while assessing models' superiority, we will rely on cut-off criteria
recommended for testing measurement invariance: a change of *CFI* (ΔCFI) less than .01
($\Delta CFI < .01$), a change of *RMSEA* of less than .015 ($\Delta RMSEA < .015$), and a change of
*SRMR* less than .01 ($\Delta SRMR < .01$) indicating that two compared models do not differ in
terms of model fit (Chen, 2008; Cheung & Rensvold, 2002). However, in case that the
equivalence of invariance will not be achieved, based on recommendations indicating that
partial invariance may allow for reasonable comparisons (see e.g., Byrne, Shavelson, &
Muthén, 1989), we will also estimate models with partial invariance.

In addition, in the revised RR, we elaborated how we will conduct the planned analyses after
performing the measurement invariance test and if some portion of the dataset should be
excluded due to the failed measurement invariance test:

Given that the number of countries to be analysed directly influences the power of the
planned multilevel modeling, we intend to conduct the multilevel modeling analyses as
follows:

IF our planned analyses can be conducted without any issue (e.g., failed convergence) with
the dataset that passed the measurement invariance test, THEN report the findings.

IF 1 failed, THEN conduct the planned analyses with the dataset that includes all countries
with $N \geq 30$ (according to the country-level exclusion criteria).

If 1 failed and findings from 2 are reported, we will discuss limitations and caveats regarding
the interpretation of the findings in the discussion section.

*I feel the authors should make it explicit from the outset that they will employ a multilevel*
*approach.*

1.2

Thank you for the suggestion. We have added a paragraph in the introduction stressing that
we use MLM analyses to test our hypotheses (see Section 1).

*The authors propose a list of country-level variables and use them as control. This is a*
*reasonable list of variables but there is no justification of their selection. This list can be*
*much longer, especially if the authors decide to include psychological dimensions of cultural*
*variability such as individualism/collectivism, tightness/looseness and relational mobility. It is*

*thus important to provide some rationale for selecting only non-psychological country-level*
*variables and for selecting these specific ones.*

1.3

I have added a paragraph explaining the justifications for the inclusion of both
individual-level and country-level control variables, following the hypotheses (see Section 4).

As this is a very large and varied dataset, psychological between-countries differences are
indeed planned to be the subject of a later analysis currently planned by other researchers in
our group. This upcoming analysis is also intended to take differences in COVID-19
responses by each government into account.

We have updated H5, to test the relationship between individual stress, social provisions and
compliance to behavior guidelines

*Related to the point above, the country-level variables are merely used as control variables*
*in the analyses. It would be great if these were included as predictors, at least for the first*
*four hypotheses. Once invariance is established and meaningful comparisons can be made*
*(see above), merely showing variability in stress, concern, trust and compliance might not tell*
*much, or might be difficult for the readers and government officials to interpret the observed*
*similarities/differences. It would be more interesting to show whether stress, concern, trust*
*and/or compliance is greater/weaker in wealthier/poorer countries, for example. (Again, the*
*list of variables could be very large and such analysis would not be attainable; by addressing*
*the point above and providing justification for a small list of variables selected, the authors*
*could provide such addition to the first hypotheses.)*

1.4

Please see comment for 1.3

*H3b reads “Across countries, trust in government efforts to slow the spread of coronavirus*
*will negatively predict concern over coronavirus, when controlling for individual- and*
*country-level covariates” but this negative prediction is not obvious to me. Greater concern*
*should lead to stronger trust, no? Or alternatively, if I don't trust the government efforts (take*
*delayed and denialistic actions by the USA and Brazil for example), I would be more*
*concerned about the pandemic, no?*

1.5

*We see how the hypothesis is phrased a bit backwards. The idea is, that if you believe that*
*the government measures will protect you, you will be less worried.*

*We have conducted further literature searches, and found some support for this quite*
*intuitive notion (literature section updated), but interestingly not much has been written. It is*
*going to be an interesting hypothesis to test.*

*I found H5 unclear. Do you mean that provisions would reduce/buffer the expected negative*
*effect of stress on compliance (H2a)? You need to spell out. Given the relations between*
*stress and concern (see H2a, H2b and H4), I believe another moderation hypothesis is*
*reasonable and could be included by examining the extent to which provisions*
*increases/enhances the expected positive effect of concern on compliance.*

1.6

Thank you for the suggestion. We have strengthened the relevant arguments in the
introduction section, and make it clear that we examine the buffering effect of social
provisions on the negative impact of stress on compliance (see Page 3 in Section 1).

*I have made comments in the original PDF. I have highlighted sentences that are unclear of
miss words, and have added missing words too (e.g., provide clear direction for some of the
hypothesis).*

1.7

We would like to thank the Reviewer for their helpful comments. We have changed the
sentences and added missing words.

Comments to Reviewer 2

Thank you for evaluating our manuscript and providing detailed feedback.

The introduction does a good job of setting the overall context of the work and why it is important, but a less good job of why these particular variables are being measured and what they will enable tests of. It is also unclear why the data is being collected over time: the planned analyses didn't seem to include time.

2.1

We have adjusted the introductory section accordingly. The text is now focused more on the specific variables we are planning to include in our analysis, and the complexities that emerge from that combined space. We believe that the link between the introductory part and the subsequent sections is now clearer. This paper is the first of several that will be coming out of our consortium. Presently we wish to take advantage of the rapid response scheme to share the current picture. Once more data has been collected, a time perspective will be forthcoming, but since it is difficult to project the future trajectory of COVID-19 related developments, and thus a pragmatic stop-rule for data collection overall, this analysis will be better suited as a retrospective analysis, using data from the coming months.

The introduction is a little disorganized; for example, the narrative started with how this research could inform policy making and behavior change communications, and then pivoted to mental health outcomes, and then back to behavior change.

2.2

We have edited the introduction to lead more clearly into the analysis.

There are many similar survey projects being launched right now. I've heard of several consortia. I was surprised to see no mention of this similar research, e.g.: <https://wintoncentre.maths.cam.ac.uk/news/how-different-countries-are-reacting-to-the-covid-19-risk-and-their-governments-responses/> data: <https://osf.io/jnu74/>. At the least, it would be useful to articulate how the current effort differs from existing results and from ongoing, related surveys.

2.3

Thank you for pointing this out. We now refer to other projects and emphasize that our project is unique in the sense that it is a global survey relying on organic sampling rather than e.g. purchased responses from predetermined subject pools (see page 1).

To the extent that the intro focuses on behavior change, it would be useful to spend more time explaining the context of previous interventions (e.g., but not limited to nudges). Also, specifically on COVID-19 this report examines behavior change to different messages: <https://www.bi.team/blogs/bright-infographics-and-minimal-text-make-handwashing-posters-most-effective/> This tracker of cross-governmental responses (e.g., lockdown severity) may be

1
2
3 *useful to integrate:*

4 <https://www.bsg.ox.ac.uk/research/research-projects/oxford-covid-19-governmentresponse-tracker>

7 2.4

8 This paper is the first of several. The tracker especially will be extremely useful in the future.

*RQ1/H2A/B assume the causal direction of (+) trust  (-) stress / (-) concern. Those are plausible, but so would be the reversed arrows. Unless the longitudinal data will be used with, e.g., HLM to test for both causal directions, the causal language should be softened and this caveat stated explicitly for all cross-sectional data. The same point applies to H4.*

2.5

We agree. The intention was not to suggest only one direction of causality. We have
softened the language to one of correlation.

Colleagues from the consortium hope that causal relationships can be studied further down
the line, as longitudinal data becomes available from the dataset.

*For H5, it might be useful to connect this directly to the famous stress buffering hypothesis from health psychology, whether or not this particular ref (18000 citations??):*
*<https://psycnet.apa.org/record/1986-01119-001>. I found this moderation underjustified*
*theoretically, for example compared to an alternate claim like (+) social provisions acting as*
*an independent, direct predictor of (+) compliance.*

2.6

As per Comment 1.6, we have strengthened the rationale of the moderating effect of social
provisions in the introduction section.

*I didn't deeply look into the estimated sample size needed per country, re: Arend and Schäfer, but n = 30 feels low.*

2.7

As we also thought that the sample size of 30 seems low at the first look, we have
thoroughly searched pre-existing literature that looks at both the number of groups
(countries) vs. the number of sample size. Maas and Hox (2005) and Snijders (2005) both
give recommendations that the number of groups is way more important to increase the
power of MLM than sample size. Both of the articles suggested that even if the sample size
in a group is really small (< 5), that group should be still included in the analysis.

*The exclusion criteria may be too strict. Why discard partial responses?*

2.8

We agree with you that it is desirable to keep as many respondents in our data as possible.
The fact that we exclude countries with less than 30 participants is a statistical decision (see
section 6 and 10). Further, we decided to exclude participants with repetitive and
unrepresentative answers to ensure quality. As we did not include attention checks, we think
that this step is necessary. Overall, we are now more explicit in pointing out the need for
these "strict exclusion criterias".

*I didn't see any discussion of how these countries were selected (e.g., feasibility;*
*connections) nor of the representativeness of those countries.*

2.9

In section 7, we now provide more details about the selection and representativeness of the
countries in our sample.

*Is alpha = .05 appropriate given the number of inferential tests? I wasn't clear whether the*
*hypotheses would be tested within each country. If so, some adjustment for multiple*
*comparisons seems appropriate, either by lowering alpha or using Bonferonni-Holm, etc.*

2.10

We intend to conduct MLM for all hypothesis tests. Thus, the multiple comparison correction
is not necessary since we will conduct only one hypothesis test per hypothesis; hypotheses
will not be tested within each country.

*Yes, although not much detail was given about recruitment techniques (e.g., email; social*
*media), materials (advertisements), or target populations (e.g., snowballing in student*
*samples?).*

2.11

We now give some more details about the recruitment in section 7.

*I think so. It is unclear how the authors would go about interpreting a null effect for one of the*
*hypothesized relationships; there aren't many positive controls.*

2.12

Thank you very much for your comment on interpreting null results. Although we do not set
any control condition in our study, as stated in the summarization table, we intend to use
Bayesian inference in addition to frequentist inference to interpret results. According to
Morey and Rouder (2011), it is possible to examine whether the null hypothesis is the case
with the resultant Bayes Factor. We plan to use both p -values as well as Bayes Factors in
our decision-making process, so by doing so will enable us to examine whether the null
hypothesis is supported by evidence and is likely to be the case. More specifically, we intend
to investigate whether the resultant Bayes Factor is smaller than $\frac{1}{3}$, which indicates
evidence at least positively supports the null hypothesis (null effect) against the alternative
hypothesis (presence of non-zero effect).

*Broadly, yes. It was not always clear to me how the MLM would be constructed, e.g., for*
*which analyses the country level would be included.*

**2.13**

MLM will be conducted to test the hypotheses 2-5, and individual responses will be nested
within each country, so that we can take into account the variability in each DV for both the
individual level and country level.

Comments to Reviewer 3

Thank you for evaluating our manuscript and providing detailed feedback.

It is understandable that with a study of events as they are unfolding, it is difficult to have very specific predictions. The predictions listed by the authors are relatively general but I find this appropriate in the current context. I suspect that the descriptive statistics might be very informative in their own right.

3.1

Thank you for acknowledging this. We hope so, and view the descriptives as a central contribution.

In the comparison across countries, it might be useful to have some measure of the stage of the pandemic that each country is in at the time of the survey, as this is likely to impact the responses. For example, if the situation in a country has stabilized somewhat (at least optically) people are likely to be less anxious than people who live in countries that are in earlier stages of the pandemic.

3.2

We agree. This paper is the first of several that will be coming out of our consortium. Presently we wish to take advantage of the rapid response scheme to share a one to two week picture. Once more data has been collected, a time perspective will be forthcoming, which will also attempt to incorporate the various stages of COVID-19 related developments/interventions in different countries.

The exclusion criteria seem reasonable, both at the country level and at the respondent level. The authors might consider including an attention check, of which there are many examples in the literature.

3.4

We agree that it is important to ensure the quality of survey data with measures such as attention checks or time-related exclusion of respondents. As the survey was very long, we decided to not include attention checks, but to rely on other checks to ensure data quality (see section 9). Unfortunately, as the survey has already started, it is not possible to add attention checks.

The proposed analyses use both frequentist and Bayesian statistics. The authors specify the manner in which they will interpret these statistics, both separately and in conjunction.

3.5

We created a table (Table 2) that summarizes how we will interpret the resultant p-values and Bayes Factor values in conjunction. We also added a brief explanation about how those two indicators will be used to improve the completeness of our analyses.

Comments to Reviewer 4

Thank you for evaluating our manuscript and providing detailed feedback.

The proposed hypotheses are perfectly reasonable. My only point is that it is not entirely clear how the country-level stress variable, which is first mentioned in H5 and the model specification section on page 15, is (1) aggregated and (2) why it appears only in H5. My suggestion would be to also control for the country differences observed in hypothesis (H1), at least as an additional analysis/check. We need to see whether the country differences on stress levels might affect the overall effects of concern on compliance for example. If you already map group differences (in this case- country differences), if these differences exist, then you need to control for them.

4.1

Thank you very much for your suggestion. We recognise that the issue you raised is indeed important: our first attempt was also to model country-level stress and examine its contextual effects. However, due to the underpower issue of the present study to detect the effect of country-level predictors (see Reviewer 5's comment 11 and our responses for further details regarding the power issue), we decided to remove the country-level stress from the model in order to adopt a more parsimonious model that corresponds to the relatively small country size and to increase the consistency between the hypotheses and analytic plan. In addition, we include country-level factors, such as the population size, GDP, etc., in our models. These factors are included to control for differences in the aforementioned factors across different countries, and are employed to control for the variability residing at the country level.

Some of the questionnaires included in the surveys are validated in the respective languages and some not, some have been validated through back-translation and some not. This will need to be clearly acknowledged, together with the limitations in interpreting the results. Having said this, this is an inevitable downside in the current context. In addition, it benefits from being shared across the world and these shortages may be overcome by sheer statistical power.

4.2

We agree that your point is a limitation of our project. As you suggested, we will explicitly point out this limitation and its implications (e.g., interpretation of results) in our discussion.

My understanding is that the data will be open access, and therefore available for anyone to replicate. Ideally, the authors should provide their code.

4.3

Exactly! All our data and code will be publicly available, as will cleaning codebooks for raw data.

*The authors provide a detailed description of the methods and analysis approach. They also*
*mention “exploratory cross-country comparisons”. Given the extraordinary circumstances, I*
*would personally support exploratory approaches too, alongside hypothesis driven ones.*

9 4.4

Thank you for stressing this - we very much agree, and intend for descriptive comparisons to
take up a good part of the final paper. Unfortunately following the RR stage 1 template, we
couldn't not specify explorations. Only hypotheses.

This is one reason why we have selected relatively simple hypotheses from the outset.

*I appreciate the power analysis which the authors conducted; however, the strength of this*
*project ill lie in its numbers, especially to allow for cross country comparisons. So, I would*
*encourage that subsequent analyses include more cases. This is especially important when*
*needing to control for a good number of confounds, including between-country differences*
*on the DVs and IVs of interest. Having said this, I see that results are needed quickly to*
*respond to current efforts and using a across-country approach will increase the study’s*
*power. Therefore, I would say that the authors have considered sufficient outcome-neutral*
*conditions, although a clear acknowledgement of the interim nature of the analysis should be*
*stated.*

27 28 4.5

As stated earlier, we will try our best to include as many countries as possible, but we also
acknowledge that the major limitation of current study lies in the insufficient number of
countries to test the direct effects of country-level predictors.

*P. 2. Line 21: “numerous universities” – requires a footnote, or a reference to the list of*
*universities*

38 4.6

Good point. We added a reference.

*Line 32: “in order to understand how different government strategies succeed or fail” – we*
*can tell if strategies succeed or fail if the number of cases drops. From this analysis, we will*
*understand if the strategies are successful in achieving behaviour change (not whether*
*strategies are successful in general – that’s a different kind of study). Just a clarification*
*needed here.*

50 4.7

Thank you. We have clarified this in the new version of the manuscript, and focus on calling
attention to the apparent complexity emerging from the literature.

*Line 51: having completed the questionnaire myself, I’m wondering if “psychological stress”*
*is the best choice of words; it looks like this should be deconstructed into the constructs that*

*the questionnaire is actually measuring: social support, attitudes, state anxiety, acute stress.*
*Although I see the need to use an overarching concept, it would be good to have clarification*
*on the constructs measured from the very beginning.*

4.8

The main DV used for the hypotheses tested here is the Perceived Stress Scale (PSS-10),
which is a state-level generalized measure. As this is only the first publication, other
elements are intended for more specific analyses as the dataset matures. We have updated
the title and literature section to make relationships more clear.

*Line 57: "a number of variables" -> clarify briefly what kind of variables ("a number of*
*variables measuring a, b, c, d")*

4.9

We agree that if we write "a number of variables", we need to specify these. As it seems not
to be necessary to specify all our variables in this paragraph, we re-worded the sentence in
a way that we do no longer write about our set of variables (see section 3).

*P. 3. Line 37: is "interact" the right word? Are the factors mediating or moderating the effect?*

4.10

We revised the part using the term "interact." In fact, we intended to mean "association"
between the factors not either "moderation" or "mediation." To avoid any unclarity, we
changed the term so that it does not imply either moderation or mediation.

*Line 43: was there a lower level of compliance? Also lower compared to what?*

4.11

This no longer appears in the revised text.

*Line 54: from the previous paragraph we already understand why this study is necessary;*
*this paragraph goes into some detail that may be somewhat redundant and a bit far-fetched.*
*For example: mentioning impaired cognitive activity and citing a couple of papers studying*
*arctic explorers may not be appropriate (there are many other reasons for cognitive*
*impairments there that do not apply to one isolating at home). Also the survey doesn't*
*measure cognition.*

4.12

We agree. The section has been re-written and e.g. the "cabin fever" elements etc. omitted.

*Line 57-58: slight oversimplification of the complex causes of stress; this needs to be*
*rephrased to say that some of the causes for stress which have been reported in these*
*papers are related to profession and family.*

4.13

We agree with this observation, and have changed the language to reflect this.

*Last paragraph on page 3 – cont. page 4: needs to be rephrased and simplified to outline*
*some of the key concepts (e.g. social support) that the survey is measuring and the*
*evidence behind it that motivated inclusion of these questionnaires.*

4.14

Sections updated to include SPS irt. stress buffering effects.

*P. 4 Line 16 – 21 – please break the sentence down for clarity of the message.*

4.15

We have rephrased this sentence for clarity.

*Line 23: “will help identify some of the psychological risk factors”*

4.16

We have rephrased this during revisions.

*Line 48-54 – please break the sentence down for clarity of the message.*

4.17

We have rephrased this and re-organised the paragraphs for clarity.

*Page 56 – “agree-disagree”: Likert? How many points?*

4.18

Thank you for pointing this out. We removed the sentences about the questions concerning
media consumption as it is not relevant for our research. As a result, we do not need to
specify anymore with what kind of Likert Scale we were measuring.

*Page 58-59: the paper will need to list where a validated measure was used or not. And*
*acknowledge this limitation if appropriate.*

4.19

We agree with you. The table of variables shows which measures are validated and which
ones are not. Of course, we will acknowledge this limitation in our discussion.

*P. 5 Line 17: can the validation data be included, including test-retest values in supplemental*
*materials? Also, an acknowledgement of the surveys where this was not possible, and the*
*consequent limitations to the interpretation of results will need to be clearly stated.*

4.20

We will provide a translation and validation protocol as supplemental material. This will be
done as soon as possible but latest with the publication of our project. We will also
acknowledge related limitations.

*Line 48: why mean and not sum? Same for page 6, line 22 (and other questionnaires where*
*mean is chosen over sum);*

4.21

Good point, we use sum instead of mean scores for those questionnaires where researchers
commonly build sum instead of mean scores (see section 3).

*P. 9 Line 4: If the authors will include 30 observations / country when potentially much more*
*data will be collected – then this is an interim analysis, which should be acknowledged as*
*such.*

4.22

A paragraph is added in the Section 6

*The authors have not mentioned the number of countries that they aim/hope to include. Is*
*there a minimum? They mention that this depends on the inclusion/exclusion criteria, but can*
*there be too few countries who meet these criteria?*

4.23

We will follow the suggestions by Maas and Hox (2005) and Snijders (2005), and try to
include as many countries as possible to increase the power of MLM. The number of
countries at maximum may be 60.

*Missing data: need to state if observations are excluded based on missingness and if yes,*
*what is the cut-off on the number of items left blank within the questionnaire*

4.24

In section 8 and 9 we now provide a detailed description on how we deal with respondents
that have "a large number" of items left blank within the questionnaire (including our
definition of "a large number"). For our analyses, we will include all respondents of the final
data set where answers for the variables of interest are present.

*Page 10, line 46 – explanation will be needed on how variables are aggregated*

4.25

We are now more clear about our procedure and refer to section 3 for details about our
variables.

Comments to Reviewer 5

Thank you for evaluating our manuscript and providing detailed feedback.

The biggest issue is that the authors do not plan to test for measurement invariance before proceeding to their focal analyses. A lack of measurement invariance can compromise the validity of conclusions. I recommend the use of multiple group models using SEM to test invariance. There are several resources I can recommend to learn about this technique if necessary: Chen (2008, JPSP) and the Kline SEM textbook are good introductions.

5.1

Based on other reviewer comments, we added a plan to perform measurement invariance tests by comparing configural invariance, metric invariance, and scalar invariance models in the quality check section (as per Milfont & Fischer, 2010). We also elaborate what we will do for our planned analyses once the equivalence of invariance is not achieved. We are aware that this is not equal to SEM but should give some of the same insights.

The authors state “Continuous variables are centred on grand means, or converted to a z-score for the convenience of interpretation” (p. 10). Centering is a good idea; standardizing (converting to a z-score) is a bad idea in multi-level models (because you disturb the variance structure across levels). Avoid standardizing data in multilevel analyses. Rely on conversions to quasi-r-squared (or quasi-r) to interpret effect sizes. Effect sizes should absolutely be interpreted alongside p-values.

5.2

As we are aware of the issue you raised, the MLM analyses for power analysis were run twice with grand-mean centering vs. standardisation of the continuous variables. Only estimated fixed effects were reported based on the modeling with standardised variables, whereas the random effects were estimated with grand-mean centering. We added a note about this rationale in Section 6

The authors have clear associations that they plan to test. At least as the introduction is currently written, these tested associations do not appear to stem from any particular theory or model. I don't necessarily have an issue with this, but the tested models are not confirmatory in the sense that they test a theory-driven hypothesis. The introduction could benefit from a model or framework if there is one. I.e., why test these associations and not other models?

5.3

This paper is the first of several that will be coming out of our consortium. Presently we wish to take advantage of the rapid response scheme to share the current picture of experienced stress and its - very probably not unidirectional - relationships to trust in preventive measures. As such, the intention behind the hypotheses is more exploratory than tied to any particular model. Following your advice we have rephrased several passages as well as the title to reflect this more clearly, and re-written the literature section.

*The authors intend to model a variety of demographic country level variables. I wonder if*
*there are other pandemic-specific country level variables that should be included – e.g., rate*
*of country-level hospital bed or protective equipment shortages, date of lockdown onset (or*
*presence of lockdown), previous pandemic incidence. Further, I wonder if time of data*
*collection should be explicitly modeled (e.g., days from date of survey to onset of lockdown).*
*I don't have strong feelings about this, but wanted to make the suggestion.*

12 5.4

We acknowledge that the proposed ideas are indeed important for current pandemic
situation. After we have thoroughly evaluated the goals of the present study and their
compatibility with the proposed control variables, we decided to keep the original design, as
we believe medical resources may have less to do with compliant behavior.

*My preference would be to see all models with and without controls included. The authors*
*could present their preferred specification (as presented here) and contrast it with models*
*without level 1 and level 2 controls.*

26 5.5

We appreciate your comment regarding the comparison of models with vs. without control
variables. We intended to compare five different models (please see our response to your
comment #6 below), so this point could also be addressed simultaneously. Once we
compare Model 3 (model with fixed effects) vs. Model 4 (Model 3 + control variables), then
we will be able to address the point that you raised.

*Hypotheses 1a to 1d should be tested with multi-level models (not ANOVAs). The reason is*
*that country should be a random intercept, not a fixed effect. Hypotheses 2-5 should include*
*and test random slopes for the predicted effects. For all hypotheses, I recommend a series*
*of nested models: a null model (no predictors, no random effects), a random intercept only*
*model (country as random intercept), a model with fixed effects of predictors, a model adding*
*control variables, a model with random slopes for predictors. Each model can be compared*
*to its predecessor via a likelihood ratio test. Strive to specify maximal models, but trim*
*random effects (usually random slopes) if there are convergence issues.*

46 5.6

Thank you very much for your suggestion regarding the inclusion of control variables in
MLM. Unfortunately, we decided to perform ANOVAs to test H1. Because we are mainly
interested in comparing the mentioned dependent variables between countries, we found
that ANOVA is more theoretically relevant than MLM. Also, to perform post-hoc tests to
examine international differences in the DVs, employing ANOVA would be necessary.
Moreover, technically, adding countries as a fixed effect may create the difficulty to have
models converged.

We appreciate your suggestion regarding the model comparison for H2-H5. We agree with
you that comparing different nested models would be informative in identifying the best
model to predict the outcome variables of interest. Thus, we revised our analysis plans so

that they include some procedures for the model comparison. Following your suggestion, we
intend to compare these five models:

- ● Model 1: Null model (no predictors or random effects added).
- ● Model 2: Random intercept-only model (Model 1 + country as a random intercept).
- ● Model 3: Model with fixed effects of predictors (Model 2 + fixed effects).
- ● Model 4: Model with control variables (Model 3 + control variables).
- ● Model 5: Full model (Model 4 + random slopes for predictors).

We plan to perform a likelihood ratio (LR) test to examine whether the addition of random effects
significantly improves the model (Model 1 vs. Model2). For other comparisons, i.e., Models 2 vs. 3,
Models 3 vs. 4, Models 4 vs. 5, we will employ other methods, such as the pseudo R^2 comparison
and/or omnibus F test because the SAS macro allows the use of the LR test only for the comparison of
models with vs. without random effects.

*Plan to report descriptive statistics for all variables (means, standard deviations, alpha or*
*omega reliability, intercorrelations between variables). I don't know these variables well*
*enough to anticipate whether there might be problems with multi-collinearity of predictors in*
*the planned models. Anticipate and plan for dealing with multi-collinearity if necessary.*

5.7

We added that we will provide descriptive statistics for all variables. Although we do not
expect to have problems with multi-collinearity, we will compute and check VIFs. We have
added this procedure including our approach to deal with potential multi-collinearity in
section 10.

*It's not clear in the proposal how country level variables will be scaled. That is, will they be*
*treated as continuous? Contrast coded categories? For individual level predictors, I do*
*wonder if the authors should plan to collapse educational categories (perhaps to just three:*
*did not finish compulsory schooling in that country; finished secondary schooling; university*
*degree or higher). If you choose to do this, I would contrast code the categories rather than*
*treat them as continuous.*

5.8

We added the necessary information to the table in section 3.

*It's not clear why for hypothesis 1c, the focus is on OECD_Institutions item 1, rather than all*
*OECD_Institutions items. Should these items be combined into a total score if possible?*
*Either way, would be good to have more reasoning behind this choice.*

5.9

Good point! We now provide more details about the variable in section 3.

Some variables are measured with single items that may have limited reliability/validity. This is unavoidable given that we cannot alter the design at this point. Just be aware that this may further compromise statistical power of some analyses.

5.10

Thank you for pointing this out. We will keep this in mind and discuss it as limitation.

Regarding power, the authors state “we plan to recruit at least 30 participants per country so as to detect both the effects of individual- and country-level predictors” (p. 9). They cite Arend and Schäfer (2019)’s Table 8 as justification. They note that in the ESS database, medium ICCs, small to medium effects of level 1 and level 2 predictors, and small cross-level interactions are observed. They expect to collect data from 44 countries (level 2 units). The level 1 sample size (individuals per country) could vary. Of the 10 hypotheses, four focus on the ICCs themselves, five focus on level 1 predictors, and one contains an interaction between two level 1 variables. In six of the hypotheses, level 2 control variables are included. Based on Arend and Schäfer (2019)’s Table 8, with 44 countries and a sample size of 30 per country, small level 1 effects are underpowered (medium and large level 1 effects are adequately powered), regardless of the ICC. Only large effects of level 2 variables are adequately powered (regardless of ICC); small and medium effects are underpowered. Only large cross-level interactions under conditions of large random slopes are adequately powered; most random slopes are underpowered. Adding additional participants per country seems to matter less than adding more countries (or more broadly, level 2 units). Thus, it seems unavoidable that the current attempt will be underpowered in some ways, mostly because there is likely not an easy way to recruit more countries into the study. Having 30 participants per country seems reasonable based on the power tables, but the authors may run into issues with convergence with this size of sample per country. In particular, the multi-group invariance models tend to do better with larger per group sizes. I don’t see a strong reason to cut data collection short; it would seem prudent to get as much data as possible while the pandemic is happening. The results of the study are not themselves time sensitive (no government is going to base decision making on what these results show). Therefore, I would err on the side of sampling strategy that allows the collection of as much data as possible.

5.11

We agreed with your comment on the underpower issue. Therefore, the scope of the present
study will focus on modeling the structure of individual responses nested within countries,
and examining our proposed hypotheses with respect to psychological factors at the
individual levels (rather than the country-level predictors). Furthermore, as we recognised
the issue with convergence, we intend to conduct the multilevel modeling analyses as
follows:

- 1. IF our planned analyses can be conducted without any issue (e.g., failed
convergence) with the dataset that passed the measurement invariance test, THEN
report the findings.
- 2. IF 1 failed, THEN conduct the planned analyses with the dataset that includes all
countries with $N \geq 30$ (according to the country-level exclusion criteria).

If 1 failed and findings from 2 are reported, we will discuss limitations and caveats regarding
the interpretation of the findings in the discussion section.

*What is the rationale for the 15% cutoff for identifying floor/ceiling effects? Why not 10 or 20?*
*Seems arbitrary. The exclusion rules for fast response time and Mahalanobis distance*
*seemed appropriate and a good practice to improve data quality.*

5.12

We agree that the cutoff is a bit arbitrary. Indeed, we had the very same discussion until we
settled on 15% based on this work, that suggested the use of the 15% threshold:

McHorney, C. A., & Tarlov, A. R. (1995). Individual-patient monitoring in clinical practice: are
available health status surveys adequate?. *Quality of Life Research*, 4(4), 293-307.

*Causal language: There is causal language in the title of the paper (“the impact of”) and*
*sprinkled throughout the paper itself. Causal language is inappropriate for this study’s*
*methods.*

5.13

We have softened the causal language in various places, including hypotheses and the title.

***Trust and compliance with preventive measures in the 2020 coronavirus pandemic:***
***relationships to trust, stress and worry across 45* countries from the COVIDiSTRESS***
***global survey***

STAGE ONE REGISTERED REPORT
REVISED MANUSCRIPT(RSOS-200589)

This urgent ~~call~~ Registered Report (RR) presents the COVIDiSTRESS global survey, which collects
data on human responses to the COVID-19 pandemic (caused by the coronavirus SARS-CoV-2) in
45* countries. This open science study was co-designed by researchers from numerous universities
across the world (for a full list of universities see section 2) to investigate how psychological and
behavioural responses to the COVID-19 pandemic differ across countries and cultures, and how this
has impacted social behaviour, coping and trust in government efforts to slow the spread of the
coronavirus.

Global data gathering efforts like the COVIDiSTRESS project are of pressing importance during the
COVID-19 crisis, in order to understand how different government measures succeed or fail in
achieving objectives to change public behaviour. By generating data rapidly, and disseminating them
continuously as open data on the Open Science Framework (OSF), the international research network
behind the COVIDiSTRESS project aims to yield rapid insights for the use of practitioners such as
health communicators, government officials and policy-makers at all levels, as well as members of the
academic community. Currently, there are many psychologists and social scientists making a great
effort to better understand the psychological effects of the COVID-19 pandemic (e.g., Wang et al.
2020) and to provide implications and suggestions for practitioners (see e.g., Winton Centre for Risk
and Evidence Communication, Harvard Kennedy School, London School of Economics). To our
knowledge, there is currently no other project that allows to compare human responses to the
COVID-19 across such a large number of countries. Further, the COVIDiSTRESS project specifically
attempts to leverage fast organic sampling, which favours a varied reach over predetermined
population criteria, and avoids the risk of unintentionally overtaxing the same limited participant
pools on e.g. Amazon Mechanical Turk or local survey bureaus for COVID-19 related studies.

[*RR stage one note to reviewers: We will update this projected number based on final number of
valid responses according to inclusion/exclusion criteria]

1. Background and Research Questions

1. What is the impact of psychological stress on trust in and compliance with government efforts to prevent COVID-19 compared across countries?
2. To what degree is psychological stress related to concerns over COVID-19 itself, and what proportion of the variance in psychological stress should be explained by other factors?

This Registered Report (RR) analyses data from a swiftly executed global survey study meant to allow both hypothesis testing and exploratory cross-country comparisons. As an open science project, the COVIDiSTRESS effort as a whole intends to supply data to the public, allowing researchers and other stakeholders to independently conduct their own analyses (Paulik, 2020). As such, the present RR investigates 10 central hypotheses, focusing on people's experiences and behaviours during the COVID-19 crisis many other variables from the COVIDiSTRESS global survey are not included in the analyses of the present RR.

We are experiencing an unprecedented outbreak of a new pandemic. To prevent the spread of coronavirus, governments are imposing a range of measures, which include, but are not restricted to, quarantines, physical distancing and limits to civil liberties. Inevitably, these changes have generated a variety of psychological responses (Brooks et al., 2020), which in turn shape the level of compliance with preventive efforts. Extant research on the factors that shape willingness to comply with public health efforts has highlighted the importance of psychological responses such as anxiety and risk aversion (Leung et al. 2003; Taylor, 2019, Wong and Jensen, 2020) as well as trust in state authorities (Abraham, 2011; Capelos et al., 2016; Seligman, 1997). A recent study on risk and compliance during COVID-19 in Singapore found that high levels of trust in the government led to lower risk perception and non-compliance with preventive measures (Wong and Jensen, 2020). This varied body of findings clearly signifies the need to explore further the psychological mechanisms at play that shape responses to preventive measures.

Not only do pandemics foster a fear of contagion, but also misinformation, uncertainty and lack of clarity about how to react (Khan & Huremovic, 2019; Cava et al. 2015). Concern over a disease outbreak has typically been found to be positively correlated with preventive behaviours or compliance with health guidelines (Chien et al., 2017; Prati et al., 2011; Taha et al., 2013). However, this relationship is complex. Worry has been found to be related to trust in disease-related information and media (Boyd & Jardine, 2011; Liao et al., 2011; Ro et al., 2017) and the perception of openness and reliability of governments and health organisations (Bish & Michie, 2010).

Concern over one's medical risk during a pandemic can also be a source of ongoing stress (e.g. in H1N1, Bults et al., 2011; in MERS, Ro et al., 2017). For instance, stress was widespread during the 2003 SARS epidemic (Cong et al., 2003; Deurenberg-Yap et al., 2005). This relationship is potentially exacerbated by the fact that even some of the most efficient methods of slowing the spread of a disease, such as self-isolation and quarantine, also take an immediate and potentially long-lasting psychological toll among affected populations (Brooks et al., 2020; Huremovic, 2019; Maunder, 2009). Surveys of quarantined healthcare workers (Reynolds et al., 2008) and members of the general

population (Hawryluck et al., 2004) following the 2003 SARS outbreak found evidence of acute stress
responses in cases with a longer duration of quarantine and greater perceived difficulty in compliance
with recommendations. Similarly, studies of patient groups typically find a negative relationship
between experienced stress and compliance with prescribed behaviors, often with direct medical
consequences (Baucom et al., 2015; Mugavero et al., 2009; Singh et al., 1996), and this correlation
has been extended to compliance with measures intended to minimise the spread of virulent disease
(Perez et al., 2012).

Both the medical situation and the psychological effects of isolation and confinement (Oliver, 1991;
Palinkas & Suedfeld, 2008) thus need to be taken into consideration when prolonged periods of
quarantine are implemented, in order to understand acceptance of government measures and
compliance with preventive measures. Compliance with medical guidelines has been shown to
decrease not just as a result of heightened stress levels (Karvinen, Murray, Arastu, & Allison, 2013),
but also of minor everyday stressors such as workplace conflict or household responsibilities
(Hitchcock, Brantley, Jones, & McKnight, 1992). Further, the personal financial impact of a pandemic
can be severe and stressful, especially for people who are already experiencing financial hardship
(Taylor 2019). Prolonged states of emergency and the chronic psychological, social, and economic
stressors related to them (Brooks et al., 2020; Huremovic, 2019) may decrease compliance with
behavioural objectives during pandemics. Sources of stress and worry may thus have direct
implications not simply for well-being, but also for the effectiveness of collective efforts and the
degree of public compliance with self-isolation measures. For this reason, we test the hypotheses that
stress and concern over the coronavirus will be adversely correlated with compliance with preventive
behavioural measures suggested in each country.

Overall, it thus seems that worry, trust and situational stress are associated and jointly influence
behaviour during pandemics. Efforts such as closing down schools and workplaces and calls for
people to self-isolate in their homes are likely to constitute a source of both existential and practical
stress, unrelated to the fear of contracting the disease. For instance, some stress is known to result
from inability to participate in work or interpersonal problems in the family (Lynch, 1999;
Neighbours, 1997). A pandemic may also exacerbate existing financial precarity, generating another
source of stress for those experiencing financial hardship (Taylor, 2019). It is thus an interesting
question whether the known negative relationship between stress and compliance will apply to the
population at large. Especially since e.g. physical distancing measures and hand-washing are
recommended to protect not only the individual in question, but also other people in their
surroundings. We thus test the hypothesis that stress will indeed be negatively correlated with
compliance with preventive measures during COVID-19. However, the extent of compliance with
preventive measures has repeatedly been found to be influenced by the perceived gravity of the
pandemic as well as trust in the preventive measures against it (Bults et al., 2011, Gilles et al. 2011,
Lin & Bautista, 2016; Prati et al. 2011). We this also expect that trust in the preventive measures taken
in each country will lead to better compliance and less worry over the COVID-19 medical situation.

In this context, it is also worth investigating what factors may alleviate perceived stress, and thus,
positively impact not just well-being, but also the likelihood of acceptance and compliance with
behavioural guidelines. Social and practical support from groups such as one's family, friends, and
colleagues moderate the effect of concern for the disease or other sources of stress on one's
psychological well-being (Mak et al., 2009; Wang et al., 2014). Indeed, interpersonal communication

has been shown to affect the relationship between risk perception and actual behaviour during health crises, where mass media messages may fall short (Griffin & Dunwoody, 2000). Various theoretical dimensions of social support, including social integration and reassurance of worth (Weiss, 1974), are captured by the Social Provisions Scale (SPS) and its shorter versions (SPS-10; Steigen & Bergg, 2019). According to the stress-buffering hypothesis (Cohen, 2004), social support ~~through e.g. interpersonal communication~~ buffers people from other negative consequences of distressing events. For instance, social support could promote less threatening interpretations of stressful situations (ibid.; Cohen & Wills, 1985), and thus, possibly more positive appraisals of institutional and governmental efforts to stop the spread of a pandemic. Empirical evidence shows that perceived social support helps build up resilience against disaster or catastrophic events, and exerts its protective function on alleviating psychological distress (Chan, Lowe, Weber & Rhodes, 2015); furthermore, social support has been found to correlate positively with compliance in patient populations (Adewuya et al., 2010). It is thus reasonable to investigate if we can extend the previous findings related to the stress-buffering hypothesis to the context of acceptance and compliance with behavioural guidelines during the pandemic. Here, we explore the relationship between experienced stress and trust in the efforts made by governments to set up behavioural guidelines to curb the spread of COVID-19, as well as in relation to social provisions. In particular, we hypothesise that social provisions should negatively moderate the negative effect of stress on compliance with behavioural objectives during the pandemic.

So far, little is known about the cross-national differences in reactions to preventive measures during the 2020 pandemic. However, cross-cultural differences have been observed in coping with and responding to stressors (Kuo, 2011; Wahid, 2017) and in acceptance of various countermeasures against pandemic outbreaks (Bennett & Carney, 2010); thus, optimal institutional responses to the COVID-19 crisis may require culturally and situationally appropriate public health interventions, both to alleviate human distress and to communicate preventive measures in an effective manner. Understanding the psychological and behavioural implications of human experiences of the current pandemic in various countries will help us identify some of the psychological risk factors for negative outcomes under the conditions of a pandemic and aid governments and other organisations to devise effective interventions structured around the stressful consequences of enacting self-isolation measures. In the present study, we will thus take a multilevel analytic approach with individual responses nested within each country in order to estimate the variability in DVs residing in both individual- and country-level (Hox, 2010). By taking into consideration non-independence among individual responses within a country, as well as the differences between countries, we can better assess the extent to which individual-level variables and country-level variables explain the amount of variance at the individual level and country level respectively.

2. Methods

A global survey study is currently being conducted with the purpose of mapping the psychological and behavioural responses to the COVID-19 pandemic across countries, and how this has impacted social behaviour and trust in government efforts to slow the spread of the coronavirus. As data collection is urgent, this stage one RR was submitted as the survey had already been launched, but before any data was extracted.

The COVIDiSTRESS survey (COVIDiSTRESS global survey network, 2020 - available at <https://osf.io/z39us/>) consists of two parts. The first part collects general demographic data including proximate effects of the COVID-19 pandemic (e.g., isolation status, first-hand experience, attenuated risk), personality traits, and self-reported variables, such as loneliness and perceived stress (Cohen, Karmarck & Mermelstein, 1983), daily behaviours including compliance with general and social preventive measures. The second part contains sets of more specific items related to distress, coping, and social provisions. Firstly, we inquire about people's experiences of distress and worry during the ongoing outbreak of coronavirus, e.g. access to amenities, loss of work, adapting work, education, and social interactions to digital platforms, the social stresses of confinement with adults and children. Also, included are questions about the factors people experience as soothing and positive to coping (e.g. social contact, staying informed, dedicating oneself to preparation, hobbies, religion) and the Social Provisions Scale (SPS; Steigen & Bergh, 2019). Validated short versions of established measures are used whenever possible and available in local languages. In order to protect participants' data and avoid sensitive information, participants are not asked about COVID-19 symptoms or other data with direct medical implications that would allow third-parties to identify them.

Translation has been completed for 46 languages and localisations (Afrikaans, Albanian, Arabic, Bangla, Indonesian, Bosnian, Bulgarian, Chinese [Simplified and Traditional], Croatian, Czech, Danish, Dutch [Belgium, Netherlands], English, Spanish [Argentina, Colombia, Cuba, Mexico, Spain], Filipino, Finnish, French, German, Greek, Hebrew, Hindi, Hungarian, isiXhosa, isiZulu, Italian, Japanese, Korean, Lithuanian, Nepali, Persian, Polish, Portuguese [Brazil, Portugal], Romanian, Russian, Slovakian, Serbian, Turkish, Urdu, Vietnamese) with the possibility for more translations in future waves. Translations were completed by a forward translator from the original English version, and then validated through both panel and back-translation process by separate translators when possible.

Given the urgent call for COVID-19 research, the survey received a waiver to commence data collection from the IRB office at Aarhus University, Denmark. The survey and data extraction schedule was pre-registered at the Open Science Framework ([COVIDiSTRESS global survey network, 2020](https://osf.io/z39us/)).

[note for RR stage two: the data set will also be submitted for publication in *Scientific Data*]

3. Key Independent and Dependent Variables

The COVIDiSTRESS global survey contains a number of variables (full survey at <https://osf.io/z39us/>, COVIDiSTRESS global survey network, 2020). This urgent call RR focuses on 10 hypotheses (see section 1 and 4) and comparisons between countries. Table 1 gives an overview of all variables that are relevant for our hypotheses.

Table 1. A list of variables of interest in the present study.

Variable name	Description	Measurement	Remarks
---------------	-------------	-------------	---------

Independent and dependent variables

[Scale_PSS10]	Perceived stress for the past week	Perceived Stress Scale (PSS-10). 10 items. 5-Point Likert Scale (1 =never, 5 = very often). Validated language versions where available. Back-translated where necessary. Sum score will be computed. Continuous variable.	Cohen, Kamarck, & Mermelstein (1983)
[OECD_institutions]	Trust in country's efforts to handle the coronavirus situation	On a scale from 0-10 (0=not at all, 10=completely), how much do you personally trust each of the institutions: "government's effort to handle coronavirus" (added). Note that 5 other institutions were rated which are not relevant to the present research. Continuous variable.	OECD guidelines on measuring institutional trust (OECD, 2017)
[Corona_concerns]	Concern over coronavirus	5 self-reported items to capture concerns about coronavirus consequences on a 6-Point Likert Scale (1=strongly disagree, 6=strongly agree): Concern about consequences of the coronavirus (1)...for yourself, (2)...for your family, (3)...for your close friends, (4)...for your country, (5)...for other countries across the globe. Mean score will be computed. Continuous variable.	-
[Compliance]	Compliance with local prevention guidelines	Item "I have done everything I could possibly do as an individual, to reduce the spread of coronavirus" captures compliance with local prevention guidelines on a 6-Point Likert Scale (1=strongly disagree, 6=strongly agree). Continuous variable.	-

Control variables (covariates)

[age]	Participants' age	Continuous variable.	-
[gender]	Participants' gender	0=male, 1=female, 2=other/would rather not say	-

			Categorical variable.	
[education]	Participants' education	1=PhD / Doctorate, 2=College degree, 3=Some College or equivalent, 4=Up to 12 years of school, 5=Up to 9 years of school, 6=Up to 6 years of school, 7=None	Categorical variable.	-
[country]	Country of residence	List of all countries where the survey is disseminated/spread.	Categorical variable (factor).	
[SPS]	Available social provisions in critical/distressing situations	Social Provisions Scale short form SPS-10. 10 items. 6-Point Likert Scale (1 =strongly disagree, 6 = strongly agree). Validated language versions where available. Back-translated where necessary. Mean scores will be computed. Continuous variable.		Steigen & Bergh (2019)
[population_size] [GDP] [edu_attainment] [unemployment] [gini_coefficient]	Country data	- Population size - GDP per capita - Education attainment - Unemployment rate - Income inequality (GINI coefficients)		Country demographic variables from OECD (OECD, 2020)
		Continuous variables.		We plan to add country-level variables (population size, GDP per capita, education attainment, unemployment rate, and GINI coefficient) as control variables to predict the DVs.

4. Hypotheses

First, this RR is intended to draw a map of differences in the psychological impact of COVID-19, and
the trust in government measures and compliance with preventive guidelines in the countries
represented in the first data extraction on Monday April 6, 2020 (see inclusion criteria in section 7).
We expect that the *COVIDiSTRESS global survey* will show significant differences between countries.

- ● H1a: Perceived stress will differ between countries.
- ● H1b: Concern over the coronavirus will differ between countries.
- ● H1c: Trust in government efforts to slow the spread of coronavirus will differ between
countries.
- ● H1d: Compliance with behavioural guidelines to slow the spread of coronavirus will differ
between countries.

Exploratory cross-country comparisons and descriptive statistics will be provided in stage two of this
RR.

Second, the present analysis focuses on how both stress and concerns about the COVID-19 pandemic
(e.g. isolation, changes in public life and worry over the virus itself) are related to compliance with
preventive guidelines and trust in government efforts intended to reduce the spread of coronavirus.

Compliance with behavioural guidelines

- ● H2a: Across countries, perceived stress will be negatively correlated with compliance with
behavioural guidelines to slow the spread of coronavirus.
- ● H2b: Across countries, concern over the coronavirus will be positively correlated with
compliance with behavioural guidelines to slow the spread of coronavirus.

Trust in government efforts

- ● H3a: Across countries, trust in government efforts to slow the spread of coronavirus will be
negatively correlated with perceived stress.
- ● H3b: Across countries, trust in government efforts to slow the spread of coronavirus will be
negatively correlated with concern over coronavirus.

Third, we analyse the sources of stress during the global COVID-19 crisis. We assume that concern
over the coronavirus, e.g. arising from the direct health threat of COVID-19 and uncontrollable nature
of the pandemic, will be a direct stressor.

- ● H4: Across countries, concern over the coronavirus will be a predictor of perceived stress.

Fourth, based on the stress buffering hypothesis, we analyse whether social support can alleviate the
negative effect of perceived stress on the extent to which people comply with preventive measures.
While stressful situations may be unavoidable during a pandemic, the availability of support systems
may promote less threatening interpretations of the situation, and thus, lead to a more positive
appraisal of efforts to combat the coronavirus.

- H5: Across countries, availability of social provisions will negatively moderate the effect of perceived stress over coronavirus on compliance with behavioural guidelines to slow the spread of coronavirus.

Lastly, individual-level control variables/covariates will be included in our multilevel analyses that examine hypotheses 2-5. These variables/covariates are respondent gender, age, and education level. According to Coffé and Bolzendahl (2010), women are less interested and ~~involve less~~ in politics than men. Because trust in government can be easily affected by salient political issues (such as government's efficacy to handle the pandemic), we expect that respondent gender may predict trust in government efforts. In addition, we expect that respondents of different ages may react differently in terms of their experienced stress and worry (Aldwin, 1996). As for respondent education, Dalton (2005) indicated that education level is related to a decrease in trust in government. Based on the above reasons, respondent age, gender and education level will serve as individual-level covariates/control variables when testing our hypotheses 2-5. In addition, we considered adding several macro-level (country-level) control variables that may be associated with the outcomes. We were particularly interested in each country's socio-economic growth (e.g. GDP, Gini coefficient, and education attainment). Country-level socio-economic factors have been reported to be related to psychological well-being indicators (e.g., stress, negative affect), so they have a merit to be included in our models (Haushofer & Fehr, 2014). Moreover, country-level education attainment is also considered given the negative association between the overall educational level and poor psychological well-being at the country level (Schütte, Chastang, Parent-Thirion, Vermeylen, & Niedhammer, 2014).

5. Conditions of assignment

No direct manipulation or assignment.

6. Number of observations to be collected and rule of termination

Given that this large-scale survey will be spread by numerous researchers all over the world, we have limited control over how many respondents we will collect in total and per country. Thus, we define a set of stopping rules that are practically feasible.

Previous research confirms that statistical power in multilevel design is rather complex and greatly depends on the nested structure of the data (Arend & Schäfer, 2019). As such, a general rule of thumb (i.e., 30-30 rule; Kreft & de Leeuw, 1998) is unlikely to be applicable to all different data structures. Therefore, we sought to refer to a more specific guideline to plan sample sizes according to the estimates acquired from pre-existing databases.

Due to a lack of previous research that could be used for reasonable power analyses, we ran a planned multilevel analysis for H5 (specified in Section 10), as there was a data set at hand, which we perceived as sufficiently suitable. We used the 2018 European Social Survey (ESS) database to

estimate the effect sizes of fixed and random effects as well as intraclass coefficients (ICCs). The ESS
database consists of 43,215 respondents from 23 European countries (in average 1,878 respondents
5 per country). Although the variables available in the ESS database are not identical to those in our
survey, given their similarity in the underlying constructs, we used the health condition variable (a
higher value indicating poorer health condition) and satisfaction about the government in the ESS data
as proxies for stress, compliance, and trust in government efforts for our original variables of interest,
respectively.

The ICC from the observed ESS data is .26, indicating that 26% of variability comes from
between-country differences. As we are aware of the issue that random effects are often distorted
while the variables are standardised, the random effects were estimated with grand-mean centred
continuous variables. The same multilevel analyses were run again with standardised variables only in
order to obtain the standardised estimates of fixed effects. Standardised estimates of fixed effects of
the individual-level predictor (individual's health condition: $\beta = .09$; social provisions: $\beta = .21$) and its
country-level predictor (averaged health condition in each country: $\beta_s = -.35$) are deemed as having
small and medium effect sizes, respectively. Their cross-level interaction effects had only small effect
sizes ($\beta_s = .03 - .10$).

Subsequently, we refer to the guidelines from Arend and Schäfer (2019), which provide a fast and
frugal power estimation for each combination of effect size, the value of ICC, as well as the size of
random slope variance. By looking up Arend and Schäfer's Table 8 of power simulation results that
give a required sample size and group size (in our case, the number of countries) to detect such effects
with 80% statistical power (p. 17), in the present study, we plan to recruit at least *30 participants per*
*country* so as to detect both the effects of individual- and country-level predictors. However, since our
data collection process is still ongoing, the potential number of group size and sample size will be
greater than we have reported here.

7. Inclusion criteria

This open science project involves numerous researchers coming from diverse countries. The surveys
were distributed in the countries of these researchers. Hence, there was no systematic "selection" of
the countries. Each country is represented in the COVIDiSTRESS data set if there is at least one
participating researcher in the open science effort who comes from this country. Having said this, it
must be noted that this limits the representativeness of our data.

All participating researchers received the same task of starting to distribute the survey in their
language on March 30, 2020 (exception: Denmark started on March 26, 2020, see section 8). All
possible channels were allowed, e.g. social media, panels, e-mails to friends or organizations, use of
media contacts such as the newspaper or TV, websites of organizations such as universities or NGOs
involved in health communication. Moreover, we also asked all participants to help spread the survey
after they finished their own responses.
**In order to be considered** for the present analysis, a country needs at least 30 respondents at the time
of extraction (i.e. April 6 2020). In order to be considered as a valid response, a respondent in the

COVIDiSTRESS global survey must have reported their country of residence, and submitted valid responses to the variables treated in each analysis (also see section 8).

8. Exclusion criteria

We define exclusion criteria on the country and on the respondent level.

Country level

Active dissemination of the survey and calls for participation are carried out via online and traditional media platforms in Afghanistan, Argentina, Australia, Austria, Bangladesh, Belgium, Bosnia, Brazil, Bulgaria, Canada, China, Colombia, Croatia, Czech Republic, Denmark, Finland, France, Germany, Greece, India, Indonesia, Israel, Italy, Japan, South Korea, Lithuania, Malaysia, Mexico, Netherlands, Pakistan, Philippines, Poland, Portugal, Russia, Serbia, Singapore, South Africa, Spain, Switzerland, Taiwan, Turkey, United Kingdom, United States of America, and Vietnam. The number of participating countries may be increased in the future as researchers and others interested in sharing the COVIDiSTRESS global survey join the project.

If fewer than 50% of all participating countries have failed to generate at least 30 valid responses by first extraction on April 6 2020, we extend the data collection for one week, that is, we utilise the weekly data extracted on April 13 2020. All countries that fail to generate 30 respondents by April 13 2020 will not be included in the final data set. Section 6 and 10 provides a detailed justification for the exclusion of countries that do not reach 30 respondents. In short, 30 respondents per country is the minimal number of respondents required to ensure sufficient statistical power.

Individual level

On the individual level, the length of the survey can lead some participants to skip questions, but also give repetitive or unrepresentative answers (e.g., Krosnick & Alwin, 1987), leading to misclassification of participants and responses that do not reflect real experience (Egleston, Miller, & Meropol, 2011). We employ the following exclusion measures to protect against these threats.

1. First, the predicted duration of the survey in Qualtrics is 22 minutes if all questions are answered and free text boxes are filled out. On that basis, we exclude all responses who completed the whole survey in less than 2 minutes and 12 seconds, equivalent to one-tenth of the estimated time.
2. Second, we use Mahalanobis distance to detect multivariate outliers due to random or carelessly invalid responses (Curan, 2016; Dupuis et al. 2019). Participants with a $p < .001$ in the chi-square test will be excluded.

9. Quality checks

Regarding quality checks, we will check for floor / ceiling effects within the following scales / item
batteries: [Scale_PSS10], [Corona_concerns], [Compliance], [SPS]. We will evaluate floor / ceiling
effects on the base of the percentage of the respondents with the minimum / maximum scores.
Therefore, we will provide the percentage (%) and n for respondents with the minimum / maximum
scores. Floor / ceiling effects will be considered as present if minimum / maximum scores occur in
15% or more of the respondents. We decided to employ the 15% threshold as suggested by
researchers (see McHorney & Tarlov, 1995).

We will also investigate the cross-cultural equivalence of the Perceived Stress Scale (PSS-10, see
Cohen et al., 1983; Cohen & Williamson, 1988) prior to any analysis. Using a multi-group factor
analysis we will compare the models assuming the two-factor structure (positive and negative, with
the latter consisting of reversed items; Chaaya et al., 2010; Roberti et al., 2006) across all countries
(configural invariance), with a model with factor loadings and latent correlations constrained to be
equal (metric invariance), and items' intercepts to be the same in all groups (scalar invariance). When
evaluating the model fit, we will rely on the usually applied criteria (Hu & Bentler, 1999), in which a
Comparative Fit Index (CFI) and Tucker Lewis Index (TLI) above .90 would indicate adequate fit,
whereas a standardised root mean square residual (SRMR) below .06, and a root mean square error of
approximation (RMSEA) below .08 indicate no misfit. When evaluating the measurement
equivalence, we will compare the configural invariance model with the metric invariance model, and
then the metric invariance model with the scalar invariance model (Milfont & Fischer, 2010). As these
models are characterised by a growing complexity (each subsequent model is nested within the
previous one), while assessing models' superiority, we will rely on cut-off criteria recommended for
testing measurement invariance: a change of CFI (ΔCFI) less than .01 ($\Delta CFI < .01$), a change of
RMSEA of less than .015 ($\Delta RMSEA < .015$), and a change of SRMR less than .01 ($\Delta SRMR < .01$)
indicating that two compared models do not differ in terms of model fit (Chen, 2008; Cheung &
Rensvold, 2002). However, in case that the equivalence of invariance is not achieved, based on
recommendations indicating that partial invariance may allow for reasonable comparisons (see e.g.,
Byrne, Shavelson, & Muthén, 1989), we will also estimate models with partial invariance. If the
equivalence of partial invariance is achieved, we will proceed with the cross-countries comparisons.
Nevertheless, considering the fact that even the equivalence of partial invariance is sometimes
difficult to achieve when comparing numerous countries (Davidov et al., 2014), we will follow
recommendations to exclude countries in which the fit is too poor (Davidov, 2009; Lomazzi, 2018),
and proceed with further analyses on remaining countries, among which the condition of equivalence
is met.
**Given** that the number of countries to be analysed directly influences the power of the planned
multilevel modeling, we intend to conduct the multilevel modeling analyses as follows:

- 1. IF our planned analyses can be conducted without any issue (e.g., failed convergence) with
the dataset that passed the measurement invariance test, THEN report the findings.
- 2. IF 1 failed, THEN conduct the planned analyses with the dataset that includes all countries
with $N \geq 30$ (according to the country-level exclusion criteria).

If 1 failed and findings from 2 are reported, we will discuss limitations and caveats regarding the
interpretation of the findings in the discussion section.

**10. Analysis plan**

Data analysis

- The analysis uses the tidyverse, BayesFactor, and brms R packages.
- Descriptive statistics are computed for all variables/scales (i.e., M, SD, α , rs).
- Multilevel models are run using SAS PROC MIXED with Restricted Maximum likelihood (REML) and Kenward-Roger denominator degrees of freedom.
- To identify the best model, we intend to compare the following five models for each hypothesis. We plan to perform a likelihood ratio (LR) test to examine whether the addition of random effects significantly improves the model (Model 1 vs. Model 2). For other comparisons, i.e., Models 2 vs. 3, Models 3 vs. 4, Models 4 vs. 5, we employ other methods, such as the pseudo R^2 comparison and/or omnibus F test because the SAS macro allows the use of the LR test only for the comparison of models with vs. without random effects. Bayesian MLM with brms is performed with the best model.
 - Model 1: Null model (no predictors or random effects added).
 - Model 2: Random intercept-only model (Model 1 + country as a random intercept).
 - Model 3: Model with fixed effects of individual-level predictors (Model 2 + fixed effects).
 - Model 4: Model with country-level control variables (Model 3 + control variables).
 - Model 5: Full model (Model 4 + random slopes for predictors).
- For frequentist analyses, 95% confidence intervals and the conventional 5% significance level ($p < .05$) are used for H_0 significance testing. Continuous variables are centred on grand means, or converted to a z-score for the convenience of interpretation. Variables for individual-level predictors (country-mean variables; see section 3) are computed and centred on grand means in order to separate variance of country part from variance of individual part (see Hox, 2010). We will use the variance inflation factor (VIF) as diagnostic for multicollinearity. If $VIF > 3$, we will do the following: If the high VIF concerns a main IV, we will exclude the IV's collinear variable(s). If the high VIF concerns covariates, we will perform a PCA to reduce the number of covariates to a set of uncorrelated covariates.
- For multilevel modelling, the significance of fixed effects is examined using Wald tests with degrees of freedom adjusted with Kenward-Roger method; random effects are tested via likelihood ratio tests ($-2\Delta LL$ with degrees of freedom equal to the number of new random effects variance and covariance). Effect size for fixed effects are examined via pseudo- R^2 for the proportion reduction in each variance component, along with the change in total R^2 , i.e., the squared correlation between actual outcome and predicted outcome by the predictive models (Hoffman, 2015). Moreover, to increase the power of multilevel modelling, we intend to perform the same analyses with two different data sets: one before and one after the respondent-level exclusion screening (see section 8 for criteria). We will primarily present the results of MLM with the whole data set and those with the screened data set in the appendix for reference. We plan to perform MLM with the whole data set before the screening to maximise the statistical power by retaining as many samples as possible.

- For Bayesian analyses, Bayes Factor ≥ 10 ($BF_{10} \geq 10$) is employed, which indicates the presence of strong evidence supporting H_1 against H_0 for Bayesian testing in general (Han et al., 2018; Wagenmakers et al., 2018). In addition to this main Bayes Factor criterion, we intend to use Bayes Factor ≥ 3 ($BF_{10} \geq 3$) that indicates the presence of positive but not strong evidence supporting our hypothesis auxiliary.
- We intend to interpret the outcomes based on both p -values and Bayes Factors. Further details regarding how to make decisions are presented in Table 2. Although p -values and Bayes Factors indicate the same direction of the result (e.g., IF Bayes Factor ≥ 3 , then p should be $< .05$; see Benjamin et al., 2018), if there is an error in either frequentist or Bayesian MLM, they might provide contradictory results. Thus, for robustness reasons, we intend to use both indicators to examine whether the planned MLM analysis is completed without any methodological errors.

Table 2. Bayesian cut-off's criteria with the interpretation.

p -value	Bayes Factor ($BF_{H_{10}}$)	Interpretation
$< .05$	≥ 10	The effect is very strongly supported by evidence.
	$3 \leq BF < 10$	The effect is positively supported by evidence but not strongly.
$\geq .05$	$1/3 < BF < 3$	The current evidence is insufficient to make any decisive decision although the non-zero effect is likely to exist.
	$1/3 < BF < 3$	The current evidence is insufficient to make any decisive decision although the null hypothesis (effect = 0) is likely to be the case.
	$1/10 < BF \leq 1/3$	The null hypothesis (effect = 0) is supported by evidence but not strongly.
	$\leq 1/10$	The null hypothesis (effect = 0) is very strongly supported by evidence.

In the following table (Table 3), we specify the analyses for all hypotheses.

Table 3. An overview of the study's hypotheses and analyses plan.

Question/Hypothesis	Sampling plan (e.g. power analysis)	Analysis	Interpretation given different outcomes
H1a Perceived stress will differ between countries	Given that the analysis that requires the greatest sample size is the planned MLM (see H5) due to its complexity, we intend to follow the sample size estimation for the MLM (see section 6 for further details). In addition, we intend to examine the resultant Bayes Factor ≥ 10 to see whether the corrected evidence is sufficient for hypothesis testing.	We will perform one-way ANOVA to examine the international differences. DV: Perceived stress ([Scale_PSS10]) IV: Country ([country]) IF $p < .05$ is reported, THEN we will perform a post-hoc test with Scheffé's method for exploration. In addition, we will perform the same one-way ANOVA with Bayesian inference with anovaBF function implemented in BayesFactor R package. We will use the non-informative Cauchy prior ($d = .00$, scale = $.707$) that was proposed by Wagenmakers et al. (2018). We will examine whether the resultant Bayes Factor (BF10) ≥ 10, which indicates the strength of evidence that strongly supports the non-zero international difference (Han et al., 2018).	We will test H1b based on both the resultant p-value and Bayes Factor for reasons of completeness. We will interpret whether the effect of perceived stress is non-zero with the criteria provided in Table 2.

H1b Concern over the coronavirus will differ between countries	See descriptions provided in H1a and section 6.	We will perform one-way ANOVA to examine the international differences. DV: Concern over the coronavirus ([Corona_concerns]) IV: Country ([country]) IF $p < .05$ is reported, THEN we will perform a post-hoc test with Scheffe's method for exploration. As for H1a, we will also perform the same one-way ANOVA with Bayesian inference (see descriptions provided in H1a).	See descriptions provided in H1a. We will apply the same criteria to examine whether the effect of the country is non-zero (H1b).
H1c Trust in government efforts to slow the spread of coronavirus will differ between countries	See descriptions provided in H1a and section 6.	We will perform one-way ANOVA to examine the international differences. DV: Trust in country's government efforts ([OECD_institutions]) IV: Country ([country]) IF $p < .05$ is reported, THEN we will perform a post-hoc test with Scheffe's method for exploration. As for H1a, we will also perform the same one-way ANOVA with Bayesian inference (see descriptions provided in H1a).	See descriptions provided in H1a. We will apply the same criteria to examine whether the effect of the country is non-zero (H1c).

H1d Compliance with behavioural guidelines to slow the spread of coronavirus will differ between countries	See descriptions provided in H1a and section 6.	We will perform one-way ANOVA to examine the international differences. DV: Compliance with behavioural guidelines to slow the spread of coronavirus ([Compliance]) IV: Country ([country]) IF $p < .05$ is reported, THEN we will perform a post-hoc test with Scheffe's method for exploration. As for H1a, we will also perform the same one-way ANOVA with Bayesian inference (see descriptions provided in H1a).	See descriptions provided in H1a. We will apply the same criteria to examine whether the effect of the country is non-zero (H1d).
---	--	--	--

H2a Across countries, perceived stress will be negatively correlated with compliance with behavioural guidelines to slow the spread of coronavirus (see the model specified in “Model Specification for Multilevel Model” section, bottom of this table)	See descriptions provided in H1a and section 6.	We will examine a MLM without interaction effects to test the relationship between stress and compliance. DV: Compliance with behavioural guidelines to slow the spread of coronavirus [Compliance] Level 1 IVs: Perceived stress ([Scale_PSS10]), individual demographic variables ([age], [gender], [edu]) Level 2 IVs: Country-level indicators and country demographic variables ([population_size], [GDP per capita],[edu_attainment],[unemployment],[gini_coefficient]) We include [country] as a random intercept in the model. In addition, we will perform the same Bayesian MLM with brms R package. We will examine Bayes Factor of the estimated B for perceived stress at Level 1 is 10 or greater ($BF_{10} \geq 10$). Because we intend to test whether B is non-zero (H_0), we will use 
[revised manuscript text omitted]

Level 2 (Country):

$$P0 = B00 + B01*(PopulationSize) + B02*(GDP) + B03*(EducationAttainment) + B04*(Unemployment) + B05*(Gini Coefficient) + R0$$

$$P1 = B10, P2 = B20, P3 = B30$$

$$P4 = B40 + R1$$

$$P5 = B50$$

$$P6 = B60$$

Compositional Model for Multilevel Analysis:

$$\text{Compliance} = B00 + B10*(Gender) + B20*(Age) + B30*(Education) + B01*(PopulationSize) + B02*(GDP) + B03*(EducationAttainment) + B04*(Unemployment) + B05*(Gini Coefficient) + (B40 + R1)*(Stress) + B50*(SocialProvisions) + B60*(Stress \times SocialProvisions) + R0 + E$$

10. New data or analysis of existing data?

We collect new data. The survey has been pre-registered at the Open Science Framework on March 30, 2020 (COVIDiSTRESS global survey network, 2020), and scheduled for weekly data extractions for public access. The data collection started on March 30, 2020 in all participating countries, with a pre-launch in Denmark on March 26, 2020 to test if the survey works in practice and get a first impression on the response rate. Prior to submitting this RR, we have not accessed or analysed any data, except following response rates to give all COVIDiSTRESS participant researchers feedback on the recruiting progress.

As this RR is submitted under the COVID-19 rapid response call for papers, upon acceptance of the stage one protocol, we will analyse the newest weekly data extraction that fit minimum criteria (see above) for the present analysis. While we intend to stop data collection at the latest on April 13, 2020 for the data analysis for this RR, the COVIDiSTRESS project will continue with the data collection until May 30, 2020. The data collected is made available to any other researcher for further hypotheses testing.

11. Results

To be added in the second stage RR.

12. Discussion

To be added in the second stage RR.

References

Adewuya, A. O., Afolabi, M. O., Ola, B. A., Ogundele, O. A., Ajibare, A. O., Oladipo, B. F., & Fakande, I. (2010). The effect of psychological distress on medication adherence in persons with HIV infection in nigeria. *Psychosomatics: Journal of Consultation and Liaison Psychiatry*, *51*(1), 68-73. doi:10.1016/S0033-3182(10)70661-7

Aldwin, C. M., Sutton, K. J., Chiara, G., & Spiro, A. (1996). Age differences in stress, coping, and appraisal: findings from the Normative Aging Study. *The Journals of Gerontology. Series B, Psychological Sciences and Social Sciences*, *51*, 179-88.

Arend, M. G., & Schäfer, T. (2019). Statistical power in two-level models: A tutorial based on Monte Carlo simulation. *Psychological Methods*, *24*(1), 1–19. Baucom, K. J. W., Queen, T. L., Wiebe, D. J., Turner, S. L., Wolfe, K. L., Godbey, E. I., . . . Berg, C. A. (2015). Depressive symptoms, daily stress, and adherence in late adolescents with type 1 diabetes. *Health Psychology*, *34*(5), 522-530. doi:10.1037/hea0000219

Benjamin, D. J., Berger, J. O., Johannesson, M., Nosek, B. A., Wagenmakers, E. J., Berk, R., ... & Cesarini, D. (2018). Redefine statistical significance. *Nature Human Behaviour*, *2*(1), 6-10. Bish, A., & Michie, S. (2010). Demographic and attitudinal determinants of protective behaviours during a pandemic: a review. *British Journal of Health Psychology*, *15*(4), 797–824.

Boyd, A. D., & Jardine, C. G. (2011). Did public risk perspectives of mad cow disease reflect media representations and actual outcomes? *Journal of Risk Research*, *14*(5), 615-630. doi:10.1080/13669877.2010.547258

Brooks, S. K., Webster, R. K., Smith, L. E., Woodland, L., Wessely, S., Greenberg, N., & Rubin, G. J. (2020). The psychological impact of quarantine and how to reduce it: rapid review of the evidence. *The Lancet*, *395*(10227), 912–920. [https://doi.org/10.1016/S0140-6736\(20\)30460-8](https://doi.org/10.1016/S0140-6736(20)30460-8)

Bults, M., Beaujean, D.J., de Zwart, O., Kok, G., van Empelen, P., van Steenbergen, J. E., Richardus, J. H., & Voeten, H. A. C. M. (2011). Perceived risk, anxiety, and behavioural responses of the general public during the early phase of the Influenza A (H1N1) pandemic in the Netherlands: Results of three consecutive online surveys. *BMC Public Health*, *11*(1), 2. <https://doi.org/10.1186/1471-2458-11-2>

Byrne, B. M., Shavelson, R. J. and Muthén, B. (1989). Testing for the equivalence of factor covariance and mean structures: the issue of partial measurement invariance. *Psychological Bulletin*, *105*, 456-466.

Capelos, T., Provost, C., Parouti, M., Barnett, J., Chenoweth, J., Fife-Schaw, C. & Kelay, T. (2016)
Ingredients of institutional reputations and citizen engagement with regulators. *Regulation &*
*Governance*, 10(4), 350–376. <https://doi.org/10.1111/rego.12097>
Cava, M. A., Fay, K. E., Beanlands, H. J., McCay, E. A., & Wignall, R. (2005). The Experience of
Quarantine for Individuals Affected by SARS in Toronto. *Public Health Nursing*, 22(5), 398–406.
<https://doi.org/10.1111/j.0737-1209.2005.220504.x>
Chaaya, M., Osman, H., Naassan, G., & Mahfoud, Z. (2010). Validation of the Arabic version of the
Cohen Perceived Stress Scale (PSS-10) among pregnant and postpartum women. *BMC Psychiatry*,
10(1), Article 111.
Chen, F., Curran, P. J., Bollen, K. A., Kirby, J., & Paxton, P. (2008). An empirical evaluation of the
use of fixed cutoff points in RMSEA test statistic in structural equation models. *Sociological Methods*
*& Research*, 36(4), 462-494. Chien, P. M., Sharifpour, M., Ritchie, B. W., & Watson, B. (2017).
Travelers' health risk perceptions and protective behavior: A psychological approach. *Journal of*
*Travel Research*, 56(6), 744-759. doi:10.1177/0047287516665479
Cheng, C. and Ng, A.-K. (2006), Psychosocial Factors Predicting SARS-Preventive Behaviors in
Four Major SARS-Affected Regions. *Journal of Applied Social Psychology*, 36, 222–247.
<https://doi.org/10.1111/j.0021-9029.2006.00059.x>
Cheung, G. W., & Rensvold, R. B. (2002). Evaluating goodness-of-fit indexes for testing
measurement invariance. *Structural Equation Modeling: A Multidisciplinary Journal*, 9(2), 233-255.
Cho, Y. J. (2008). *Culture, sex-role, mutual social support and adult attachment as predictors of*
*Korean couples' relationship satisfaction* (Doctoral dissertation, University of Missouri-Columbia).
Coffé, H., & Bolzendahl, C. (2010). Same game, different rules? Gender differences
in political participation. *Sex roles*, 62(5), 318-333.
Cohen, S. (2004). Social Relationships and Health. *American Psychologist*, 59(8), 676–684.
Cohen, S., Kamarck, T., & Mermelstein, R. (1983). A Global Measure of Perceived Stress. *Journal of*
*Health and Social Behavior*, 24(4), 385–396.
Cohen, S., Kamarck, T., & Mermelstein, R. (1983). A global measure of perceived stress. *Journal of*
*Health and Social Behavior*, 24(4), 385-396.
Cohen, S., & Williamson, G. (1988). Perceived stress in a probability sample of the United States. In
S. Spacapan & S. Oskamp (Eds.), *Social Psychology of Health* (pp. 31-67). Sage.
Cohen, S., & Wills, T. A. (1985). Stress, social support, and the buffering hypothesis. *Psychological*
*Bulletin*, 98(2), 310. Curran, P. G. (2016). Methods for the detection of carelessly invalid responses in
survey data. *Journal of Experimental Social Psychology*, 66, 4–19.
<https://doi.org/10.1016/j.jesp.2015.07.006>

Cong, Z., Lv, Q., Yan, J. and Huang, X. (2003) Mental stress and crisis intervention in the patients
with SARS and the people related. *Beijing Da Xue Xue Bao*, 35 (Suppl.), 47–50

Dalton, R.J. (2005) The Social Transformation of Trust in Government. *International*
*Review of Sociology*, 15(1), 133-154.

Dar, A. Wahid. (2017). Cross Cultural Understanding of Stress and Coping Mechanisms.
*International Journal of Research in Engineering, IT and Social Sciences*, 7(10),75–80. Retrieved
from:
[https://www.researchgate.net/publication/331372626_Cross_Cultural_Understanding_of_Stress_and](https://www.researchgate.net/publication/331372626_Cross_Cultural_Understanding_of_Stress_and_Coping_Mechanisms)
[Coping_Mechanisms](https://www.researchgate.net/publication/331372626_Cross_Cultural_Understanding_of_Stress_and_Coping_Mechanisms) [last accessed 05.04.2020].

Davidov, E. (2009). Measurement equivalence of nationalism and constructive patriotism in the ISSP:
34 countries in a comparative perspective. *Political Analysis*, 17(1), 64-82.

Davidov, E., Meuleman, B., Cieciuch, J., Schmidt, P., & Billiet, J. (2014). Measurement equivalence
in cross-national research. *Annual review of sociology*, 40, 55-75.

Deurenberg-Yap, M., Foo, L. L., Low, Y. Y., Chan, S. P., Vijaya, K., & Lee, M. (2005). The
singaporean response to the SARS outbreak: Knowledge sufficiency versus public trust. *Health*
*Promotion International*, 20(4), 320-326. doi:10.1093/heapro/dai010

Dupuis, M., Meier, E., & Cuneo, F. (2019). Detecting computer-generated random responding in
questionnaire-based data: A comparison of seven indices. *Behavior Research Methods*, 51(5),
2228–2237. <https://doi.org/10.3758/s13428-018-1103-y>

Egleston, B. L., Miller, S. M., & Meropol, N. J. (2011). The impact of misclassification due to survey
response fatigue on estimation and identifiability of treatment effects. *Statistics in Medicine*, 30(30),
3560–3572.

Fung, I. C. H., & Cairncross, S. (2007). How often do you wash your hands? A review of studies of
hand-washing practices in the community during and after the SARS outbreak in 2003. *International*
*Journal of Environmental Health Research*, 17(3), 161–183.

Gilles, I., Bangerter, A., Clemence, A., Green, G. T. E., Krings, F., Staerke, C. & Wagner-Egger P.
(2011). Trust in medical organizations predicts pandemic (H1N1) 2009 vaccination behavior and
perceived efficacy of protection measures in the Swiss public. *European Journal of Epidemiology*,
26(3), 203–211

Griffin, R. J., & Dunwoody, S. (2000). The relation of communication to risk judgment and
preventive behavior related to lead in tap water. *Health Communication*, 12(1), 81-107.

Goldman, M., Gier, J. A., & Smith, D. E. (1981). Compliance as affected by task difficulty and order
of tasks. *The Journal of Social Psychology*, 114(1), 75–83.

Gunderson, E. E. (1974). Psychological studies in Antarctica. *Human Adaptability in Antarctic*
*conditions, Antarctic Research Series, Vol. 22*, 115–131.

Han, H., Park, J., & Thoma, S. J. (2018). Why do we need to employ Bayesian statistics and how can
we employ it in studies of moral education?: With practical guidelines to use JASP for educators and
researchers. *Journal of Moral Education*, 47(4), 519–537.

Haushofer, J., & Fehr, E. (2014). On the psychology of poverty. *Science*, 344(6186), 862–867.
doi:10.1126/science.1232491

Hawryluck, L., Gold, W. L., Robinson, S., Pogorski, S., Galea, S., & Styra, R. (2004). SARS Control
and Psychological Effects of Quarantine, Toronto, Canada. *Emerging Infectious Diseases*, 10(7),
1206–1212. <https://doi.org/10.3201/eid1007.030703>

Hitchcock, P. B., Brantley, P. J., Jones, G. N., & McKnight, G. T. (1992). Stress and social support as
predictors of dietary compliance in hemodialysis patients. *Behavioral Medicine*, 18(1), pp. 13–20.

Hoffman, L. (2015). *Longitudinal analysis: Modeling within-person fluctuation and change*. New
York, NY: Routledge Press.

Hox, J. (2010). *Multilevel Analysis: Techniques and Applications*, Mahwah, NJ: Lawrence Erlbaum
Associates.

Hu, L. T., & Bentler, P. M. (1999). Cutoff criteria for fit indexes in covariance structure analysis:
Conventional criteria versus new alternatives. *Structural Equation Modeling: A Multidisciplinary*
*Journal*, 6(1), 1-55.

Huremović, D. (Ed.). (2019). *Psychiatry of Pandemics: A Mental Health Response to Infection*
*Outbreak*. Springer Nature Switzerland.

Karvinen, K. H., Murray, N. P., Arastu, H., & Allison, R. R. (2013). Stress Reactivity, Health
Behaviors, and Compliance to Medical Care in Breast Cancer Survivors. *Oncology Nursing Forum*,
40(2), 149–156. <https://doi.org/10.1188/13.ONF.149-156>

Kehoe, J. P., & Abbott, A. P. (1975). Suicide and attempted suicide in the Yukon territory. *Canadian*
*Psychiatric Association Journal*, 20, 15–23.

Khan, S. & Huremović, D. (Ed.). (2019). Psychology of the Pandemic. In Huremović, D. (Ed.)
*Psychiatry of Pandemics: A Mental Health Response to Infection Outbreak*. (pp. 37–44). Springer
Nature Switzerland.

Kim, S-Y., Kim, J-M., Yoo, J-A., Bae, K-Y., Kim, S-W., Yang, S-J., Shin, I-S., & Yoon, J-S. (2010).
Standardization and validation of Big Five Inventory-Korean Version(BFI-K) in elders. *Korean*
*Journal of Biological Psychiatry*, 17(1), 15–25.

Klein, R. A., Ratliff, K. A., Vianello, M., Adams Jr., R. B., Bahník, Š., Bernstein, M. J., ... Nosek, B.
5 A. (2014). Investigating variation in replicability: A “many labs” replication project. *Social*
*Psychology*, 45(3), 142–152. <https://doi.org/10.1027/1864-9335/a000178>

Krosnick, J. A., & Alwin, D. F. (1987). An evaluation of a cognitive theory of response-order effects
in survey measurement. *Public Opinion Quarterly*, 51(2), 201–219.

Lau, J. T., Yang, X., Tsui, H., Pang, E., & Kim, J. H. (2004). SARS preventive and risk behaviours of
Hong Kong air travellers. *Epidemiology & Infection*, 132(4), 727–736.

Lee, J-H., Shin, C-M., Ko, Y-H., Lim, J-H., Joe, S-H., Kim, S-H., Jung, I-K., Han, C-S. (2012). The
reliability and validity studies of the Korean version of the Perceived Stress Scale. *Korean Journal of*
*Psychosomatic Medicine*, 20(2), 127–134.

Leung, C. M., Lam, T-H., Ho L-M., Ho S-Y., Chan B. H. Y., Wong I., O., L. & Hedley, A. J. (2003).
The impact of community psychological responses on outbreak control for severe acute respiratory
syndrome in Hong Kong. *Journal of Epidemiological Community Health*, 57, 857–863.

Liao, Q., Cowling, B. J., Lam, W. W. T., & Fielding, R. (2011). The influence of social-cognitive
factors on personal hygiene practices to protect against influenzas: Using modelling to compare avian
A/H5N1 and 2009 pandemic A/H1N1 influenzas in hong kong. *International Journal of Behavioral*
*Medicine*, 18(2), 93-104. doi:10.1007/s12529-010-9123-8

Lin, T. T. C., & Bautista, J. R. (2016). Predicting intention to take protective measures during haze:
The roles of efficacy, threat, media trust, and affective attitude. *Journal of Health Communication*,
21(7), 790-799. doi:/10.1080/10810730.2016.1157657

Lomazzi, V. (2018). Using alignment optimization to test the measurement invariance of gender role
attitudes in 59 countries. *Methods, data, analyses: a journal for quantitative methods and survey*
*methodology (mda)*, 12(1), 77-103.

Lynch, D. A. (1999). Sources of stress, perceived stress levels, and levels of self-esteem among career
oriented mothers who are employed full-time, part-time, and those who opted to stay home full-time
(Doctoral dissertation, Seton Hall University, NJ, 1999). *Dissertation Abstracts International*,
60(3-B), 1307.

Mak, W. W., Law, R. W., Woo, J., Cheung, F. M., & Lee, D. (2009). Social support and
psychological adjustment to SARS: The mediating role of self-care self-efficacy. *Psychology &*
*Health*, 24(2), 161–174. <https://doi.org/10.1080/08870440701447649>

Marjanovic, Z., Greenglass, E.R., Coffey S. (2007). The relevance of psychosocial variables and
working conditions in predicting nurses' coping strategies during the SARS crisis: an online
questionnaire survey. *International Journal of Nursing Studies*, 44, 991–998.

Maunder, R. G. (2009). Was SARS a mental health catastrophe? [Editorial]. *General Hospital*
*Psychiatry*, 31(4), 316–317. <https://doi.org/10.1016/j.genhosppsy.2009.04.004>

McHorney, C. A., & Tarlov, A. R. (1995). Individual-patient monitoring in clinical practice: are
available health status surveys adequate?. *Quality of Life Research*, 4(4), 293-307.
Mellins, C. A., & Ehrhardt, A. A. (1994). Families affected by pediatric acquired immunodeficiency
syndrome: Sources of stress and coping. *Journal of Developmental and Behavioral Pediatrics*, 15(3),
54–60.
Milfont, T. L., & Fischer, R. (2010). Testing measurement invariance across groups: Applications in
cross-cultural research. *International Journal of psychological research*, 3(1), 111-130.
Mugavero, M. J., Raper, J. L., Reif, S., Whetten, K., Leserman, J., Thielman, N. M., & Pence, B. W.
(2009). Overload: Impact of incident stressful events on antiretroviral medication adherence and
virologic failure in a longitudinal, multisite human immunodeficiency virus cohort study.
*Psychosomatic Medicine*, 71(9), 920-926. doi:10.1097/PSY.0b013e3181bfe8d2
Neighbors, H. W. (1997). Husbands, wives, family, and friends: Sources of stress, sources of support.
In R. J. Taylor, J. S. Jackson & L. M. Chatters (Eds.), *Family Life in Black America* (pp. 277–292).
Thousand Oaks, CA: Sage Publications.
O'Connor, D. B., Ferguson, E., & O'Connor, R. C. (2005). Intentions to use hormonal male
contraception: The role of message framing, attitudes and stress appraisals. *British Journal of*
*Psychology*, 96(3), 351–369.
Organisation for Economic Co-operation and Development (2017). *OECD guidelines on measuring*
*trust*. OECD Publishing. <http://dx.doi.org/10.1787/9789264278219-en>
Oliver, D. C. (1991). Psychological effects of isolation and confinement of a winter-over group at
McMurdo Station, Antarctica. In *From Antarctica to Outer Space* (pp. 217–227). Springer, New
York, NY.
Palinkas, L. A., & Suedfeld, P. (2008). Psychological effects of polar expeditions. *The Lancet*,
371(9607), 153–163.
Pauliuk, S. (2020). Making sustainability science a cumulative effort. *Nature Sustainability*, 3(1), 2-4.
Perez, V., Uddin, M., Galea, S., Monto, A. S., & Aiello, A. E. (2012). Stress, adherence to preventive
measures for reducing influenza transmission and influenza-like illness. *Journal of Epidemiology and*
*Community Health*, 66(7), 605-610. doi:10.1136/jech.2010.117002
Petersen, M.B. (2020, March 9). The unpleasant truth is the best protection against coronavirus (M. B.
Petersen, Trans.) *Politiken*. Retrieved from
https://pure.au.dk/portal/files/181464339/The_unpleasant_truth_is_the_best_protection_against_coronavirus_Michael_Bang_Petersen.pdf [last accessed 05.04.2020]
Prati, G., Pietrantoni, L. & Zani B. (2011). Compliance with recommendations for pandemic influenza
H1N1 2009: the role of trust and personal beliefs. *Health Education Research*, 26(5), 761–769.

Raue, M., Streicher, B., Lermer, E., & Frey, D. (2015). How far does it feel? Construal level and
decisions under risk. *Journal of Applied Research in Memory and Cognition*, 4(3), 256–264.
Reynolds, D. L., Garay, J. R., Deamond, S. L., Moran, M. K., Gold, W., & Styra, R. (2008).
Understanding, compliance and psychological impact of the SARS quarantine experience.
*Epidemiology and Infection*, 136(7), 997–1007. <https://doi.org/10.1017/S0950268807009156>
Ro, J-S., Lee, J-S., Kang, S-C., Jung, H-M. (2017). Worry experienced during the 2015 Middle East
Respiratory Syndrome (MERS) pandemic in Korea. *PLoS ONE*, 12(3), e0173234.
<https://doi.org/10.1371/journal.pone.0173234>
Roberti, J. W., Harrington, L. N., & Storch, E. A. (2006). Further psychometric support for the
10-item version of the Perceived Stress Scale. *Journal of College Counseling*, 9(2), 135-147.
Roudier, J. N., & Morey, R. D. (2012). Default Bayes factors for model selection in regression.
*Multivariate Behavioral Research*, 47(6), 877–903.
Singh, N., Squier, C., Sivek, C., Wagener, M., Nguyen, M. H., & Yu, V. L. (1996). Determinants of
compliance with antiretroviral therapy in patients with human immunodeficiency virus: Prospective
assessment with implications for enhancing compliance. *AIDS Care*, 8(3), 261-269.
doi:10.1080/09540129650125696
Schütte, S., Chastang, J. F., Parent-Thirion, A., Vermeylen, G., & Niedhammer, I. (2014). Social
inequalities in psychological well-being: A European comparison. *Community Mental Health Journal*,
50(8), 987-990.
Seligman, A. B. (1997). *The Problem of Trust*. Princeton, NJ: Princeton University Press
Steigen, A. M., & Bergh, D. (2019). The Social Provisions Scale: psychometric properties of the
SPS-10 among participants in nature-based services. *Disability and Rehabilitation*, 41(14),
1690–1698. <https://doi.org/10.1080/09638288.2018.1434689>
Taylor, Steven. 2019 *The Psychology of Pandemics: Preparing for the Next Global Outbreak of*
*Infectious Disease*. Cambridge Scholars
Taha, S. A., Matheson, K., & Anisman, H. (2013). The 2009 H1N1 influenza pandemic: The role of
threat, coping, and media trust on vaccination intentions in Canada. *Journal of Health Communication*,
18(3), 278-290. doi:10.1080/10810730.2012.727960
The COVIDiSTRESS global survey network (2020). *COVIDiSTRESS global survey pre-registration*.
<https://doi.org/10.17605/osf.io/z39us>.
Trautmann, S. T., & van de Kuilen, G. (2012). Prospect theory or construal level theory?: Diminishing
sensitivity vs. psychological distance in risky decisions. *Acta Psychologica*, 139(1), 254–260.

Trope, Y., & Liberman, N. (2010). Construal-level theory of psychological distance. *Psychological*
*Review*, 117(2), 440.

Van't Riet, J., Cox, A. D., Cox, D., Zimet, G. D., De Bruijn, G., Van den Putte, B., . . . Ruiter, R. A.
C. (2014). Does perceived risk influence the effects of message framing? A new investigation of a
widely held notion. *Psychology & Health*, 29(8), 933–949.

Wagenmakers, E.-J., Love, J., Marsman, M., Jamil, T., Ly, A., Verhagen, A. J., Selker, R., Gronau, Q.,
Dropmann, D., Boutin, B., Meerhoff, F., Knight, P., Raj, A., Kesteren, E.-J., Van Doorn, J., Šmíra, M.,
Epskamp, S., Etz, A., Matzke, D. & Morey, R. (2017). Bayesian inference for psychology. Part II:
Example applications with JASP. *Psychonomic Bulletin & Review*, 25, 1–19.
<https://doi.org/10.3758/s13423-017-1323-7>

Wang, X., Cai, L., Qian, J., & Peng, J. (2014). Social support moderates stress effects on depression.
*International Journal of Mental Health Systems*, 8(1), 41. <https://doi.org/10.1186/1752-4458-8-41>

Weiss RS. The provisions of social relationships. In: Rubin Z, editor. Doing unto others: joining,
molding, conforming, helping, loving. Englewood Cliffs, N.J.: Prentice-Hall; 1974. p 17-26.

Wong, T. W., Gao, Y., & Tam, W. W. S. (2007). Anxiety among university students during the SARS
epidemic in Hong Kong. *Stress and Health: Journal of the International Society for the Investigation*
*of Stress*, 23(1), 31–35.

Wong C. M, L. & Jensen O. (2020): The paradox of trust: perceived risk and public compliance
during the COVID-19 pandemic in Singapore. *Journal of Risk Research*.
<https://doi.org/10.1080/13669877.2020.1756386>

Appendix E

Response letter

Associate Editor

1. The revised manuscript was returned to four of the original reviewers. The good news is that the manuscript is now substantially closer to IPA, however some there remain some wrinkles to iron out, especially in addressing the points raised by Reviewer 5 (including the concerns in point 1, 2, 6 and 11, all of which must be satisfied). Reviewer 1 and 2 also offer a range of constructive suggestions for improving clarity, avoiding unnecessary spin, and citing additional literature.

Dear Drs. Chambers and Dunn,

Thank you for the quick treatment of the newest version of our manuscript.

We have read all the referee comments carefully, and believe that we have found ways of addressing them all.

The text has also updated more generally to correct minor errors and accommodate reviewer perspectives.

We have paid specific attention to the points raised by reviewer 5. To this end we have responded below, and now complied with all the specific suggestions. We have, however, also submitted an alternative proposal for an updated sampling strategy (reviewer 5's point 5, originally 11), to avoid this RR unintentionally becoming a longitudinal study or introducing noise from changing Coronavirus circumstances over time. We would like your consideration in this matter, but will accept the reviewer's suggestion of using all data available (as now implemented in the revised RR) if that is a hard condition for progression to stage two.

Further, based on Taciano Milfont's helpful comments and offer to assist us with measurement invariance analysis and alignment techniques, we have invited him to join our group as a co-author in the next stage. As you suggested, we will make a note of this in the following iteration of the paper.

We thank you for all your help in the process, and look forward to the reviewers' responses.

Best regards

The COVIDiSTRESS team

Reviewer 1

1. *The new section on measurement invariance testing is very good. The authors elected to follow a traditional multigroup confirmatory factor analysis approach, but practical applications with cross-cultural data indicate it is hard to achieve scalar invariance with complex measures and several groups, which is the case in this planned study. For this reason, I recommend the authors to mention the alignment method as an alternative if scalar invariance is not achieved (this comparison article might be informative: <https://journals.sagepub.com/doi/full/10.1177/0022022117697844>). Or the authors could mention the new approach proposed by Marsh et al. (2018; the alignment-within-CFA approach) that combines both: <https://www.ncbi.nlm.nih.gov/pubmed/28080078>.*

Thank you very much for your suggestion for the use of the alignment method. We have added information about an alternative approach to testing the equivalence of invariance as suggested (i.e. using the alignment method), if the traditional approach fails. More specifically, we elaborated how to adjust factor scores to be analyzed when the traditional multigroup CFA fails to achieve the measurement invariance with the alignment method implemented in *sirt*.

2. *I recommend mentioning the measurement invariance tests briefly in the analytical plan (section 10).*

We agree with the Reviewer that measurement invariance tests should be mentioned in the analytical plan. We have added it accordingly.

3. *There is a recently published review piece the authors should cite: <https://www.nature.com/articles/s41562-020-0884-z>. We have also a COVID-19 manuscript under the second round of revisions in *American Psychologist* the authors could consider citing; the pre-print is available here: <https://psyarxiv.com/cx6qa>*

We would like to thank the reviewer for bringing these to our attention. We now include Van Bavel et al. in discussing the implications on cultural variance and reactions to stress. We have also included the Sibley et al. article in our discussion on the studies that explore the implications of COVID-19 focusing in particular on their findings related to trust.

4. I have made small suggestions in the PDF file, uploaded to the system. Thank you - these were helpful.

Reviewer 2

1. One weakness of the current draft, although not a condition for acceptance for me, is how the sampling is oversold, e.g., "...leverage fast organic sampling, which favours a varied reach over predetermined population criteria" and in the author response: "organic sampling rather than e.g. purchased responses from predetermined subject pools". These sentences are more marketing copy than clear scientific writing, particularly "organic". I assume the sampling strategy is mostly due to convenience (lower cost), and that is okay and even okay to write. Beyond that, if there are advantages to the sampling strategy, then please make them explicit and stated clearly in separate sentences. For example, two implied problems are participants being paid and participants being fatigued. Please justify how either might influence the specific hypotheses here before claiming that the other strategy is better. Otherwise, I assume the loss of population representativeness and therefore generalizability compared to paid, national samples is the main difference between strategies. This can also be stated directly (Discussion would be okay).

We understand your point. We have rewritten our description of the sampling process to reflect this, while still making the limits and characteristics of this specific sample clear.

2. *Next, "Colleagues from the consortium hope that causal relationships can be studied further down the line, as longitudinal data becomes available from the dataset." This reflects a lack of planning. It is possible to design those tests now. This is not a condition for acceptance.*

The lack of top-down planning is by design. A large group of researchers have been involved in collecting the data and designing the survey as an open science endeavor. To allow us to progress rapidly in response to COVID-19, we worked in a fairly flat hierarchy to allow different people to follow their strengths and interests, and keep the dataset open to new approaches. Time has progressed somewhat since the original submission, however, so these analyses *are* being planned now.

3. *Also, "We are experiencing an unprecedented outbreak of a new pandemic." might not read as well in 2022 or 2023. I suggest revision.*

Thank you for raising this point. We have rephrased this introductory section with more neutral language.

4. Use spellcheck again, e.g., "complex.Worry"

Thanks - we have had another round of corrections.

Reviewer 5

Thank you for all your good points. We hope that they are addressed to your satisfaction.

1. Point 1: Corona concerns also needs to be tested for invariance, in a similar way as perceived stress. The five items are implicitly assumed to reflect one underlying factor. SPS is also a multi-item scale that should be tested in this way.

We agree with the reviewer, and this was our intention. We have specified testing the invariance equivalence for all the scales, i.e. PSS-10, coronavirus concerns and SPS in the Section 8 and 10.

2. Point 2: I did not understand the authors' response to this concern. I no longer find any mention of centering in the manuscript, and the noted response in Section 6 does not appear to be there. I believe this concern stands as previously noted.

We are sorry that we didn't make them clear enough. The centering method is mentioned under the 6th bullet point of Data Analysis under Analysis Plan. We note that we will use grand-mean centering for all the individual-level continuous variables. In addition, as we agree with your suggestion about reporting quasi-r-squared, we further described the methods for examining effect sizes: "Effect size for fixed effects are examined via pseudo-R-squared for the proportion reduction in each variance component, along with the change in total R-squared" under the 7th bullet point of Data Analysis.

As for Section 6 (Power Analysis), we should have highlighted the noted response in our revised RR, when we stated that "*as we are aware of the issue that random effects are often distorted while the variables are standardised, the random effects were estimated with grand-mean centred continuous variables. The same multilevel analyses were run again with standardised variables only in order to obtain the standardised estimates of fixed effects.*" In Section 6, we run multilevel analysis one time with standardising method only for the purpose of obtaining standardised estimates required for the guidelines from Arend and Schäfer (2019). For all the planned analyses listed in the Analysis Plan, we will use only grand-mean centering, as we are highly aware of and totally agree with you upon the issue you have raised.

3. Point 4: I disagree that the severity of infection in each country would not influence individual behavior. It seems obvious that people would change their behavior more if the perceived threat is higher. Nonetheless, the authors are free to disagree, so this concern is addressed.

Thank you. We see your point, and are not proposing that the rates of infection cannot be important. However, in this study, we limit the scope of our hypotheses to more subjective variables.

4. Point 6: This is not an optional change. The model is misspecified if the authors do not include a random intercept for country. My recommendation remains to test the models using the nested models previously described. Post hoc between country comparisons can still be done if there are significant random intercepts. Perhaps it

would help the authors conceptualize it to think of the analysis as a mixed ANOVA rather than a multi-level model (although these are equivalent statistically).

We will follow the recommendation. The analysis for H1a to H1d has been revised accordingly - We will include random intercepts for countries (i.e., each country has its own distinct intercept) in the model. After the inclusion of country random intercepts, we will test the fixed effect of countries, and as you proposed, we will do post-hoc comparisons to identify the between-country differences.

5. Point 11: The authors do not appear to have modified their sampling strategy and intend to focus only on the data collected through April 6. I continue to think they should use all available data at the time of Stage 1 RR acceptance.

This makes some sense, given how more data is now available. We have updated the text and timeline to use the most recent data extraction at the time of acceptance to comply with your recommendation.

We would, however, prefer to limit our data extraction to a narrower window, as this analysis was not conceived as longitudinal. If the reviewer agrees, we propose using a set data window limited strictly to the first two weeks, i.e. until April 13. Would this be an acceptable alternative?

Either way, since time has passed, we have eliminated the notion of adding another week if too few countries have reached a minimum of participants. Instead we now exclude any countries that do not live up to the minimums required by our analyses at the time of extraction.

Appendix F

Dear Drs. Chambers and Dunn,

Thank you for handling and reviewing our manuscript on the *COVIDiSTRESS global survey* so far.

We hereby submit our stage two report for your consideration and review.

The process was slowed somewhat by the nature of our large globally distributed consortium, but we have now completed the stage two manuscript, including all the planned analyses and reviewer recommendations from stage one of the RR review process.

Access to the full COVIDiSTRESS global survey including weekly data extractions, code books and information about the consortium is available at <https://osf.io/z39us/>. Royal Society Open Science registration files including electronic supplementary materials and the *in principle accepted* stage one report are found at <https://osf.io/vz73d/>. Both are linked on p. 5 of the manuscript.

As specified in the stage one report, data collection was ongoing at the time of submission to RSOS, but we confirm that no data for the present analysis was collected after in principle approval.

We have left heading 5 *Conditions of assignment* with no content, to conform with Dr Chambers' Registered Report template, even though the design had no experimental conditions. Would you like this to remain, or should we delete it in the next version? If you could let us know, we will update the section accordingly in the next version of the manuscript.

Note: The current list of authors includes only those who have worked specifically on the present paper, but not the broader COVIDiSTRESS-consortium who all contributed to the design, translation and local data collection. As mentioned earlier, we would like to add all these contributors at a later stage, once the final review is complete.

Again, thank you very much for your time and consideration.

Sincerely,

On behalf of the COVIDiSTRESS consortium,

Andreas Lieberoth
Corresponding author

Appendix G

I reviewed RSOS-200589 (Stage 2 RR). Overall I find this project super exciting. It is an amazing consortium reflecting major collaborative and logistic efforts, culminating in a useful and unique dataset. Well done and I'm looking forward to it being shared. I think there is still major revision to do before finishing Stage 2, though.

1. *Whether the data are able to test the authors' proposed hypotheses by passing the approved outcome-neutral criteria (such as absence of floor and ceiling effects or success of positive controls or other quality checks). Failure to pass these conditions may lead to manuscript rejection.*

The floor and ceiling effects did not seem to pose a major threat to the analyses or interpretation. More description of the normality of major variables would be helpful, especially of the novel composites (e.g., through histograms, in the Supplement if appropriate).

2. *Whether the Introduction, rationale and stated hypotheses are the same as the approved Stage1 submission (required).*

Confirmed.

3. *Whether the authors adhered precisely to the registered experimental procedures.*

Yes.

Exclusions 1: so far as I can tell, the pre-registered plan was to include any country above $N = 30$ (see "7. Exclusion"), but the final analysis only includes countries above $N = 100$. If this is a deviation from the original plan, this should be disclosed more transparently.

Exclusions 2: why use 10% of the estimated survey duration length rather than the actual length, now that it is available? Maybe I misunderstood that the plan was always to use the estimated data. At the least, please compare this estimated value here to the actual median or other central estimate.

4. *Where applicable, whether any unregistered exploratory statistical analyses are justified, methodologically sound, and informative.*

Alignment process in measurement invariance: the logic appears sound to me, but please state clearly for the readers that using these adjusted factor scores was a deviation from the Stage 1 submission. Perhaps another reviewer could weigh in on whether this needs further justification or explanation.

5. *Whether the authors' conclusions are justified given the data. Please note that editorial decisions will not be based on the perceived importance, novelty, or conclusiveness of the results.*

Yes, but overall the description and interpretation of results overly focused on significance. Especially with large data like this, there's an opportunity to more precisely estimate effect sizes, compare them with previous findings, and interpret the meaning of their size.

Abstract

The abstract has several typos and needs to be copyedited more before acceptance, for example: "*map e.g. how e.g. different*". There are also typos elsewhere, like obviously in the first sentence of the section 'Number of observations to be collected and rule of termination', and less obviously, e.g., "the key hypotheses testing analyses" or "as more formal hypothesis test including".

Results

In general, I found the results relatively too focused on inferences combining variables. This dataset is an amazing resource for descriptives of key variables, and comparisons across countries. For example, can the reader easily determine the answer to questions like: what was the level of concern about the virus at this time? This would support the somewhat under-justified argument in the Discussion that these results are policy-relevant.

Please change "average" to "mean" where that estimate is described.

Figure 1: please change the legend to make it more clear which country is higher stress in a comparison. Am I reading this correctly that the Netherlands is reporting high stress compared to almost everyone?

Figure 3: it might be confusing to some readers that the scale differs in this figure compared to Figure 1 and Figure 2. Consider whether it makes sense to remind the reader of the range of this measure (optional).

Figures 5-9: I'm confused by these. They seem to only show the trend line, not variation in the underlying data. Surely the countries don't all fit the trend line this exactly on every variable? In the current state, these are not very informative and should be dropped. Could data points be included as well, for example in a low-contrast cloud? I don't require any specific fix here, but it looks like something should change.

What's the justification for analyzing and graphing some variables in unstandardized units (e.g., trust in Figure 3) and some in standardized units (e.g., social support in Figure 10)? I might have missed this logic.

When reporting compliance, I would suggest adding "self-reported" more often. This being such a complex report, it might be easy to forget that this was not objectively measured.

Data

It's currently difficult to locate the final dataset in OSF, mostly because the combined data folder is within the country-level data folder. Why is this? I downloaded the data and looked at it and the codebook.

Discussion

I don't think the hypotheses should be re-stated here, although I'm happy to be overruled on this. Perhaps the results section would benefit from a summary table of the hypotheses and whether they were met (broadly speaking).

I was underwhelmed by the Discussion. It restates the key results but doesn't go far in integrating the main findings with other findings about risk and concern across countries, e.g., by the Winton Centre (but many others). Do the main estimates here, e.g., which countries are most concerned, align with other empirical efforts? That is, there is little attention to the convergent validity of the results with other projects prior to the inferential hypothesis testing. At the other end of this point, a statement like "This is extremely intuitive to stress researchers" (line 41, pp 47 of 116) does not demonstrate such validity, but merely appeals to unspecified authority.

I didn't find the last two paragraphs very useful, and would have preferred more suggestions for future research or further implications of the current findings.

Appendix H

Dear Drs. Chambers and Dunn,

Thank you very much for the prompt treatment. We feel fortunate that the stage one reviewer was available, even in these busy and trying times.

We hereby submit a revised version of our manuscript. We are grateful for your and the reviewers' feedback, and believe that the comments and suggestions have indeed made the revised paper stronger than the previous version.

As discussed in our email, I have added the consortium co-authorship statement at the very end of the document, as well as a link to a full list of contributor details on the OSF site.

Below we provide a detailed response to your and the reviewers' comments.

Thank you very much for considering the revised sections of our manuscript. We look forward to hearing from you.

Sincerely,

Andreas Lieberoth & COVIDiSTRESS global survey team

Responses to Associate Editor Prof. Christopher Chambers,

Thank you for evaluating our manuscript and providing feedback.

Concerning the reviewer's comment that "the results section would benefit from a summary table of the hypotheses and whether they were met" -- I agree and please either add such a table (abbreviated form of Table 3) or augment the existing Table 3 to include the results in an additional column (which would require moving Table 3 to the main text).

This is a good suggestion. We added a short overview of the results in the form of an additional column in Table 3.

Responses to Reviewer

I reviewed RSOS-200589 (Stage 2 RR). Overall I find this project super exciting. It is an amazing consortium reflecting major collaborative and logistic efforts, culminating in a useful and unique dataset. Well done and I'm looking forward to it being shared. I think there is still major revision to do before finishing Stage 2, though.

Thank you for evaluating our manuscript and providing feedback.

1. Whether the data are able to test the authors' proposed hypotheses by passing the approved outcome-neutral criteria (such as absence of floor and ceiling effects or success of positive controls or other quality checks). Failure to pass these conditions may lead to manuscript rejection.

The floor and ceiling effects did not seem to pose a major threat to the analyses or interpretation. More description of the normality of major variables would be helpful, especially of the novel composites (e.g., through histograms, in the Supplement if appropriate).

We now provide an overview of the distribution of our major variables in the Supplementary Material (Figures S1-S4). In the manuscript, we refer to this overview in section 8 (quality checks).

2. Whether the Introduction, rationale and stated hypotheses are the same as the approved Stage 1 submission (required).

Confirmed.

3. Whether the authors adhered precisely to the registered experimental procedure

Yes.

Exclusions 1: so far as I can tell, the pre-registered plan was to include any country above N = 30 (see "7. Exclusion"), but the final analysis only includes countries above N = 100. If this is a deviation from the original plan, this should be disclosed more transparently.

As the Editor and Stage 1 reviewers may remember, the original predictions considered mean differences in the key constructs across countries, but we did not consider measurement invariance in the first versions of the Stage 1 manuscript. It became clear to us that we needed to examine the equivalence of the measures before comparing means -- and this issue was pointed out by one of the Stage 1 reviewers. After discussion with all other co-authors and the Editor, we invited this reviewer (Taciano L. Milfont) to join as a co-author and to contribute with the measurement invariance analyses to the Stage 2 manuscript. A sample of 30 per country was considered too small for measurement invariance testing and it was decided to include 100 participants per country, which would give a reasonable number of responses per scale items. We now explicitly mention that this procedure changed from the original plan due to a valuable comment from a reviewer. In

fact, the modification was inevitable in order to perform the measurement invariance test required by the reviewer.

Exclusions 2: why use 10% of the estimated survey duration length rather than the actual length, now that it is available? Maybe I misunderstood that the plan was always to use the estimated data. At the least, please compare this estimated value here to the actual median or other central estimate.

We preregistered to exclude participants who completed the whole survey in less than 2 minutes and 12 seconds (i.e. 10% of the time that Qualtrics expected; see section 7 in our manuscript). Given that we specified this procedure in the accepted stage one registered report, we would prefer to keep it. Nevertheless, we agree that it is reasonable to analyse whether our procedure deviates from procedures that use data-based estimates (like you suggest). By doing so, we have found that the outcome of our procedure does not strongly deviate from the procedure that you suggested: To examine how the filtering criterion performed, we examined the actual duration. The median of the actual duration was 2118.38 seconds. Our cutoff, 132 seconds (i.e. 2 minutes and 12 seconds, 10% of the time that Qualtrics expected), corresponded to the top 1% of all participants. It suggests that we filtered out the top 1% fastest responses from the planned analyses when the criterion was applied.

Based on this added calculation, we conclude that our initial strategy was appropriate.

4. Where applicable, whether any unregistered exploratory statistical analyses are justified, methodologically sound, and informative.

Alignment process in measurement invariance: the logic appears sound to me, but please state clearly for the readers that using these adjusted factor scores was a deviation from the Stage 1 submission. Perhaps another reviewer could weigh in on whether this needs further justification or explanation.

Thank you for noting this issue. We have made it clear in the revised Stage 2 manuscript that using latent scores from the alignment measurement invariance analysis was a deviation of Stage 1 submission. After comments about the need to include measurement invariance testing before testing our hypothesis, we added information about this analysis using a traditional approach. However, the literature makes it clear that the traditional measurement approach is problematic when multiple countries are considered (e.g., see Byrne & van de Vijver, 2017). We are comfortable with this departure from the original plan, as the latent scores used are supported by Monte Carlo simulations -- as detailed in the Supplementary Material.

5. Whether the authors' conclusions are justified given the data. Please note that editorial decisions will not be based on the perceived importance, novelty, or conclusiveness of the results.

Yes, but overall the description and interpretation of results overly focused on significance. Especially with large data like this, there's an opportunity to more precisely estimate effect sizes, compare them with previous findings, and interpret the meaning of their size.

We sincerely appreciate your suggestion about reporting effect sizes in addition to the results of significant tests. We calculated effect sizes of predictors of interest according to your suggestion. In the cases of H1a-H1d, the effect size of the country in each hypothesis testing was already reported in terms of ICC. While revising the manuscript, we calculated Cohen's Ds of predictors for H2a-H5 by employing `lme.dscore` implemented in an R package, `EMAtools`, that estimates Cohen's D in MLM. The related procedures are now described in "9.4 Effect size analysis". In addition, in each subsection in the results section reporting the result of each hypothesis testing, we added the calculated Cohen's D for additional information.

Abstract

The abstract has several typos and needs to be copyedited more before acceptance, for example: "map e.g. how e.g. different". There are also typos elsewhere, like obviously in the first sentence of the section ' Number of observations to be collected and rule of termination', and less obviously, e.g., " the key hypotheses testing analyses" or "as more formal hypothesis test including".

Thank you for pointing this out. We edited the abstract.

Results

In general, I found the results relatively too focused on inferences combining variables. This dataset is an amazing resource for descriptives of key variables, and comparisons across countries. For example, can the reader easily determine the answer to questions like: what was the level of concern about the virus at this time? This would support the somewhat under-justified argument in the Discussion that these results are policy-relevant.

Please change "average" to "mean" where that estimate is described.

We changed this throughout the whole manuscript.

Figure 1: please change the legend to make it more clear which country is higher stress in a comparison. Am I reading this correctly that the Netherlands is reporting high stress compared to almost everyone?

We added a subtitle within each figure to clearly point out the meaning of the color-coded tiles.

Figure 3: it might be confusing to some readers that the scale differs in this figure compared to Figure 1 and Figure 2. Consider whether it makes sense to remind the reader of the range of this measure (optional).

We added short reminders about the original scale to the figures. We think that it is now less confusing for readers. Thank you.

Figures 5-9: I'm confused by these. They seem to only show the trend line, not variation in the underlying data. Surely the countries don't all fit the trend line this exactly on every variable? In the current state, these are not very informative and should be dropped. Could data points be included as well, for example in a low-contrast cloud? I don't require any specific fix here, but it looks like something should change.

We appreciate your comment regarding Figures 5-9 that could be confusing among potential readers. Given the large data size and complexity of the testing results, we decided to remove the figures from the revised manuscript as you suggested.

What's the justification for analyzing and graphing some variables in unstandardized units (e.g., trust in Figure 3) and some in standardized units (e.g., social support in Figure 10)? I might have missed this logic.

Thank you very much for your comment regarding the units used in figures.

Because of our hypothesis testing procedures and the nature of each dependent variable, the created figures might present different types of variables. For instance, in the case of trust in Figure 3, we did not have to perform measurement alignment because trust was measured with only one single item. As a result, the unit of the variable in Figure 3, trust, reflects the raw score. However, when we examined variables that were measured with multiple items (e.g., social support in Figure 10), we had to perform measurement alignment to enable cross-cultural comparison of the constructs. As a result, those figures demonstrate the aligned dependent variables in the y-axis unlike Figure 3 that demonstrates trust in a raw score. In the revised manuscript, we clarified which variables were influenced by measurement alignment in figure captions.

When reporting compliance, I would suggest adding "self-reported" more often. This being such a complex report, it might be easy to forget that this was not objectively measured.

Thank you for pointing this out. We added "self-reported" more often.

Data

It's currently difficult to locate the final dataset in OSF, mostly because the combined data folder is within the country-level data folder. Why is this? I downloaded the data and looked at it and the codebook.

We agree and therefore decided to provide a direct link to the final version of the cleaned data file (<https://osf.io/cjxua/>).

Discussion

I don't think the hypotheses should be re-stated here, although I'm happy to be overruled on this. Perhaps the results section would benefit from a summary table of the hypotheses and whether they were met (broadly speaking).

Thank you very much for your suggestion to improve the presentation of hypothesis testing results. We also referred to the associate editor's suggestion of updating Table 3, the table describing hypotheses and hypothesis testing procedures, to report test results in a more integrative manner. Hence, In the revised manuscript, in Table 3, we added one more column presenting whether each hypothesis was supported.

I was underwhelmed by the Discussion. It restates the key results but doesn't go far in integrating the main findings with other findings about risk and concern across countries, e.g., by the Winton Centre (but many others). Do the main estimates here, e.g., which countries are most concerned, align with other empirical efforts? That is, there is little attention to the convergent validity of the results with other projects prior to the inferential hypothesis testing. At the other end of this point, a statement like "This is extremely intuitive to stress researchers" (line 41, pp 47 of 116) does not demonstrate such validity, but merely appeals to unspecified authority.

I didn't find the last two paragraphs very useful, and would have preferred more suggestions for future research or further implications of the current findings.

We agree with your comment that the discussion has some potential for improvement. In line with your comment, we refrain from restating all our results in detail. Further, we now put our findings in the context of other findings. This included the last two paragraphs.

References

Byrne, B. M., & van de Vijver, F. J. R. (2017). The maximum likelihood alignment approach to testing for approximate measurement invariance: A paradigmatic cross-cultural application. Psicothema, 29(4), 539–551.